# FEDERATED ZEROTH-ORDER OPTIMIZATION USING TRAJECTORY-INFORMED SURROGATE GRADIENTS

## ABSTRACT

Federated optimization, an emerging paradigm which finds wide real-world applications such as federated learning, enables multiple clients (e.g., edge devices) to collaboratively optimize a global function. The clients do not share their local datasets and typically only share their local gradients. However, the gradient information is not available in many applications of federated optimization, which hence gives rise to the paradigm of federated *zeroth-order optimization* (ZOO). Existing federated ZOO algorithms typically suffer from the limitations of query and communication round inefficiency, which can be attributed to (a) their reliance on a substantial number of function queries for gradient estimation and (b) the significant disparity between their realized local updates and the intended global updates. To this end, we (a) introduce *trajectory-informed gradient surrogates* which is able to use the history of function queries during optimization for accurate and query-efficient gradient estimation, and (b) develop the technique of *adaptive gradient correction* using these gradient surrogates to mitigate the aforementioned disparity. With these, we propose the *federated zeroth-order optimization using trajectory-informed surrogate gradients* (FZooS) algorithm for query- and communication round-efficient federated ZOO. FZooS achieves theoretical improvements over the existing approaches, which is supported by our real-world experiments on federated black-box adversarial attack and non-differentiable metric optimization.

## 1 INTRODUCTION

Due to the growing computational power of edge devices and increasing privacy concerns, recent years have witnessed a surging interest in *federated optimization*, which finds real-world applications such as federated learning (McMahan et al., 2017a). Federated optimization allows the agents to retain their local datasets but share their locally computed gradients. However, in many important applications of federated optimization such as federated black-box adversarial attack (Fang et al., 2022), the gradient information is not available. This gives rise to the paradigm of federated *zeroth-order optimization* (ZOO), in which the global function to be optimized is an aggregation of the local functions that are distributed on edge devices (i.e., clients) and are only accessible via function queries (Fang et al., 2022). To tackle federated ZOO, existing algorithms (Fang et al., 2022) follow the framework of using *finite difference* (FD) for local gradient estimation and hence resorting to federated *first-order optimization* (FOO) algorithms (e.g., FedAvg (McMahan et al., 2017b)) for optimization.[1] Nevertheless, these algorithms usually suffer from both query and communication round inefficiency for local and global functions that are not only expensive-to-evaluate but also heterogeneous. This impedes their practical applicability, especially in the scenarios with restricted query times and communication rounds. However, little attention has been dedicated to achieving query- and communication round-efficient federated ZOO algorithms in the literature.

To address this problem, it is imperative to firstly identify the challenges faced by existing federated ZOO algorithms which are responsible for their query and communication round inefficiency (Sec. 3). Federated ZOO requires multiple *communication* rounds for central server aggregation; between consecutive communication rounds, every client performs several iterations of local optimization

---

[1]So, existing federated FOO algorithms (e.g., FedProx (Li et al., 2020a), SCAFFOLD (Karimireddy et al., 2020a) and etc.) can be easily adapted to this framework (refer to Sec. 3). We refer to this simple integration of FD methods and federated FOO algorithms as the *existing federated ZOO algorithms* throughout this paper.

using their estimated gradients which are usually approximated via additional function *queries* (e.g., based on FD). Firstly, we show (Sec. 3) that the *query inefficiency* of existing federated ZOO algorithms arises from their employment of FD for local gradient estimation, which often requires an excessive number of additional function queries. Therefore, addressing the challenge of query efficiency in federated ZOO calls for a gradient estimation method that requires minimal (ideally zero) additional function queries. Secondly, we show (Sec. 3) that the *communication round inefficiency* of these existing algorithms results from the disparity between their realized local updates and the intended global updates, which is typically caused by client heterogeneity. Hence, resolving the challenge of communication round efficiency requires developing a high-quality gradient correction technique to mitigate such a disparity.

To this end, we propose the *federated zeroth-order optimization using trajectory-informed surrogate gradients* (FZooS) algorithm to address the aforementioned challenges, and hence to achieve query- and communication round-efficient federated ZOO. Firstly, we introduce the recent *derived Gaussian process* (Shu et al., 2023), which only requires the optimization trajectory (i.e., the history of function queries during optimization) for gradient estimation, as the local gradient surrogates for the clients, thereby realizing query-efficient gradient estimation in federated ZOO (Sec. 4.1). Secondly, based on these local gradient surrogates, we use *random Fourier features* (RFF) approximation (Rahimi and Recht, 2007) to produce a transferable global gradient surrogate (without transferring raw observations), which is an accurate estimate of the gradient of the global function (Sec. 4.2.1). Using these surrogates, we develop the technique of *adaptive gradient correction* using adaptive gradient correction vector and length to mitigate the disparity between our local updates and the intended global updates, and consequently to improve the communication round efficiency of federated ZOO (Sec. 4.2.2).

We verify that our FZooS has addressed the aforementioned challenges via both theoretical analysis and empirical experiments. We firstly theoretically bound the disparity between our realized local updates in FZooS and the intended global updates in the federated ZOO problems with heterogeneous clients. It shows that our local update is superior to those employed by the previous works because it achieves both a better query efficiency and smaller disparity error (Sec. 5.1). Based on this, we then prove the convergence of our FZooS and show that FZooS also enjoys an improved communication round efficiency over the existing algorithms (Sec. 5.2). Lastly, we use extensive experiments, such as synthetic experiments, federated black-box adversarial attack and federated non-differentiable metric optimization, to show that our FZooS consistently outperforms the existing federated ZOO algorithms in terms of both query efficiency and communication round efficiency (Sec. 6).

## 2 PROBLEM SETUP AND NOTATIONS

In the federated *zeroth-order optimization* (ZOO) setting (Fang et al., 2022), we aim to minimize a global function $F$ defined on the domain $\mathcal{X} \triangleq [0,1]^d$, which is the arithmetic average of $N$ local functions $\{f_1, \cdots, f_N\}$ distributed on $N$ different clients with $|f_i(\boldsymbol{x})| \leq 1$ for any $\boldsymbol{x} \in \mathcal{X}$ and $i \in [N]$ without sharing these local functions:

$$\min_{\boldsymbol{x} \in \mathcal{X}} F(\boldsymbol{x}) \triangleq \frac{1}{N} \sum_{i \in [N]} f_i(\boldsymbol{x}). \tag{1}$$

A central server is typically introduced to periodically aggregate the updated inputs sent from the distributed clients after their several iterations of local optimization. Of note, in this federated ZOO setting, the gradients of the local functions are either not accessible or too computationally expensive to obtain. Consequently, the gradients can not be directly employed for optimization, which is our main difference from the standard federated *first-order optimization* (FOO) setting (Konečný et al., 2015; Wang et al., 2021; Reddi et al., 2021). Instead, given an input $\boldsymbol{x} \in \mathcal{X}$, agent $i$ is only allowed to observe a noisy output $y_i(\boldsymbol{x}) \triangleq f_i(\boldsymbol{x}) + \zeta$ of the local function $f_i$, in which $\zeta \sim \mathcal{N}(0, \sigma^2)$. Moreover, we focus on federated ZOO with heterogeneous clients, i.e., the local functions $\{f_i\}_{i=1}^N$ differ from the global function $F$. Besides, we adopt a common assumption on $\{f_i\}_{i=1}^N$: We assume that every local function $f_i$ is sampled from a *Gaussian process* (GP), i.e., $f_i \sim \mathcal{GP}(\mu(\cdot), k(\cdot, \cdot))$ (Shu et al., 2023), in which $k$ is a shift-invariant kernel and is assumed to have $\|\partial_{\boldsymbol{z}} \partial_{\boldsymbol{z}'} k(\boldsymbol{z}, \boldsymbol{z}')|_{\boldsymbol{z}=\boldsymbol{z}'=\boldsymbol{x}}\| \leq \kappa, \|\partial_{\boldsymbol{z}} k(\boldsymbol{z}, \boldsymbol{x}')|_{\boldsymbol{z}=\boldsymbol{x}}\| \leq L \; (\forall \boldsymbol{x}, \boldsymbol{x}' \in \mathcal{X})$ for some $\kappa > 0$ and $L > 0$. This encompasses commonly used kernels such as the squared exponential kernel (Rasmussen and Williams, 2006). Unless specified otherwise, we use $\|\cdot\|$ to denote the norm $\|\cdot\|_2$ and $[Z]$ to denote the set $\{1, \cdots, Z\}$. We will use $i \in [N]$ to denote the formulas related to client $i$ in this paper.

## 3 FRAMEWORK AND CHALLENGES FOR FEDERATED ZOO

Here we firstly summarize the framework to solve the federated ZOO problem (Sec. 3.1), and then identify the challenges which existing algorithms following this framework fail to address (Sec. 3.2).

### 3.1 OPTIMIZATION FRAMEWORK

To solve (1), a general optimization framework is to estimate the gradients of $\{f_i\}_{i=1}^N$ using only function queries and then employ the standard federated FOO algorithms for the optimization, as in Algo. 1. Specifically, in round $r$, every client performs $T$ iterations of local gradient decent updates in parallel (line 2-5 of Algo. 1), in which $\widehat{\boldsymbol{g}}_{r,t-1}^{(i)} \in \mathbb{R}^d$ denotes the estimated gradient by client $i$ for the local update in iteration $t$ of round $r$. After that, each client sends its locally updated input $\boldsymbol{x}_{r,T}^{(i)}$ to server (line 6 of Algo. 1). After receiving the updated inputs from all clients (i.e., $\{\boldsymbol{x}_{r,T}^{(i)}\}_{i=1}^N$), the server aggregates them (e.g., via arithmetic average) to produce a globally updated input $\boldsymbol{x}_r$, and then sends it back to the clients for the optimization in the next round (line 7-8 of Algo. 1).

The aforementioned $\widehat{\boldsymbol{g}}_{r,t-1}^{(i)}$ used in the literature can be summarized into the following general form:
$$\widehat{\boldsymbol{g}}_{r,t-1}^{(i)} \triangleq \boldsymbol{g}_{r,t-1}^{(i)} + \gamma_{r,t-1}^{(i)}\left(\boldsymbol{g}_{r-1}(\boldsymbol{x}') - \boldsymbol{g}_{r-1}^{(i)}(\boldsymbol{x}'')\right) \tag{2}$$
where $\boldsymbol{g}_{r,t-1}^{(i)} \in \mathbb{R}^d$ is an estimate of $\nabla f_i(\boldsymbol{x}_{r,t-1}^{(i)})$ and is usually obtained using the *finite difference* (FD) methods (refer to Sec. 3.2). In addition, the *gradient correction vector* $\boldsymbol{g}_{r-1}(\boldsymbol{x}') - \boldsymbol{g}_{r-1}^{(i)}(\boldsymbol{x}'') \in \mathbb{R}^d$ is usually obtained from the previous round $r-1$. This aims to make the resulting $\widehat{\boldsymbol{g}}_{r,t-1}^{(i)}$ better aligned with $\nabla F(\boldsymbol{x}_{r,t-1}^{(i)})$, such that the local update on each client (i.e., line 5 of Algo. 1) can better approximate the intended global update along the direction of $\nabla F(\boldsymbol{x}_{r,t-1}^{(i)})$. It is especially important in the presence of client heterogeneity, i.e., $\{\nabla f_i\}_{i=1}^N$ differ from $\nabla F$. Intuitively, to accomplish this alignment, $\boldsymbol{g}_{r-1}(\boldsymbol{x}')$ and $\boldsymbol{g}_{r-1}^{(i)}(\boldsymbol{x}'')$ should be good estimates of $\nabla F(\boldsymbol{x}_{r,t-1}^{(i)})$ and $\nabla f_i(\boldsymbol{x}_{r,t-1}^{(i)})$, respectively, which we theoretically justify in Sec. 3.2. Of note, the form of $\boldsymbol{g}_{r-1}(\boldsymbol{x}') - \boldsymbol{g}_{r-1}^{(i)}(\boldsymbol{x}'')$ for gradient correction usually aims to ensure that the estimation biases from $\boldsymbol{g}_{r-1}(\boldsymbol{x}')$ and $\boldsymbol{g}_{r-1}^{(i)}(\boldsymbol{x}'')$ could cancel out (Johnson and Zhang, 2013). Finally, $\gamma_{r,t-1}^{(i)} \in [0,1]$ denotes the *gradient correction length*, which can be adjusted to trade off the utilization of the gradient correction vector (Sec. 3.2).

Remarkably, (2) subsumes the forms of gradient updates employed in many existing federated ZOO algorithms, and hence Algo. 1 can reduce to the corresponding optimization algorithms (more details in Appx. D). E.g., when $\gamma_{r,t-1}^{(i)} = 0$ and $\boldsymbol{g}_{r,t-1}^{(i)}$ is obtained using FD, Algo. 1 becomes the FedZO algorithm (Fang et al., 2022); when $\gamma_{r,t-1}^{(i)}=1$, $\boldsymbol{g}_{r-1}(\boldsymbol{x}')=\frac{1}{NT}\sum_{i,t=1}^{N,T}\boldsymbol{g}_{r-1,t-1}^{(i)}$, and $\boldsymbol{g}_{r-1}^{(i)}(\boldsymbol{x}'') = \frac{1}{T}\sum_{t=1}^{T}\boldsymbol{g}_{r-1,t-1}^{(i)}$, (2) reduces to the gradient update in (Karimireddy et al., 2020a) and hence Algo. 1 becomes the SCAFFOLD (Type II) algorithm in the federated ZOO setting; let the gradient correction vector $\boldsymbol{g}_{r-1}(\boldsymbol{x}') - \boldsymbol{g}_{r-1}^{(i)}(\boldsymbol{x}'')$ in (2) be $\boldsymbol{x}_{r,t-1}^{(i)} - \boldsymbol{x}_r$, Algo. 1 is then equivalent to FedProx (Li et al., 2020a) in the federated ZOO setting.

### 3.2 EXISTING CHALLENGES

Existing federated ZOO algorithms aiming to solve the problem in Sec. 2 typically fail to address the challenges of query efficiency and communication round efficiency, which we discuss in detail below.

**Challenge of Query Efficiency.** Similar to standard ZOO algorithms (Nesterov and Spokoiny, 2017; Cheng et al., 2021), existing federated ZOO algorithms (e.g., (Fang et al., 2022)) also commonly apply the FD methods (Berahas et al., 2022) for gradient estimation. Specifically, given a parameter $\lambda > 0$ and directions $\{\boldsymbol{u}_q\}_{q=1}^Q$, the gradient of the function $f_i$ on client $i$ at $\boldsymbol{x}$ can be estimated as
$$\nabla f_i(\boldsymbol{x}) \approx \boldsymbol{\Delta}^{(i)}(\boldsymbol{x}) \triangleq \frac{1}{Q}\sum_{q\in[Q]} \frac{y_i(\boldsymbol{x}+\lambda\boldsymbol{u}_q) - y_i(\boldsymbol{x})}{\lambda}\boldsymbol{u}_q. \tag{3}$$

That is, for existing federated ZOO algorithms, $\boldsymbol{g}_{r,t-1}^{(i)} = \boldsymbol{\Delta}^{(i)}(\boldsymbol{x}_{r,t-1}^{(i)})$ in (2). As implied in (3), $Q$ additional function queries are required for the gradient estimation at every local updated input $\boldsymbol{x}_{r,t-1}^{(i)}$. This therefore results in $NTQ\times$ more function queries than the standard federated FOO algorithms (Li et al., 2020a; Karimireddy et al., 2020a) in every communication round, which is unsatisfying in practice especially when $\{f_i\}_{i=1}^N$ are prohibitively costly to evaluate. So, tackling the challenge of query efficiency in federated ZOO requires designing query-efficient gradient estimators.

| **Algorithm 1:** The General Optimization Framework for Federated ZOO | **Algorithm 2:** FZooS |
|---|---|
| **Input:** Initial $\boldsymbol{x}_0$, rounds $R$, learning rate $\eta$, iterations $T$ for each round, number of clients $N$ | **Input:** Input of Algo. 1, length $\gamma$, $M$ features |
| 1 **for** *each round* $r \in [R]$ **do** | 1 **for** *each round* $r \in [R]$ **do** |
|    // Client-Side Update |    // Client-Side Update |
| 2   **for** *each client* $i \in [N]$ *in parallel* **do** | 2   **for** *each client* $i \in [N]$ *in parallel* **do** |
| 3     $\boldsymbol{x}_{r,0}^{(i)} \leftarrow \boldsymbol{x}_{r-1}$ | 3     $\boldsymbol{x}_{r,0}^{(i)} \leftarrow \boldsymbol{x}_{r-1}$, $\nabla\widehat{\mu}_{r-1}$ based on $\boldsymbol{w}_{r-1}$ |
| 4     **for** *each iteration* $t \in [T]$ **do** | 4     **for** *each iteration* $t \in [T]$ **do** |
| 5       $\boldsymbol{x}_{r,t}^{(i)} \leftarrow \boldsymbol{x}_{r,t-1}^{(i)} - \eta\,\widehat{\boldsymbol{g}}_{r,t-1}^{(i)}$ | 5       $\nabla\mu_{r,t-1}^{(i)}$ conditioned on $\mathcal{D}_{r,t-1}^{(i)}$ |
| 6     Send $\boldsymbol{x}_{r,T}^{(i)}$ to receive $\boldsymbol{x}_r$ back | 6       $\boldsymbol{x}_{r,t}^{(i)} \leftarrow \boldsymbol{x}_{r,t-1}^{(i)} - \eta\,\widehat{\boldsymbol{g}}_{r,t-1}^{(i)}$ with (8) |
|    // Server-Side Update | 7     Send $\boldsymbol{x}_{r,T}^{(i)}$ to receive $\boldsymbol{x}_r$, query around $\boldsymbol{x}_r$ |
| 7   $\boldsymbol{x}_r \leftarrow \frac{1}{N}\sum_{i\in[N]}\boldsymbol{x}_{r,T}^{(i)}$ | 8     Approx. $\nabla\mu_{r,T}^{(i)}$ via RFF to get $\boldsymbol{w}_{r,T}^{(i)}$ |
| 8   Send $\boldsymbol{x}_r$ back to each client | 9     Send $\boldsymbol{w}_{r,T}^{(i)}$ to receive $\boldsymbol{w}_r$ back |
| |    // Server-Side Update |
| | 10   $\boldsymbol{x}_r \leftarrow \frac{1}{N}\sum_{i\in[N]}\boldsymbol{x}_{r,T}^{(i)}$, $\boldsymbol{w}_r \leftarrow \frac{1}{N}\sum_{i\in[N]}\boldsymbol{w}_{r,T}^{(i)}$ |
| | 11   Send $\boldsymbol{x}_r$ back first and then $\boldsymbol{w}_r$ to each client |

**Challenge of Communication Round Efficiency.** When $\widehat{\boldsymbol{g}}_{r,t-1}^{(i)} = \nabla F(\boldsymbol{x}_{r,t-1}^{(i)})$ in (2), Algo. 1 is then able to attain the convergence of centralized FOO algorithms, which is known to be better than the one in the federated setting (Karimireddy et al., 2020a). Therefore, intuitively, the convergence or the communication round efficiency (i.e., the number of communication rounds $R$ required to achieve an $\epsilon$ convergence error) of Algo. 1 depends on the disparity between (2) and $\nabla F(\boldsymbol{x}_{r,t-1}^{(i)})$. Define the gradient disparity $\Xi_{r,t}^{(i)} \triangleq \|\widehat{\boldsymbol{g}}_{r,t-1}^{(i)} - \nabla F(\boldsymbol{x}_{r,t-1}^{(i)})\|^2$, we propose the following Prop. 1 (proof in Appx. C.1) to show the condition for the best-performing (2) and thus to justify the challenge in communication round efficiency that existing federated ZOO algorithms typically fail to address well.

**Proposition 1.** *Let* $\boldsymbol{g}_{r-1}^{(i)}(\boldsymbol{x}'') \neq \boldsymbol{g}_{r-1}(\boldsymbol{x}')$, *the minimum of* $\Xi_{r,t}^{(i)}$ *w.r.t* $\gamma_{r,t-1}^{(i)}$ *is achieved when*

$$\gamma_{r,t-1}^{(i)} = \gamma_{r,t-1}^{(i)*} \triangleq \left(\nabla F(\boldsymbol{x}_{r,t-1}^{(i)}) - \boldsymbol{g}_{r-1}^{(i)}\right)^{\top}\left(\boldsymbol{g}_{r-1}(\boldsymbol{x}') - \boldsymbol{g}_{r-1}^{(i)}(\boldsymbol{x}'')\right)\left\|\boldsymbol{g}_{r-1}(\boldsymbol{x}') - \boldsymbol{g}_{r-1}^{(i)}(\boldsymbol{x}'')\right\|^{-2}.$$

*When* $\gamma_{r,t-1}^{(i)*} = 1$, $\Xi_{r,t}^{(i)} = 0$ *iff we have* $\boldsymbol{g}_{r-1}(\boldsymbol{x}') - \boldsymbol{g}_{r-1}^{(i)}(\boldsymbol{x}'') = \nabla F(\boldsymbol{x}_{r,t-1}^{(i)}) - \boldsymbol{g}_{r-1}^{(i)}$.

Prop. 1 shows that to achieve a small gradient disparity, $\gamma_{r,t-1}^{(i)}$ should be adaptive w.r.t. the alignment between the *gradient correction vector* $\boldsymbol{g}_{r-1}(\boldsymbol{x}') - \boldsymbol{g}_{r-1}^{(i)}(\boldsymbol{x}'')$ and the *drift* $\nabla F(\boldsymbol{x}_{r,t-1}^{(i)}) - \boldsymbol{g}_{r,t-1}^{(i)}$. We have shown (Appx. C.1) that a better alignment between the gradient correction vector and the drift leads to a smaller gradient disparity, Prop. 1 further shows that a zero gradient disparity (i.e., $\Xi_{r,t}^{(i)} = 0$ for any $r \in [R], t \in [T]$) can be reached when these two are perfectly aligned. To achieve such an alignment, i.e., to make $\boldsymbol{g}_{r-1}(\boldsymbol{x}') = \nabla F(\boldsymbol{x}_{r,t-1}^{(i)})$ and $\boldsymbol{g}_{r-1}^{(i)}(\boldsymbol{x}'') = \boldsymbol{g}_{r,t-1}^{(i)}$ hold more likely, it requires not only *(a)* accurate gradient surrogates $\boldsymbol{g}_{r-1}$ and $\boldsymbol{g}_{r-1}^{(i)}$ to accurately represent $\nabla F$ and $\nabla f_i$, respectively, but also *(b)* adaptive $\boldsymbol{x}', \boldsymbol{x}''$ to avoid the discrepancy between $\boldsymbol{x}_{r,t-1}^{(i)}$ and $\boldsymbol{x}', \boldsymbol{x}''$.

Consequently, resolving the challenge of communication round efficiency in federated ZOO mainly requires **(A)** *accurate* local and global surrogates (i.e., $\boldsymbol{g}_{r-1}^{(i)}$ and $\boldsymbol{g}_{r-1}$) for the gradient correction in (2), and **(B)** *adaptive* gradient correction in (2) with both adaptive $\boldsymbol{x}', \boldsymbol{x}''$ and adaptive $\gamma_{r,t-1}^{(i)}$. However, existing federated ZOO algorithms usually fail to address them well: Firstly, these algorithms rely on the FD methods for gradient estimation, which usually lead to poor estimation quality and consequently inaccurate gradient correction vectors in (2) when the query budget is very limited. Secondly, although $\boldsymbol{x}_{r,t-1}^{(i)}$ changes during local updates, existing algorithms typically rely on $\boldsymbol{g}_{r-1}, \boldsymbol{g}_{r-1}^{(i)}$ evaluated at a fixed input $\boldsymbol{x}_{r-1} = \boldsymbol{x}' = \boldsymbol{x}''$ to estimate $\nabla F$ or $\nabla f_i$ (e.g., (Li et al., 2020a; Karimireddy et al., 2020a)), leading to large discrepancies between $\boldsymbol{x}_{r,t-1}^{(i)}$ and $\boldsymbol{x}', \boldsymbol{x}''$. Thirdly, existing algorithms use a fixed gradient correction length (e.g., $\gamma_{r,t-1}^{(i)} = 0$ in (Fang et al., 2022) and $\gamma_{r,t-1}^{(i)} = 1$ in (Karimireddy et al., 2020a)), which is likely to result in misspecified gradient correction length.

## 4 FZOOS ALGORITHM

To address the aforementioned challenges, we propose our *federated zeroth-order optimization using trajectory-informed surrogate gradients* (FZooS) algorithm in Algo. 2, which improves the query and communication round efficiency of existing algorithms thanks to our two major contributions, correspondingly. Firstly, we introduce the *trajectory-informed derived Gaussian Process* in (Shu et al.,

2023) as local gradient surrogates for query-efficient gradient estimations (Sec. 4.1). Secondly, we use *random Fourier features* (RFF) approximation (Rahimi and Recht, 2007) to attain a transferable global gradient surrogate that can accurately estimate the gradient of the global function (Sec. 4.2.1); based on these surrogates, we then develop the technique of *adaptive gradient correction* with both adaptive gradient correction vector and length for communication round-efficient federated ZOO by mitigating the disparity between our local updates and the intended global updates (Sec. 4.2.2).

## 4.1 TRAJECTORY-INFORMED GRADIENT ESTIMATION FOR QUERY EFFICIENCY

Of note, we assumed that $f_i \sim \mathcal{GP}(\mu(\cdot), k(\cdot, \cdot)), \forall i \in [N]$ (Sec. 2). Then, in iteration $t$ of communication round $r$ (Algo. 2), conditioned on the optimization trajectory $\mathcal{D}_{r,t-1}^{(i)} \triangleq \{(\boldsymbol{x}_\tau^{(i)}, y_\tau^{(i)})\}_{\tau=1}^{T(r-1)+t-1}$ of client $i$,[2] $\nabla f_i$ follows a *derived posterior Gaussian Process* (Shu et al., 2023):

$$\nabla f_i \sim \mathcal{GP}\left(\nabla \mu_{r,t-1}^{(i)}(\cdot), \partial\left(\sigma_{r,t-1}^{(i)}\right)^2(\cdot, \cdot)\right) \tag{4}$$

where the mean function $\nabla \mu_{r,t-1}^{(i)}(\boldsymbol{x})$ and the covariance function $\partial(\sigma_{r,t-1}^{(i)})^2(\boldsymbol{x}, \boldsymbol{x}')$ are defined as

$$\nabla \mu_{r,t-1}^{(i)}(\boldsymbol{x}) \triangleq \partial_{\boldsymbol{x}} \boldsymbol{k}_{r,t-1}^{(i)}(\boldsymbol{x})^\top \left(\mathbf{K}_{r,t-1}^{(i)} + \sigma^2 \mathbf{I}\right)^{-1} \boldsymbol{y}_{r,t-1}^{(i)},$$

$$\partial\left(\sigma_{r,t-1}^{(i)}\right)^2(\boldsymbol{x}, \boldsymbol{x}') \triangleq \partial_{\boldsymbol{x}} \partial_{\boldsymbol{x}'} k(\boldsymbol{x}, \boldsymbol{x}') - \partial_{\boldsymbol{x}} \boldsymbol{k}_{r,t-1}^{(i)}(\boldsymbol{x})^\top \left(\mathbf{K}_{r,t-1}^{(i)} + \sigma^2 \mathbf{I}\right)^{-1} \partial_{\boldsymbol{x}'} \boldsymbol{k}_{r,t-1}^{(i)}(\boldsymbol{x}').$$

$$\tag{5}$$

Both $\boldsymbol{k}_{r,t-1}^{(i)}(\boldsymbol{x})^\top \triangleq [k(\boldsymbol{x}, \boldsymbol{x}_\tau^{(i)})]_{\tau=1}^{T(r-1)+t-1}$ and $(\boldsymbol{y}_{r,t-1}^{(i)})^\top \triangleq [y_\tau^{(i)}]_{\tau=1}^{T(r-1)+t-1}$ are $[T(r-1)+t-1]$-dimensional row vectors, and $\mathbf{K}_{r,t-1}^{(i)} \triangleq [k(\boldsymbol{x}_\tau^{(i)}, \boldsymbol{x}_{\tau'}^{(i)})]_{\tau,\tau'=1}^{T(r-1)+t-1}$ is a $[T(r-1)+t-1] \times [T(r-1)+t-1]$-dimensional matrix.

We propose to use the posterior mean $\nabla \mu_{r,t-1}^{(i)}(\boldsymbol{x})$ (5) as the local gradient surrogate for client $i$ since it is a prediction of the gradient $\nabla f_i(\boldsymbol{x})$, and $\partial(\sigma_{r,t-1}^{(i)})^2(\boldsymbol{x}) \triangleq \partial(\sigma_{r,t-1}^{(i)})^2(\boldsymbol{x}, \boldsymbol{x})$ provides a principled uncertainty measure for this gradient surrogate (Shu et al., 2023). Of note, our gradient surrogate only requires the optimization trajectory (i.e., the history of function queries $\mathcal{D}_{r,t-1}^{(i)}$ till iteration $t-1$ of round $r$) and thus *eliminates the need for additional queries* required by the FD methods adopted by existing federated ZOO (Sec. 3.2). This therefore leads to more query-efficient gradient estimations in federated ZOO. Moreover, the aforementioned uncertainty measure can theoretically guarantee the quality of our gradient estimation, and provide theoretical support for our technique of using active queries to further improve the local gradient estimations (Sec. 5.1).

## 4.2 HIGH-QUALITY GRADIENT CORRECTION FOR COMMUNICATION ROUND EFFICIENCY

### 4.2.1 TRANSFERABLE GLOBAL GRADIENT SURROGATE

Of note, our local gradient surrogates from Sec. 4.1 can produce not only query-efficient but also accurate gradient estimations (Shu et al., 2023). So, these local surrogates can be used to construct an accurate global gradient surrogate, which then satisfies requirement **(A)** for communication round-efficient federated ZOO from Sec. 3.2: accurate local and global gradient surrogates. Unfortunately, due to the non-parametric nature of Gaussian processes, (4) cannot be transferred to the server without sending the raw observations. To this end, we introduce the idea of *random Fourier features* (RFF) approximation from (Rahimi and Recht, 2007) to approximate the mean of (4) and then transfer this approximated mean to server for the construction of high-quality global gradient surrogate.

We firstly approximate the mean of (4) on each client $i \in [N]$ to ease its transfer between the clients and the server. Since $k(\cdot, \cdot)$ is assumed to be shift-invariant, it can be approximated by a finite number of random features (Rahimi and Recht, 2007). That is, we have that $k(\boldsymbol{x}, \boldsymbol{x}') \approx \phi(\boldsymbol{x})^\top \phi(\boldsymbol{x}')$ where pre-defined function $\phi : \mathbb{R}^d \mapsto \mathbb{R}^M$ produces $M$ random features and its parameters are shared across all clients and the server (Appx. B). By incorporating this approximation into (5), the local gradient surrogates on each client $i$ at the end of every round $r$ (i.e., $\nabla \mu_{r,T}^{(i)}(\boldsymbol{x})$) can then be approximated as

$$\nabla \widehat{\mu}_{r,T}^{(i)}(\boldsymbol{x}) \triangleq \nabla \phi(\boldsymbol{x})^\top \boldsymbol{\Phi}_{r,T}^{(i)} \left(\widehat{\mathbf{K}}_{r,T}^{(i)} + \sigma^2 \mathbf{I}\right)^{-1} \boldsymbol{y}_{r,T}^{(i)} \tag{6}$$

where $\nabla \phi(\boldsymbol{x})$ is an $M \times d$-dimensional matrix, $\boldsymbol{\Phi}_{r,T}^{(i)} \triangleq [\phi(\boldsymbol{x}_\tau^{(i)})]_{\tau=1}^{rT}$ is an $M \times rT$-dimensional matrix, and $\widehat{\mathbf{K}}_{r,T}^{(i)} \triangleq [\phi(\boldsymbol{x}_\tau^{(i)})^\top \phi(\boldsymbol{x}_{\tau'}^{(i)})]_{\tau,\tau'=1}^{rT}$ is an $rT \times rT$-dimensional matrix. Define an $M$-dimensional column vector $\boldsymbol{w}_{r,T}^{(i)} \triangleq \boldsymbol{\Phi}_{r,T}^{(i)}(\widehat{\mathbf{K}}_{r,T}^{(i)} + \sigma^2 \mathbf{I})^{-1} \boldsymbol{y}_{r,T}^{(i)}$, (6) can be rewritten as $\nabla \widehat{\mu}_{r,t-1}^{(i)}(\boldsymbol{x}) =$

---

[2]We slightly abuse notation and use $(\boldsymbol{x}_\tau^{(i)}, y_\tau^{(i)})$ to denote a historical query till iteration $t-1$ of round $r$.

$\nabla\phi(\boldsymbol{x})^{\top}\boldsymbol{w}_{r,T}^{(i)}$ (line 8 of Algo. 2). So, each client only needs to calculate and send the $M$-dimensional vector $\boldsymbol{w}_{r,T}^{(i)}$ to the server for constructing the global gradient surrogate (line 9 of Algo. 2).

After receiving $\{\boldsymbol{w}_{r,T}^{(i)}\}_{i=1}^{N}$ from all clients, the server can construct the global gradient surrogate at the end of every round $r$ by averaging the local gradient surrogates (6) from all clients, i.e.,

$$\nabla\widehat{\mu}_{r}(\boldsymbol{x}) \triangleq \frac{1}{N}\sum_{i\in[N]}\widehat{\mu}_{r,T}^{(i)}(\boldsymbol{x}) = \nabla\phi(\boldsymbol{x})^{\top}\left(\frac{1}{N}\sum_{i\in[N]}\boldsymbol{w}_{r,T}^{(i)}\right). \tag{7}$$

To transfer this global gradient surrogate to clients, we only need to send the $M$-dimensional vector $\boldsymbol{w}_{r} \triangleq \frac{1}{N}\sum_{i=1}^{N}\boldsymbol{w}_{r,T}^{(i)}$ back (lines 10-11 of Algo. 2). Importantly, after receiving $\boldsymbol{w}_{r}$ from the server, each client can calculate the global gradient surrogate *at any input in the domain*. Although this global gradient surrogate incurs an additional transmission of $M$-dimensional vectors compared with existing federated ZOO algorithms (Algo. 1), it enjoys the advantage of achieving an improved gradient correction with theoretical guarantees (Sec. 5.1), which is known to be essential for addressing federated ZOO with heterogeneous clients (Sec. 3.2) and is thus able to outweigh its drawback of increased transmission burden in practice. To further improve the quality of this surrogate, we can actively query in the neighbourhood of the updated input $\boldsymbol{x}_{r}$ on every client (line 7 of Algo. 2) as supported in Sec. 5.1. This incurs an additional server-clients transmission because the transmission of the gradient surrogates via $\boldsymbol{w}_{r,T}^{(i)}$ needs to happen after the active queries (i.e., after the gradient surrogates are improved), which is consistent with SCAFFOLD (Type I) (Karimireddy et al., 2020a).

### 4.2.2 ADAPTIVE GRADIENT CORRECTION

By exploiting our aforementioned high-quality local and global gradient surrogates, we then develop the technique of adaptive gradient correction to meet requirement **(B)** for communication round-efficient federated ZOO from Sec. 3.2. Specifically, thanks to the ability of our gradient surrogates to *estimate the gradient at any input in the domain*, we can let $\boldsymbol{x}' = \boldsymbol{x}'' = \boldsymbol{x}_{r,t-1}^{(i)}$ in (2) to realize a more accurate gradient correction vector during optimization. Moreover, we propose to employ an adaptive gradient correction length $\gamma_{r,t-1}$ (shared across all clients) to better trade off the utilization of our gradient correction vector during optimization.

That is, for every iteration $t$ of round $r$, we propose to use the following $\widehat{\boldsymbol{g}}_{r,t-1}^{(i)}$ on each client $i\in[N]$:

$$\widehat{\boldsymbol{g}}_{r,t-1}^{(i)} = \nabla\mu_{r,t-1}^{(i)}(\boldsymbol{x}_{r,t-1}^{(i)}) + \gamma_{r,t-1}\left(\nabla\widehat{\mu}_{r-1}(\boldsymbol{x}_{r,t-1}^{(i)}) - \nabla\widehat{\mu}_{r-1,T}^{(i)}(\boldsymbol{x}_{r,t-1}^{(i)})\right), \tag{8}$$

(i.e., line 6 of Algo. 2) where $\nabla\widehat{\mu}_{r-1,T}^{(i)}$ is the local gradient surrogate of client $i$ with RFF approximation at the end of round $r-1$ from (6), $\nabla\widehat{\mu}_{r-1}$ is our global gradient surrogate from (7), and $\gamma_{r,t-1}$ is a theoretically inspired adaptive gradient correction length which we will discuss in Sec. 5.1. Of note, the advantage of this adaptive gradient correction can be theoretically justified (Sec. 5.1).

## 5 THEORETICAL ANALYSIS

In this section, we present our theoretical analysis on the gradient disparity of our local gradient update (8) in Sec. 5.1 and the convergence of our FZooS (Algo. 2) in Sec. 5.2.

### 5.1 GRADIENT DISPARITY ANALYSIS

We assume that $\frac{1}{N}\sum_{i=1}^{N}\|\nabla f_{i}(\boldsymbol{x}) - \nabla F(\boldsymbol{x})\|^{2} \leq G$ for any $\boldsymbol{x}\in\mathcal{X}$, which is a common assumption in the analysis of federated optimization (Reddi et al., 2021). Here a larger $G$ indicates a larger degree of client heterogeneity. By making use of the uncertainty measure from (5), we derive an upper bound on the gradient disparity of our (8) in Thm. 1 below (proof in Appx. C.2).

**Theorem 1.** *Define* $\rho_{i} \triangleq \max_{\boldsymbol{x}\in\mathcal{X},r\geq1,t\geq1}\|\partial(\sigma_{r,t}^{(i)})^{2}(\boldsymbol{x})\|/\|\partial\left(\sigma_{r,t-1}^{(i)}\right)^{2}(\boldsymbol{x})\|$ *and* $\rho \triangleq \frac{1}{N}\sum_{i=1}^{N}\rho_{i}$, $\rho,\rho_{i}\in[\frac{1}{1+1/\sigma^{2}},1]$. *Given constant* $\omega>0$ *and* $\epsilon = \mathcal{O}(\frac{1}{M})$, *the following holds with constant probability*

$$\frac{1}{N}\sum_{i\in[N]}\Xi_{r,t}^{(i)} \leq \underbrace{4\omega\kappa\rho^{(r-1)T+t-1}}_{\text{①}} + \gamma_{r,t-1}^{2}\underbrace{(8\omega\kappa\rho^{(r-1)T} + 8N\epsilon)}_{\text{②}} + (1-\gamma_{r,t-1})^{2}\underbrace{4G}_{\text{③}}.$$

**Corollary 1.** *Thm. 1 implies a better-performing choice of* $\gamma_{r,t-1}$, *i.e.,* $\gamma_{r,t-1} = \frac{G}{G+2\omega\kappa\rho^{(r-1)T}+2N\epsilon}$.

In the upper bound of Thm. 1, term ① represents the error of estimating $\{\nabla f_{i}(\cdot)\}_{i=1}^{N}$ using our local gradient surrogates in Sec. 4.1, and term ② characterizes the disparity between our gradient

correction vector in (8) and its corresponding ground truth $\{\nabla F(\cdot) - \nabla f_i(\cdot)\}_{i=1}^N$. The $\epsilon$ within term ② denotes the RFF approximation error for our global gradient surrogate in Sec. 4.2.1 and $\epsilon$ decreases with a larger number $M$ of random features. Term ③ results from the client heterogeneity in federated ZOO. Compared with the gradient disparity of existing algorithms (provided in Appx. D), Thm. 1 shows that our (8) enjoys a number of major advantages: **(a)** Our (8) is more query-efficient since it does not require any additional function query for gradient estimation, in contrast to existing algorithms which incur $\mathcal{O}(NQ)$ additional function queries in every iteration. **(b)** The estimation error in our (8) (i.e., terms ① and ②) can be exponentially decreasing when $\rho < 1$ and $\epsilon$ is small, whereas other existing algorithms only achieve a reduction rate of $\mathcal{O}(1/Q)$, which implies that our gradient estimation is significantly more accurate. Of note, $\rho_i < 1$ is likely to be satisfied as justified in (Shu et al., 2023) and more importantly, $\rho < 1$ is even easier to be realized as it only needs one of the clients to satisfy $\rho_i < 1$. **(c)** Our (8) mitigates the disparity caused by the fixed gradient correction vector adopted by existing works, i.e., in contrast to FedProx and SCAFFOLD, our Thm. 1 does not contain an additional disparity term of $\sum_{i=1}^N \|\boldsymbol{x}_{r,t-1}^{(i)} - \boldsymbol{x}_{r-1}\|^2$. **(d)** Our (8) can trade off between the impacts of our gradient correction vector and client heterogeneity, and can consequently urther improve the gradient estimation when $\gamma_{r,t-1}$ is chosen intelligently while accounting for this trade-off. Specifically, the upper bound in Thm. 1 has characterized such a trade-off: When the estimation error of our gradient correction vector (i.e., term ②) is relatively small compared with the client heterogeneity (i.e., term ③), a large $\gamma_{t-1}$ is preferred to reduce the impact of client heterogeneity and hence to achieve a small gradient disparity. Furthermore, this also implies a theoretically better choice of $\gamma_{r,t-1}$ in our Cor. 1 (refer to Appx. C.3 for a more practical choice of $\gamma_{r,t-1}$).

In addition to the theoretical insights above, Thm. 1 also offers valuable insights to enhance the practical efficacy of our (8). Firstly, during local updates, we can actively query more function values on each client to further decrease the uncertainty (i.e., $\|\partial(\sigma_{r,t}^{(i)})^2(\boldsymbol{x})\|$) of our local gradient surrogates, which improves our (8) by decreasing term ① in Thm. 1 with a larger exponent. Secondly, after receiving $\boldsymbol{x}_r$ from the server (i.e., at the end of every round $r$ of our Algo. 2), we can actively query in the neighborhood of $\boldsymbol{x}_r$ on every client, in order to decrease term ② in Thm. 1 using a larger exponent and thus to improve the quality of gradient correction in our (8). Thirdly, we can use a large number $M$ of random features to achieve a small RFF approximation error $\epsilon$ in term ② of Thm. 1. Fourthly, we can choose an adaptive gradient correction length $\gamma_{r,t-1}$ (e.g., the $\gamma_{r,t-1}$ in Cor. 1) to better trade off the impacts of the gradient correction and client heterogeneity.

## 5.2 CONVERGENCE ANALYSIS

We prove the convergence of our FZooS (measured by the number of communication rounds to achieve $\epsilon$ convergence error) under different assumptions, in addition to assuming that $F$ is $\beta$-smooth.

**Theorem 2.** *Define $D_0 \triangleq \|\boldsymbol{x}_0 - \boldsymbol{x}^*\|^2$ and $D_1 \triangleq F(\boldsymbol{x}_0) - F(\boldsymbol{x}^*)$, to achieve an $\epsilon$ convergence error for our FZooS (Algo. 2) with a constant probability when $\rho < 1$, the number $M$ of random features and the number $R$ of communication rounds need to satisfy the following,*

(i) *If $F$ is strongly convex and $\eta \leq \frac{1}{10\beta T}$, $M = \mathcal{O}\left(\frac{NG}{\epsilon^2}\right)$ and $R = \mathcal{O}\left(\frac{1}{\eta T}\ln\frac{D_0}{\epsilon} + \ln\frac{\sqrt{G}}{\epsilon}\right)$.*

(ii) *If $F$ is convex and $\eta \leq \frac{1}{10\beta T}$, $M = \mathcal{O}\left(\frac{NG}{\epsilon^2} + \frac{d^2 NG}{\epsilon^4}\right)$ and $R = \mathcal{O}\left(\frac{D_0}{\eta T\epsilon} + \frac{\sqrt{G}+\sqrt[4]{d^2 G}}{\epsilon}\right)$.*

(iii) *If $F$ is non-convex and $\eta \leq \frac{7}{100\beta T}$, $M = \mathcal{O}\left(\frac{NG}{\epsilon^2}\right)$ and $R = \mathcal{O}\left(\frac{D_1}{\eta T\epsilon} + \frac{\sqrt{G}}{\epsilon}\right)$.*

The proof is in Appx. C.5.[3] Thm. 2 suggests that the learning rate $\eta$ in FZooS should be proportionally reduced w.r.t. the number $T$ of local updates, which is in fact consistent with the results in federated FOO (Karimireddy et al., 2020a). Thm. 2 also shows that when client heterogeneity (i.e., measured by $G$) increases, both the number $M$ of random features and the number $R$ of communication rounds in our FZooS should be increased in order to achieve the same convergence error, which is also empirically verified in our Sec. 6 and Appx. F. Moreover, Thm. 2 has revealed that given a constant learning rate $\eta$ that satisfies the conditions in Thm. 2 under various $T$, a larger $T$ usually improves the communication round efficiency (i.e., $R$) of our FZooS (see Appx. F). More importantly, compared with the convergence of other existing algorithms (provided in Appx. D), FZooS enjoys an improved

---

[3]The poor convergence of our FZooS under convex $F$ (vs. the one under non-convex $F$) results from the drawback of the commonly applied proof technique for convex $F$ rather than the algorithm itself. This has been widely recognized in the literature (Harvey et al., 2019; Liu et al., 2023).

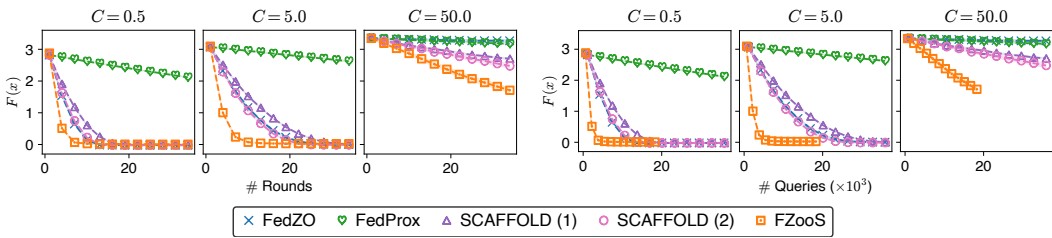

Figure 1: Comparison of the communication round and query efficiency between our FZooS and other existing baselines on the federated synthetic functions with varying client heterogeneity (controlled by $C \geq 0$), where a larger $C$ implies larger client heterogeneity. The $x$-axes of the first and last three plots are the number of rounds and total queries required by these algorithms. SCAFFOLD (1) and (2) stand for SCAFFOLD (Type I) and SCAFFOLD (Type II) algorithms, respectively.

communication round efficiency, which can be attributed to the advantages of our (8) as discussed in Sec. 5.1 (see Appx. D for a detailed comparison).

## 6 EXPERIMENTS

In this section, we demonstrate that our FZooS outperforms existing federated ZOO algorithms using synthetic experiments (Sec. 6.1), as well as real-world experiments on federated black-box adversarial attack (Sec. 6.2) and federated non-differentiable metric optimization (6.3).

### 6.1 SYNTHETIC EXPERIMENTS

We firstly employ federated synthetic functions to illustrate the superiority of our proposed FZooS over a number of existing federated ZOO baselines such as FedZO, FedProx, and SCAFFOLD in the federated ZOO setting (see Appx. D for their specific forms). We refer to Appx. E.1 for the details of these synthetic functions and the experimental setting applied here. Fig. 1 provides the results with $d = 300$, $N = 5$, and varying $C$ to control the client heterogeneity (more results in Appx. F.1). It shows that our FZooS considerably outperforms the other baselines in terms of both communication round and query efficiency, which can be attributed to the superiority of our (8). When $C$ is increased, a larger number of communication rounds and total queries is required to achieve the same convergence error, which empirically verifies our Thm. 2. Interestingly, SCAFFOLD (Type II) consistently outperforms SCAFFOLD (Type I) while Type II in fact is an approximation of Type I in (Karimireddy et al., 2020a). This is likely because SCAFFOLD (Type II) achieves improved gradient correction by implicitly increasing the number of additional function queries for a smaller approximation error of $\nabla F$ (refer to Appx. D). This thus indicates the necessity of achieving an accurate approximation of $\nabla F$ for federated ZOO with heterogeneous clients, which is achieved by our FZooS. Meanwhile, when client heterogeneity is small (i.e., $C \leq 5.0$), both FedProx and SCAFFOLD (Type I) perform worse than FedZO which does not apply any gradient correction. This is likely because the impact of the inaccurate gradient correction applied in these two algorithms outweighs that of client heterogeneity as justified in our Appx. D. This corroborates the importance of developing improved gradient correction for federated ZOO of varying client heterogeneity.

### 6.2 FEDERATED BLACK-BOX ADVERSARIAL ATTACK

Following the practice of (Fang et al., 2022), we then examine the advantages of our FZooS in the task of federated black-box adversarial attack. Here we aim to find a small perturbation $\boldsymbol{x}$ to be added to an input image $\boldsymbol{z}$ such that the perturbed image $\boldsymbol{z} + \boldsymbol{x}$ will be wrongly classified by the *majority* of the private ML models on various clients through only the function queries of these models. Specifically, we randomly select 15 images from CIFAR-10 (Krizhevsky et al., 2009) and then attempt to find one single perturbation ( $d = 32 \times 32$ ) for every image to make the averaged output of $N = 10$ deep neural networks trained using private datasets on different clients misclassify the image using federated ZOO algorithms (refer to Appx. E.2 for more details). Fig. 2 illustrates the success rates on these 15 images achieved by various federated ZOO algorithms during optimization (more results in Appx. F.2). Remarkably, our FZooS again achieves consistently improved communication round efficiency over the other baselines under varying client heterogeneity. Thanks to this improved

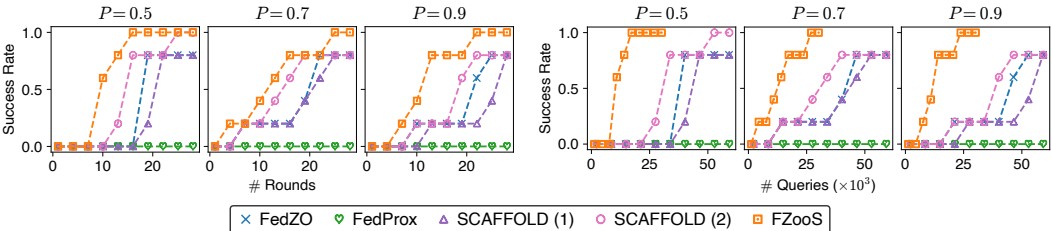

Figure 2: Comparison of the success rate in federated black-box adversarial attack achieved by FZooS and other existing federated ZOO algorithms on CIFAR-10 under varying client heterogeneity (controlled by $P \in [0, 1]$, a larger $P$ implies smaller client heterogeneity). The $x$ and $y$-axis are the number of rounds/queries and the corresponding success rate (higher is better).

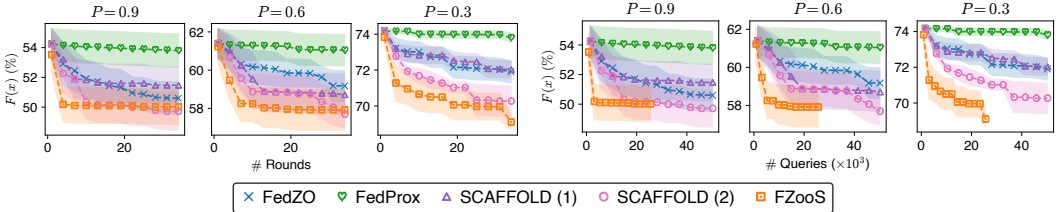

Figure 3: Comparison of the non-differentiable metric optimization between FZooS and other existing federated ZOO algorithms under varying client heterogeneity (controlled by $P \in [0, 1]$, a larger $P$ implies smaller client heterogeneity). The $y$-axis is $(1 - \text{precision}) \times 100\%$ and each curve is the mean $\pm$ standard error from five independent runs.

communication round efficiency and the ability of (8) to avoid a large number of additional function queries in every communication round, FZooS also achieves a substantial improvement in query efficiency. Overall, these results support the superiority of FZooS over the other existing approaches in real-world federated ZOO problems in terms of both communication round and query efficiency.

### 6.3 Federated Non-Differentiable Metric Optimization

Inspired by (Shu et al., 2023), we lastly demonstrate the superior performance of our FZooS in the task of federated non-differentiable metric optimization, which has received a surging interest recently (Hiranandani et al., 2021; Huang et al., 2021). Specifically, we employ federated ZOO algorithms to fine-tune a fully trained MLP model ($d = 2189$) to optimize a non-differentiable metric such as precision and recall, using the Covertype dataset (Dua and Graff, 2017) distributed on $N = 7$ clients (refer to Appx. E.3 for more details). This is similar to the widely applied federated learning setting (McMahan et al., 2017a) whereas the gradient information here is unavailable due to the non-differentiability of these metrics. Fig. 3 reports the comparison among various federated ZOO algorithms under varying client heterogeneity (more results in Appx. F.3). The results show that in the task of federated non-differentiable metric optimization with varying client heterogeneity, our FZooS is still able to consistently outperform the other existing federated ZOO algorithms in terms of both communication round and query efficiency, which therefore further substantiates the superiority of our FZooS in optimizing high-dimensional non-differentiable functions in the federated setting.

## 7 Conclusion and Discussion

In this paper, we first identify the challenges of query and communication round inefficiency faced by federated ZOO algorithms in the presence of client heterogeneity (Sec. 3) and then introduce our FZooS algorithm to address these challenges (Sec. 4). We employ both theoretical justifications (Sec. 5) and empirical demonstrations (Sec. 6) to show that FZooS is indeed able to address these challenges and consequently to achieve considerably improved query and communication round efficiency over the existing federated ZOO algorithms. Of note, the limitation of our FZooS lies in one major aspect: As discussed in Sec. 4.2.1, FZooS incurs an additional transmission of $M$-dimensional vectors for every communication round compared with existing algorithms, which therefore results in a trade-off between communication overhead and communication rounds. However, in the scenario when communication rounds are more important to be reduced (e.g., to reduce the total queries of expensive-to-evaluate local function on every client), our FZooS will be more advanced.

## REPRODUCIBILITY STATEMENT

In regards to our theoretical results, we have explicated underlying assumptions in the main paper and comprehensive proofs in Appx. C. As for our empirical findings, we have divulged our thorough experimental configurations in Appx. E and furnished our software codes in the supplementary resources, specifically the zip file.

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

## APPENDIX A    RELATED WORK

**Federated Learning and Federated First-Order Optimization.**    Federated learning (FL) has become a paradigm of applying multiple edge devices (i.e., clients) to collaboratively train a global model without sharing the private data on these edge devices (McMahan et al., 2017a). We refer to the surveys (Li et al., 2020b; Kairouz et al., 2021) for more comprehensive reviews of FL. Such a paradigm then gives rise to recent interest in federated optimization or more precisely federated first-order optimization (FOO) (Wang et al., 2021) to broaden its real-world application. Since the first federated FOO algorithm FedAvg proposed in (McMahan et al., 2017b), a number of techniques have been developed to further improve its performance in different aspects, e.g., federated FOO with momentum (Wang et al., 2020) and adaptive learning rates (Reddi et al., 2021; Yuan and Ma, 2020; Jin et al., 2022) for convergence speedup, federated FOO with local posterior sampling for de-biased client updates (Al-Shedivat et al., 2021), and federated FOO with regularized functions (Li et al., 2020a; 2019) and control variates (Karimireddy et al., 2020a;b) for the challenge of heterogeneous clients, in which the global function to be optimized differs from the local functions on clients.

**Federated Zeroth-Order Optimization.**    Despite the success of federated FOO algorithms, some important applications, e.g., federated black-box adversarial attack in (Fang et al., 2022), suggests the development of federated zeroth-order (ZOO) algorithms for the federated optimization where gradient information is not available. Nevertheless, very limited efforts have been devoted to the development of federated zeroth-order (ZOO) algorithms especially when the clients are heterogeneous. To the best of our knowledge, Fang et al. (2022) are the first to consider federated ZOO, in which they simply combine FedAvg with existing FD methods as their FedZO algorithm. Similar to the FedAvg algorithm in federated FOO, the FedZO algorithm also likely performs poorly in the heterogeneous setting. This thus encourages the design of federated ZOO algorithms for heterogeneous federated ZOO problems. Following the practice of FedZO, existing federated FOO algorithms for heterogeneous clients, e.g., (Li et al., 2020a; Karimireddy et al., 2020a), can be simply adapted to the corresponding federated ZOO algorithms for this kind of problem. However, these algorithms shall be query- and communication round-inefficient in practice, which therefore raises the question of how to improve query efficiency and the communication round efficiency of these algorithms. To answer this question, we first identify the challenges of such an improvement and then develop a federated ZOO algorithm to overcome these challenges in this paper.

## APPENDIX B    RANDOM FOURIER FEATURES

According to (Rahimi and Recht, 2007), the random Fourier features can usually be represented as a $M$-dimensional row vector $\phi(\boldsymbol{x})^\top = \left[\frac{2}{\sqrt{M}}\cos(\boldsymbol{v}_j\boldsymbol{x} + b_j)\right]_{j=1}^{M}$ where every $\boldsymbol{v}_j$ is independently randomly sampled from a distribution $p(\boldsymbol{v})$ and every $b_j$ is independently randomly sampled from the uniform distribution over $[0, 2\pi]$. Particularly, for the squared exponential kernel $k(\boldsymbol{x}, \boldsymbol{x}') = \exp\left(-\|\boldsymbol{x} - \boldsymbol{x}'\|^2 / (2l^2)\right)$ in which $l$ is the length scale, $p(\boldsymbol{v}) = \mathcal{N}(0, \frac{1}{l^2}\mathbf{I})$. In FZooS, we typically adopt the squared exponential kernel for the optimization. Importantly, before the start of our FZooS, $\{\boldsymbol{v}_j\}_{j=1}^{M}$ and $\{b_j\}_{j=1}^{M}$ need to be sampled and shared across all clients as well as server (as mentioned in Sec. 4.2.1), which however will only happen once for whole optimization process.

## APPENDIX C    THEORETICAL ANALYSES

### C.1    PROOF OF PROPOSITION 1

Based on the definition of $\Xi_{r,t}^{(i)}$ in Sec. 3.2, we have that

$$
\begin{aligned}
\Xi_{r,t}^{(i)} &= \left\| \widehat{\boldsymbol{g}}_{r,t-1}^{(i)} - \nabla F(\boldsymbol{x}_{r,t-1}^{(i)}) \right\|^2 \\
&= \left\| \boldsymbol{g}_{r,t-1}^{(i)} + \gamma_{r,t-1}^{(i)} \left( \boldsymbol{g}_{r-1}(\boldsymbol{x}') - \boldsymbol{g}_{r-1}^{(i)}(\boldsymbol{x}'') \right) - \nabla F(\boldsymbol{x}_{r,t-1}^{(i)}) \right\|^2 \\
&= \left\| \boldsymbol{g}_{r,t-1}^{(i)} - \nabla F(\boldsymbol{x}_{r,t-1}^{(i)}) \right\|^2 - 2\gamma_{r,t-1}^{(i)} \left( \nabla F(\boldsymbol{x}_{r,t-1}^{(i)}) - \boldsymbol{g}_{r,t-1}^{(i)} \right)^\top \left( \boldsymbol{g}_{r-1}(\boldsymbol{x}') - \boldsymbol{g}_{r-1}^{(i)}(\boldsymbol{x}'') \right) + \\
&\quad \left( \gamma_{r,t-1}^{(i)} \right)^2 \left\| \boldsymbol{g}_{r-1}(\boldsymbol{x}') - \boldsymbol{g}_{r-1}^{(i)}(\boldsymbol{x}'') \right\|^2 ,
\end{aligned}
\tag{9}
$$

which is a quadratic function w.r.t. $\gamma_{r,t-1}^{(i)}$. It is easy to show that when

$$
\gamma_{r,t-1}^{(i)} = \gamma_{r,t-1}^{(i)*} \triangleq \frac{\left( \nabla F(\boldsymbol{x}_{r,t-1}^{(i)}) - \boldsymbol{g}_{r,t-1}^{(i)} \right)^\top \left( \boldsymbol{g}_{r-1}(\boldsymbol{x}') - \boldsymbol{g}_{r-1}^{(i)}(\boldsymbol{x}'') \right)}{\left\| \boldsymbol{g}_{r-1}(\boldsymbol{x}') - \boldsymbol{g}_{r-1}^{(i)}(\boldsymbol{x}'') \right\|} ,
\tag{10}
$$

$\Xi_{r,t}^{(i)}$ can achieve its global minimum w.r.t. $\gamma_{r,t-1}^{(i)}$ as

$$
\Xi_{r,t}^{(i)} = \left\| \boldsymbol{g}_{r,t-1}^{(i)} - \nabla F(\boldsymbol{x}_{r,t-1}^{(i)}) \right\|^2 - \frac{\left\| \left( \nabla F(\boldsymbol{x}_{r,t-1}^{(i)}) - \boldsymbol{g}_{r,t-1}^{(i)} \right)^\top \left( \boldsymbol{g}_{r-1}(\boldsymbol{x}') - \boldsymbol{g}_{r-1}^{(i)}(\boldsymbol{x}'') \right) \right\|^2}{\left\| \boldsymbol{g}_{r-1}(\boldsymbol{x}') - \boldsymbol{g}_{r-1}^{(i)}(\boldsymbol{x}'') \right\|^2} .
\tag{11}
$$

This therefore finishes the proof of the fist-part result in Prop. 1. Interestingly, (11) implies that given the $\gamma_{r,t-1}^{(i)}$ in (10), a better alignment between the gradient correction vector $\boldsymbol{g}_{r-1}(\boldsymbol{x}') - \boldsymbol{g}_{r-1}^{(i)}(\boldsymbol{x}'')$ and the shift $\nabla F(\boldsymbol{x}_{r,t-1}^{(i)}) - \boldsymbol{g}_{r,t-1}^{(i)}$ leads to a smaller gradient disparity $\Xi_{r,t}^{(i)}$.

Given the $\gamma_{r,t-1}^{(i)*} = 1$ in (10), when $\boldsymbol{g}_{r-1}(\boldsymbol{x}') - \boldsymbol{g}_{r-1}^{(i)}(\boldsymbol{x}'') = \nabla F(\boldsymbol{x}_{r,t-1}^{(i)}) - \boldsymbol{g}_{r,t-1}^{(i)}$, we can easily verify that $\Xi_{r,t}^{(i)}$ in (10) has $\Xi_{r,t}^{(i)} = 0$. On the contrary, when $\Xi_{r,t}^{(i)} = 0$, we have that

$$
\left\| \boldsymbol{g}_{r,t-1}^{(i)} - \nabla F(\boldsymbol{x}_{r,t-1}^{(i)}) \right\| = \frac{\left\| \left( \nabla F(\boldsymbol{x}_{r,t-1}^{(i)}) - \boldsymbol{g}_{r,t-1}^{(i)} \right)^\top \left( \boldsymbol{g}_{r-1}(\boldsymbol{x}') - \boldsymbol{g}_{r-1}^{(i)}(\boldsymbol{x}'') \right) \right\|}{\left\| \boldsymbol{g}_{r-1}(\boldsymbol{x}') - \boldsymbol{g}_{r-1}^{(i)}(\boldsymbol{x}'') \right\|} ,
\tag{12}
$$

which implies that $\nabla F(\boldsymbol{x}_{r,t-1}^{(i)}) - \boldsymbol{g}_{r,t-1}^{(i)}$ and $\boldsymbol{g}_{r-1}(\boldsymbol{x}') - \boldsymbol{g}_{r-1}^{(i)}(\boldsymbol{x}'')$ are linear dependent according to the Cauchy-Schwarz inequality. Since $\gamma_{r,t-1}^{(i)*} = 1$, we further have

$$
\left\| \nabla F(\boldsymbol{x}_{r,t-1}^{(i)}) - \boldsymbol{g}_{r,t-1}^{(i)} \right\| = \left\| \boldsymbol{g}_{r-1}(\boldsymbol{x}') - \boldsymbol{g}_{r-1}^{(i)}(\boldsymbol{x}'') \right\| .
\tag{13}
$$

These two results, i.e., (12) and (13) thus imply that $\nabla F(\boldsymbol{x}_{r,t-1}^{(i)}) - \boldsymbol{g}_{r,t-1}^{(i)} = \boldsymbol{g}_{r-1}(\boldsymbol{x}') - \boldsymbol{g}_{r-1}^{(i)}(\boldsymbol{x}'')$, which therefore concludes our proof.

## C.2    PROOF OF THEOREM 1

### C.2.1    GRADIENT ESTIMATION ERROR USING UNCERTAINTY

We introduce the following lemma that is adapted from (Shu et al., 2023) to bound the estimation error of our local gradient surrogates using the uncertainty measure in our (5).

**Lemma C.1.** *Let* $\delta \in (0, 1)$ *and* $\omega \triangleq d + 2(\sqrt{d} + 1)\ln(1/\delta)$. *For any* $\boldsymbol{x} \in \mathcal{X}$, $i \in [N]$, $r \geq 1$ *and* $t \geq 1$, *the following holds with probability of at least* $1 - \delta$,

$$\left\| \nabla \mu_{r,t}^{(i)}(\boldsymbol{x}) - \nabla f_i(\boldsymbol{x}) \right\|^2 \leq \omega \left\| \partial \left( \sigma_{r,t}^{(i)} \right)^2 (\boldsymbol{x}) \right\| .$$

### C.2.2    RFF APPROXIMATION ERROR FOR GLOBAL GRADIENT SURROGATE

**Lemma C.2** (Laurent and Massart (2000)). *If* $\mathrm{x}_1, \cdots, \mathrm{x}_k$ *are independent standard normal random variables, for* $\mathrm{y} = \sum_{i=1}^{k} \mathrm{x}_i^2$ *and any* $\epsilon$,

$$\mathbb{P}(\mathrm{y} - k \geq 2\sqrt{k\epsilon} + 2\epsilon) \leq \exp(-\epsilon) .$$

Following the general idea in (Rahimi and Recht, 2007), we present the following Lemma C.3 to bound the difference of our approximated kernel using random features and the ground truth kernel $k$, as well as the difference between their partial derivatives first. To ease our presentation, we let the kernel $k$ be defined by an infinite dimensional vector $\psi(\boldsymbol{x})$, which is defined by the corresponding infinite number of features for $k$, throughout this section. That is, $k(\boldsymbol{x}, \boldsymbol{x}') = \psi(\boldsymbol{x})^\top \psi(\boldsymbol{x}')$ for any $\boldsymbol{x}, \boldsymbol{x}' \in \mathcal{X}$.

**Lemma C.3.** *Let* $\delta \in (0, 1)$. *Assume that* $\mathbb{E}\left[ \|\boldsymbol{v}\|^2 \right] \leq V$, *for any* $\boldsymbol{x}, \boldsymbol{x}' \in \mathcal{X}$, *the following holds with probability of at least* $1 - \delta$,

$$\left| \boldsymbol{\phi}(\boldsymbol{x})^\top \boldsymbol{\phi}(\boldsymbol{x}') - \boldsymbol{\psi}(\boldsymbol{x})^\top \boldsymbol{\psi}(\boldsymbol{x}') \right| \leq \sqrt{8\ln(2/\delta)/M} ,$$
$$\left\| \nabla \boldsymbol{\phi}(\boldsymbol{x})^\top \boldsymbol{\phi}(\boldsymbol{x}') - \nabla \boldsymbol{\psi}(\boldsymbol{x})^\top \boldsymbol{\psi}(\boldsymbol{x}') \right\| \leq \sqrt{4V/(M\delta)}$$

*where* $M$ *is the number of random Fourier features.*

*Proof.* Recall that $\boldsymbol{\phi}(\boldsymbol{x})^\top \boldsymbol{\phi}(\boldsymbol{x}') = 1/M \sum_{j=1}^{M} 2\cos(\boldsymbol{v}_j^\top \boldsymbol{x} + b_j)\cos(\boldsymbol{v}_j^\top \boldsymbol{x}' + b_j)$ as shown in Appx. B. Then, according to (Rahimi and Recht, 2007), for any $j \in [M]$,

$$\mathbb{E}\left[ 2\cos(\boldsymbol{v}_j^\top \boldsymbol{x} + b_j)\cos(\boldsymbol{v}_j^\top \boldsymbol{x}' + b_j) \right] = \boldsymbol{\psi}(\boldsymbol{x})^\top \boldsymbol{\psi}(\boldsymbol{x}') ,$$
$$\mathbb{E}\left[ \boldsymbol{\phi}(\boldsymbol{x})^\top \boldsymbol{\phi}(\boldsymbol{x}') \right] = \boldsymbol{\psi}(\boldsymbol{x})^\top \boldsymbol{\psi}(\boldsymbol{x}') . \tag{14}$$

Since $2\cos(\boldsymbol{v}_j^\top \boldsymbol{x} + b_j)\cos(\boldsymbol{v}_j^\top \boldsymbol{x}' + b_j) \in [-2, 2]$ and both $\{\boldsymbol{v}_1, \cdots, \boldsymbol{v}_M\}$ and $\{b_1, \cdots, b_M\}$ are randomly independently sampled, according to Hoeffding's inequality, the following inequality holds for any $\epsilon > 0$

$$\mathbb{P}\left( \left| \boldsymbol{\phi}(\boldsymbol{x})^\top \boldsymbol{\phi}(\boldsymbol{x}') - \boldsymbol{\psi}(\boldsymbol{x})^\top \boldsymbol{\psi}(\boldsymbol{x}') \right| \geq \epsilon \right) \leq 2\exp\left( -\frac{M\epsilon^2}{8} \right) . \tag{15}$$

Choose $\delta = 2\exp(M\epsilon^2)$, the following holds with a probability of at least $1 - \delta$,

$$\left| \boldsymbol{\phi}(\boldsymbol{x})^\top \boldsymbol{\phi}(\boldsymbol{x}') - \boldsymbol{\psi}(\boldsymbol{x})^\top \boldsymbol{\psi}(\boldsymbol{x}') \right| \leq \sqrt{\frac{8\ln(2/\delta)}{M}} . \tag{16}$$

Moreover, based on the interchangeability of derivative and expectation, we then have the following results derived from (14)

$$\mathbb{E}\left[ -2\sin(\boldsymbol{v}_j^\top \boldsymbol{x} + b_j)\cos(\boldsymbol{v}_j^\top \boldsymbol{x}' + b_j)\boldsymbol{v}_j^\top \right] = \nabla \boldsymbol{\psi}(\boldsymbol{x})^\top \boldsymbol{\psi}(\boldsymbol{x}') ,$$
$$\mathbb{E}\left[ \nabla \boldsymbol{\phi}(\boldsymbol{x})^\top \boldsymbol{\phi}(\boldsymbol{x}') \right] = \nabla \boldsymbol{\psi}(\boldsymbol{x})^\top \boldsymbol{\psi}(\boldsymbol{x}') . \tag{17}$$

Since both $\{\boldsymbol{v}_1, \cdots, \boldsymbol{v}_M\}$ and $\{b_1, \cdots, b_M\}$ are randomly independently sampled, we then can bound the variance $\mathbb{E}\left[\left\|\nabla\boldsymbol{\phi}(\boldsymbol{x})^\top \boldsymbol{\phi}(\boldsymbol{x}') - \nabla\boldsymbol{\psi}(\boldsymbol{x})^\top \boldsymbol{\psi}(\boldsymbol{x}')\right\|^2\right]$ as below

$$
\begin{aligned}
& \mathbb{E}\left[\left\|\nabla\boldsymbol{\phi}(\boldsymbol{x})^\top \boldsymbol{\phi}(\boldsymbol{x}') - \nabla\boldsymbol{\psi}(\boldsymbol{x})^\top \boldsymbol{\psi}(\boldsymbol{x}')\right\|^2\right] \\
&\stackrel{(a)}{=} \mathbb{E}\left[\left\|\frac{1}{M}\sum_{j=1}^{M}\left(-2\sin(\boldsymbol{v}_j^\top \boldsymbol{x} + b_j)\cos(\boldsymbol{v}_j^\top \boldsymbol{x}' + b_j)\boldsymbol{v}_j - \nabla\boldsymbol{\psi}(\boldsymbol{x})^\top \boldsymbol{\psi}(\boldsymbol{x}')\right)\right\|^2\right] \\
&\stackrel{(b)}{=} \frac{1}{M^2}\mathbb{E}\left[\sum_{j=1}^{M}\left\|-2\sin(\boldsymbol{v}_j^\top \boldsymbol{x} + b_j)\cos(\boldsymbol{v}_j^\top \boldsymbol{x}' + b_j)\boldsymbol{v}_j - \nabla\boldsymbol{\psi}(\boldsymbol{x})^\top \boldsymbol{\psi}(\boldsymbol{x}')\right\|^2\right] \\
&\stackrel{(c)}{=} \frac{1}{M^2}\sum_{j=1}^{M}\left(\mathbb{E}\left[\left\|-2\sin(\boldsymbol{v}_j^\top \boldsymbol{x} + b_j)\cos(\boldsymbol{v}_j^\top \boldsymbol{x}' + b_j)\boldsymbol{v}_j\right\|^2\right] - \mathbb{E}\left[\left\|\nabla\boldsymbol{\psi}(\boldsymbol{x})^\top \boldsymbol{\psi}(\boldsymbol{x}')\right\|^2\right]\right) \quad (18) \\
&\stackrel{(d)}{\leq} \frac{1}{M^2}\sum_{j=1}^{M}\mathbb{E}\left[\left\|-2\sin(\boldsymbol{v}_j^\top \boldsymbol{x} + b_j)\cos(\boldsymbol{v}_j^\top \boldsymbol{x}' + b_j)\boldsymbol{v}_j\right\|^2\right] \\
&\stackrel{(e)}{\leq} \frac{4}{M^2}\sum_{j=1}^{M}\mathbb{E}\left[\left\|\boldsymbol{v}_j\right\|^2\right] \\
&\stackrel{(f)}{\leq} \frac{4V}{M}
\end{aligned}
$$

where $(b)$ is from the independence among $\{\boldsymbol{v}_1, \cdots, \boldsymbol{v}_M\}$ and $\{b_1, \cdots, b_M\}$ for variance derivation and $(c)$ is based on the definition of variance. In addition, $(e)$ is due to the fact that $\sin(\boldsymbol{v}_j^\top \boldsymbol{x} + b_j), \cos(\boldsymbol{v}_j^\top \boldsymbol{x}' + b_j) \in [-1, 1]$ and $(f)$ is because of the assumption that $\mathbb{E}\left[\left\|\boldsymbol{v}\right\|^2\right] \leq V$.

Therefore, according to Chebyshev's inequality, we have the following inequalities for any $\epsilon > 0$

$$
\begin{aligned}
\mathbb{P}\left(\left\|\nabla\boldsymbol{\phi}(\boldsymbol{x})^\top \boldsymbol{\phi}(\boldsymbol{x}') - \nabla\boldsymbol{\psi}(\boldsymbol{x})^\top \boldsymbol{\psi}(\boldsymbol{x}')\right\| > \epsilon\right) &\leq \frac{\mathbb{E}\left[\left\|\nabla\boldsymbol{\phi}(\boldsymbol{x})^\top \boldsymbol{\phi}(\boldsymbol{x}') - \nabla\boldsymbol{\psi}(\boldsymbol{x})^\top \boldsymbol{\psi}(\boldsymbol{x}')\right\|^2\right]}{\epsilon^2} \\
&\leq \frac{4V}{M\epsilon^2} .
\end{aligned}
\quad (19)
$$

Choose $\epsilon = \sqrt{4V/(M\delta)}$, the following holds for a probability of at least $1 - \delta$,

$$
\left\|\nabla\boldsymbol{\phi}(\boldsymbol{x})^\top \boldsymbol{\phi}(\boldsymbol{x}') - \nabla\boldsymbol{\psi}(\boldsymbol{x})^\top \boldsymbol{\psi}(\boldsymbol{x}')\right\| \leq \sqrt{\frac{4V}{M\delta}} ,
\quad (20)
$$

which finally completes the proof. $\square$

**Lemma C.4.** *For any* $\boldsymbol{x}, \boldsymbol{x}' \in \mathcal{X}$ *and* $i \in [N]$*, assume that* $\mathbb{E}\left[\left\|\boldsymbol{v}\right\|^2\right] \leq V, \left\|\nabla\boldsymbol{\psi}(\boldsymbol{x})^\top \boldsymbol{\psi}(\boldsymbol{x}')\right\| \leq L$ *and* $|f_i(\boldsymbol{x})| \leq 1$*, then the following holds with a constant probability for all* $r \in [R]$*,*

$$
\left\|\nabla\widehat{\mu}_{r,T}^{(i)}(\boldsymbol{x}) - \nabla\mu_{r,T}^{(i)}(\boldsymbol{x})\right\|^2 \leq \mathcal{O}\left(\frac{1}{M}\right) .
$$

*Proof.* Based on the definition in (5) and (6), we have that:

$$
\begin{aligned}
&\left\| \nabla \widehat{\mu}_{r,T}^{(i)}(\boldsymbol{x}) - \nabla \mu_{r,T}^{(i)}(\boldsymbol{x}) \right\| \\
&\overset{(a)}{=} \left\| \nabla \boldsymbol{\phi}(\boldsymbol{x})^\top \boldsymbol{\Phi}_{r,t-1}^{(i)} \left( \widehat{\mathbf{K}}_{r,T}^{(i)} + \sigma^2 \mathbf{I} \right)^{-1} \boldsymbol{y}_{r,T}^{(i)} - \nabla \boldsymbol{\psi}(\boldsymbol{x})^\top \boldsymbol{\Psi}_{r,T}^{(i)} \left( \mathbf{K}_{r,T}^{(i)} + \sigma^2 \mathbf{I} \right)^{-1} \boldsymbol{y}_{r,T}^{(i)} \right\| \\
&\overset{(b)}{\leq} \left\| \nabla \boldsymbol{\phi}(\boldsymbol{x})^\top \boldsymbol{\Phi}_{r,T}^{(i)} \left( \widehat{\mathbf{K}}_{r,T}^{(i)} + \sigma^2 \mathbf{I} \right)^{-1} - \nabla \boldsymbol{\psi}(\boldsymbol{x})^\top \boldsymbol{\Psi}_{r,T}^{(i)} \left( \mathbf{K}_{r,T}^{(i)} + \sigma^2 \mathbf{I} \right)^{-1} \right\| \left\| \boldsymbol{y}_{r,T}^{(i)} \right\| \\
&\overset{(c)}{=} \underbrace{\left\| \nabla \boldsymbol{\phi}(\boldsymbol{x})^\top \boldsymbol{\Phi}_{r,T}^{(i)} \left( \widehat{\mathbf{K}}_{r,T}^{(i)} + \sigma^2 \mathbf{I} \right)^{-1} - \nabla \boldsymbol{\psi}(\boldsymbol{x})^\top \boldsymbol{\Psi}_{r,T}^{(i)} \left( \widehat{\mathbf{K}}_{r,T}^{(i)} + \sigma^2 \mathbf{I} \right)^{-1} \right\| \left\| \boldsymbol{y}_{r,T}^{(i)} \right\|}_{\textcircled{1}} + \\
&\quad \underbrace{\left\| \nabla \boldsymbol{\psi}(\boldsymbol{x})^\top \boldsymbol{\Psi}_{r,T}^{(i)} \left( \widehat{\mathbf{K}}_{r,T}^{(i)} + \sigma^2 \mathbf{I} \right)^{-1} - \nabla \boldsymbol{\psi}(\boldsymbol{x})^\top \boldsymbol{\Psi}_{r,T}^{(i)} \left( \mathbf{K}_{r,T}^{(i)} + \sigma^2 \mathbf{I} \right)^{-1} \right\| \left\| \boldsymbol{y}_{r,T}^{(i)} \right\|}_{\textcircled{2}}
\end{aligned}
\tag{21}
$$

where $(b)$ and $(c)$ are from the Cauchy–Schwarz inequality and the triangle inequality, respectively.

We bound term $\textcircled{1}$, term $\textcircled{2}$ and $\left\| \boldsymbol{y}_{r,T}^{(i)} \right\|$ above separately. Firstly, the following holds with probability of at least $1 - rT\delta'$

$$
\begin{aligned}
\textcircled{1} &\overset{(a)}{=} \left\| \nabla \boldsymbol{\phi}(\boldsymbol{x})^\top \boldsymbol{\Phi}_{r,T}^{(i)} \left( \widehat{\mathbf{K}}_{r,T}^{(i)} + \sigma^2 \mathbf{I} \right)^{-1} - \nabla \boldsymbol{\psi}(\boldsymbol{x})^\top \boldsymbol{\Psi}_{r,T}^{(i)} \left( \widehat{\mathbf{K}}_{r,T}^{(i)} + \sigma^2 \mathbf{I} \right)^{-1} \right\| \\
&\overset{(b)}{\leq} \left\| \nabla \boldsymbol{\phi}(\boldsymbol{x})^\top \boldsymbol{\Phi}_{r,T}^{(i)} - \nabla \boldsymbol{\psi}(\boldsymbol{x})^\top \boldsymbol{\Psi}_{r,T}^{(i)} \right\| \left\| \left( \widehat{\mathbf{K}}_{r,T}^{(i)} + \sigma^2 \mathbf{I} \right)^{-1} \right\| \\
&\overset{(c)}{\leq} \sqrt{\sum_{\tau=1}^{rT} \left\| \nabla \boldsymbol{\phi}(\boldsymbol{x})^\top \boldsymbol{\phi}(\boldsymbol{x}_\tau^{(i)}) - \nabla \boldsymbol{\psi}(\boldsymbol{x})^\top \boldsymbol{\psi}(\boldsymbol{x}_\tau^{(i)}) \right\|^2} \left\| \left( \widehat{\mathbf{K}}_{r,T}^{(i)} + \sigma^2 \mathbf{I} \right)^{-1} \right\| \\
&\overset{(d)}{\leq} \frac{1}{\sigma^2} \sqrt{\frac{4rTV}{M\delta'}}
\end{aligned}
\tag{22}
$$

Where $(b)$ comes from the Cauchy–Schwarz inequality and $(c)$ follows from the fact that for any matrix $A$ with $n$ rows and each row identified as $\boldsymbol{a}_i$ we have $\|A\| \leq \|A\|_{\mathrm{F}} \triangleq \sqrt{\sum_{i=1}^n \|\boldsymbol{a}_i\|^2}$. Finally, $(d)$ is due to the fact that $\widehat{\mathbf{K}}_{r,T}^{(i)}$ is positive semi-definite and therefore $\widehat{\mathbf{K}}_{r,T}^{(i)} + \sigma^2 \mathbf{I} \succcurlyeq \sigma^2 \mathbf{I}$ as well as the results in Lemma C.3.

Secondly, the following holds with probability of at least $1 - r^2 T^2 \delta''$,

$$
\begin{aligned}
\textcircled{2} &\overset{(a)}{=} \left\| \nabla \boldsymbol{\psi}(\boldsymbol{x})^\top \boldsymbol{\Psi}_{r,T}^{(i)} \left( \widehat{\mathbf{K}}_{r,T}^{(i)} + \sigma^2 \mathbf{I} \right)^{-1} - \nabla \boldsymbol{\psi}(\boldsymbol{x})^\top \boldsymbol{\Psi}_{r,T}^{(i)} \left( \mathbf{K}_{r,T}^{(i)} + \sigma^2 \mathbf{I} \right)^{-1} \right\| \\
&\overset{(b)}{\leq} \left\| \nabla \boldsymbol{\psi}(\boldsymbol{x})^\top \boldsymbol{\Psi}_{r,t-1}^{(i)} \right\| \left\| \left( \widehat{\mathbf{K}}_{r,T}^{(i)} + \sigma^2 \mathbf{I} \right)^{-1} - \left( \mathbf{K}_{r,T}^{(i)} + \sigma^2 \mathbf{I} \right)^{-1} \right\| \\
&\overset{(c)}{=} \left\| \nabla \boldsymbol{\psi}(\boldsymbol{x})^\top \boldsymbol{\Psi}_{r,T}^{(i)} \right\| \left\| \left( \mathbf{K}_{r,T}^{(i)} - \widehat{\mathbf{K}}_{r,T}^{(i)} \right) \left( \widehat{\mathbf{K}}_{r,T}^{(i)} + \sigma^2 \mathbf{I} \right)^{-1} \left( \mathbf{K}_{r,T}^{(i)} + \sigma^2 \mathbf{I} \right)^{-1} \right\| \\
&\overset{(d)}{\leq} \sqrt{\sum_{\tau=1}^{rT} \left\| \nabla \boldsymbol{\psi}(\boldsymbol{x})^\top \boldsymbol{\psi}(\boldsymbol{x}_\tau^{(i)}) \right\|^2} \left\| \mathbf{K}_{r,T}^{(i)} - \widehat{\mathbf{K}}_{r,T}^{(i)} \right\| \left\| \left( \widehat{\mathbf{K}}_{r,T}^{(i)} + \sigma^2 \mathbf{I} \right)^{-1} \right\| \left\| \left( \mathbf{K}_{r,T}^{(i)} + \sigma^2 \mathbf{I} \right)^{-1} \right\| \\
&\overset{(e)}{\leq} \frac{L}{\sigma^4} \sqrt{rT} \sqrt{\sum_{\tau,\tau'=1}^{rT} \left\| \boldsymbol{\psi}(\boldsymbol{x}_\tau^{(i)})^\top \boldsymbol{\psi}(\boldsymbol{x}_{\tau'}^{(i)}) - \boldsymbol{\phi}(\boldsymbol{x}_\tau^{(i)})^\top \boldsymbol{\phi}(\boldsymbol{x}_{\tau'}^{(i)}) \right\|^2} \\
&\overset{(f)}{\leq} \frac{L (rT)^{3/2}}{\sigma^4} \sqrt{\frac{8 \ln(2/\delta'')}{M}}
\end{aligned}
\tag{23}
$$

where $(b)$ is from the Cauchy–Schwarz inequality. Besides, $(c)$ and $(e)$ come from the aforementioned inequality $\|A\| \le \|A\|_F$. In addition, $(f)$ is based on the assumption that $\left\|\nabla\psi(\boldsymbol{x})^\top \psi(\boldsymbol{x}')\right\| \le L$, $\|A\| \le \|A\|_F$, $\widehat{\mathbf{K}}_{r,T}^{(i)} + \sigma^2 \mathbf{I} \succcurlyeq \sigma^2 \mathbf{I}$ and $\mathbf{K}_{r,T}^{(i)} + \sigma^2 \mathbf{I} \succcurlyeq \sigma^2 \mathbf{I}$.

Thirdly, the following holds with probability of at least $1 - rT\delta'''$,

$$
\begin{aligned}
\left\|\boldsymbol{y}_{r,T}^{(i)}\right\| &\overset{(a)}{=} \sqrt{\sum_{\tau=1}^{rT} \left(f_i(\boldsymbol{x}_\tau) + \zeta_\tau\right)^2} \\
&\overset{(b)}{\le} \sqrt{\sum_{\tau=1}^{rT} 2f_i^2(\boldsymbol{x}_\tau) + 2\zeta_\tau^2} \\
&\overset{(c)}{\le} \sqrt{2rT + 2\sigma^2 \sum_{\tau=1}^{rT} \left(\frac{\zeta_\tau}{\sigma}\right)^2} \\
&\overset{(d)}{\le} \sqrt{2rT + 2\sigma^2 \left(rT + 2\sqrt{rT \ln(1/\delta''')} + 2\ln(1/\delta''')\right)}
\end{aligned}
\tag{24}
$$

where $\zeta_\tau$ denote the observation noise associated with the input $\boldsymbol{x}_\tau$. Besides, $(c)$ is from the assumption that $\zeta_\tau \sim \mathcal{N}(0, \sigma^2)$ for any $\tau$ in Sec. 2 and $|f_i(\boldsymbol{x})| \le 1$ for any $\boldsymbol{x} \in \mathcal{X}$. Finally, $(d)$ comes from our Lemma C.2.

By introducing (22), (23) and (24) with $\delta' = \frac{\delta}{3rT}$, $\delta'' = \frac{\delta}{3r^2T^2}$ and $\delta''' = \frac{\delta}{3rT}$ into (21), the following then holds with probability of at least $1 - \delta$,

$$
\begin{aligned}
&\left\|\nabla\widehat{\mu}_{r,T}^{(i)}(\boldsymbol{x}) - \nabla\mu_{r,T}^{(i)}(\boldsymbol{x})\right\| \\
&\le \left(\frac{rT}{\sigma^2}\sqrt{\frac{12V}{M\delta}} + \frac{4L(rT)^{3/2}}{\sigma^4}\sqrt{\frac{\ln(6rT/\delta)}{M}}\right) \sqrt{2rT + 2\sigma^2\left(rT + 2\sqrt{rT\ln(3rT/\delta)} + 2\ln(3rT/\delta)\right)} \\
&= \mathcal{O}\left(\frac{rT\sqrt{rT}}{\sqrt{M}} + \frac{r^2T^2\sqrt{\ln(rT)}}{\sqrt{M}}\right) .
\end{aligned}
\tag{25}
$$

Of note, it is easy to show that when (25) holds for $r = R$, it must hold for any $r \le R$. Therefore, the following finally holds with a constant probability for all $r \in [R]$,

$$
\left\|\nabla\widehat{\mu}_{r,T}^{(i)}(\boldsymbol{x}) - \nabla\mu_{r,T}^{(i)}(\boldsymbol{x})\right\|^2 \le \mathcal{O}\left(\frac{1}{M}\right) ,
\tag{26}
$$

which concludes our proof. $\qquad\square$

**Remark.** Note that the assumption $\mathbb{E}\left[\|\boldsymbol{v}\|^2\right] \le V$ implies that the distribution $p(\boldsymbol{v})$ in Appx. B has a bounded mean and covariance since $\mathbb{E}\left[\|\boldsymbol{v}\|^2\right] = \|\mathbb{E}[\boldsymbol{v}]\|^2 + \mathbb{E}\left[\|\boldsymbol{v} - \mathbb{E}[\boldsymbol{v}]\|^2\right]$. This is usually valid for the widely applied kernels (e.g., the squared exponential kernel in Appx. B) in practice.

Remarkably, (25) with $r = R$ has demonstrated that a larger number $M$ of random features is preferred to maintain the approximation quality of $\nabla\widehat{\mu}_{R,T}^{(i)}(\boldsymbol{x}) \approx \nabla\mu_{R,T}^{(i)}$ when the number $R$ of communication rounds and the number $T$ of local iterations increase. This in fact aligns with the intuition that a larger hypothesis space (defined by the $M$ random features) should be used when the target function (defined by the existing $RT$ function queries) becomes more complex. However, for any communication round $r + 1 \in [R]$ in our FZooS, the approximation of $\nabla\mu_{r,T}^{(i)}$ using $\nabla\widehat{\mu}_{r,T}^{(i)}(\boldsymbol{x})$ needs to be accurate only at the local updated inputs $\{\boldsymbol{x}_{r+1,t-1}^{(i)}\}_{t\in[T],i\in[N]}$ with a relatively small $T$ (i.e., $T \le 20$), which consequently usually does not requires an extremely large $M$ to realize a good approximation quality in practice. This has actually been supported by the empirical results in our Sec. 6 and Appx. F.

### C.2.3 FINAL GRADIENT DISPARITY ANALYSIS USING UNCERTAINTY

We introduce the following Lemma C.5 and Lemma C.6 from (Shu et al., 2023) to ease our proof of Thm. 1:

**Lemma C.5.** *Let $\{v_1, \ldots, v_\tau\}$ be any $\tau$ vectors in $\mathbb{R}^d$. Then the following holds for any $a > 0$:*

$$\|v_i\| \|v_j\| \le \frac{a}{2} \|v_i\|^2 + \frac{1}{2a} \|v_j\|^2 \ , \tag{27}$$

$$\|v_i + v_j\|^2 \le (1 + a) \|v_i\|^2 + \left(1 + \frac{1}{a}\right) \|v_j\|^2 \ , \tag{28}$$

$$\left\| \sum_{i=1}^{\tau} v_i \right\|^2 \le \tau \sum_{i=1}^{\tau} \|v_i\|^2 \ . \tag{29}$$

*Proof.* For (27), we have that

$$\frac{a}{2} \|v_i\|^2 + \frac{1}{2a} \|v_j\|^2 \ge 2\sqrt{\frac{a}{2} \|v_i\|^2 \cdot \frac{1}{2a} \|v_j\|^2} = \|v_i\| \|v_j\| \ . \tag{30}$$

For (28), we have that

$$
\begin{aligned}
(1 + a) \|v_i\|^2 + \left(1 + \frac{1}{a}\right) \|v_j\|^2 &= \|v_i\|^2 + \|v_j\|^2 + \left(a \|v_i\|^2 + \frac{1}{a} \|v_j\|^2\right) \\
&\ge \|v_i\|^2 + \|v_j\|^2 + 2\sqrt{a \|v_i\|^2 \cdot \frac{1}{a} \|v_j\|^2} \\
&= \|v_i + v_j\|^2 \ .
\end{aligned}
\tag{31}
$$

For (29), we can directly employ the convexity of function $h(x) = \|x\|^2$ and Jensen's inequality:

$$\left\| \frac{1}{\tau} \sum_{i=1}^{\tau} v_i \right\|^2 \le \frac{1}{\tau} \sum_{i=1}^{\tau} \|v_i\|^2 \ . \tag{32}$$

By multiplying the inequality above with $\tau^2$, we conclude the proof. □

**Lemma C.6.** *Define $\rho_i \triangleq \max_{x \in \mathcal{X}, r \ge 1, t \ge 1} \left\| \partial \left(\sigma_{r,t}^{(i)}\right)^2 (x) \right\| / \left\| \partial \left(\sigma_{r,t-1}^{(i)}\right)^2 (x) \right\|$, we have that $\rho_i \in \left[1/(1 + 1/\sigma^2), 1\right]$, and that for any $x \in \mathcal{X}, r \ge 1, t \ge 1$ the following holds,*

$$\left\| \partial \left(\sigma_{r,t}^{(i)}\right)^2 (x) \right\| \le \kappa \rho_i^{(r-1)T+t} \ .$$

Let $\delta \in (0,1)$, $\epsilon = \mathcal{O}(\frac{1}{M})$ and $\omega = d + 2(\sqrt{d} + 1)\ln(2NRT/\delta)$, the following inequalities then hold with a probability of at least $1 - \delta$:

$$
\left\| \frac{1}{N} \sum_{j=1,j\neq i}^{N} \left( \nabla\widehat{\mu}_{r-1,T}^{(j)}(\boldsymbol{x}_{r,t-1}^{(i)}) - \nabla f_j(\boldsymbol{x}_{r,t-1}^{(i)}) \right) \right\|^2
$$

$$
\overset{(a)}{\leq} \frac{N-1}{N^2} \sum_{j=1,j\neq i}^{N} \left\| \nabla\widehat{\mu}_{r-1,T}^{(j)}(\boldsymbol{x}_{r,t-1}^{(i)}) - \nabla f_j(\boldsymbol{x}_{r,t-1}^{(i)}) \right\|^2
$$

$$
\overset{(b)}{=} \frac{N-1}{N^2} \sum_{j=1,j\neq i}^{N} \left\| \nabla\widehat{\mu}_{r-1,T}^{(j)}(\boldsymbol{x}_{r,t-1}^{(i)}) - \nabla\mu_{r-1,T}^{(j)}(\boldsymbol{x}_{r,t-1}^{(i)}) + \nabla\mu_{r-1,T}^{(j)}(\boldsymbol{x}_{r,t-1}^{(i)}) - \nabla f_j(\boldsymbol{x}_{r,t-1}^{(i)}) \right\|^2
$$

$$
\overset{(c)}{\leq} \frac{N-1}{N^2} \sum_{j=1,j\neq i}^{N} \left( \frac{N}{N-1} \left\| \nabla\mu_{r-1,T}^{(j)}(\boldsymbol{x}_{r,t-1}^{(i)}) - \nabla f_j(\boldsymbol{x}_{r,t-1}^{(i)}) \right\|^2 + N \left\| \nabla\widehat{\mu}_{r-1,T}^{(j)}(\boldsymbol{x}_{r,t-1}^{(i)}) - \nabla\mu_{r-1,T}^{(j)}(\boldsymbol{x}_{r,t-1}^{(i)}) \right\|^2 \right)
$$

$$
\overset{(d)}{\leq} \frac{\omega}{N} \sum_{j=1,j\neq i}^{N} \left\| \partial \left( \sigma_{r-1,T}^{(j)} \right)^2 (\boldsymbol{x}_{r,t-1}^{(i)}) \right\| + \frac{(N-1)^2}{N}\epsilon \,,
$$

$$\tag{33}$$

in which $(a)$ is from (29) and $(c)$ is from (28) with $a = \frac{1}{N-1}$. In addition, $(d)$ comes from Lemma C.1 and Lemma C.4.

$$
\frac{(N-1)^2}{N^2} \left\| \nabla f_i(\boldsymbol{x}_{r,t-1}^{(i)}) - \nabla\widehat{\mu}_{r-1,T}^{(i)}(\boldsymbol{x}_{r,t-1}^{(i)}) \right\|^2
$$

$$
\overset{(a)}{=} \frac{(N-1)^2}{N^2} \left\| \nabla f_i(\boldsymbol{x}_{r,t-1}^{(i)}) - \nabla\mu_{r-1,T}^{(i)}(\boldsymbol{x}_{r,t-1}^{(i)}) + \nabla\mu_{r-1,T}^{(i)}(\boldsymbol{x}_{r,t-1}^{(i)}) - \nabla\widehat{\mu}_{r-1,T}^{(i)}(\boldsymbol{x}_{r,t-1}^{(i)}) \right\|^2
$$

$$
\overset{(b)}{\leq} \frac{(N-1)^2}{N^2} \left( \frac{N}{N-1} \left\| \nabla f_i(\boldsymbol{x}_{r,t-1}^{(i)}) - \nabla\mu_{r-1,T}^{(i)}(\boldsymbol{x}_{r,t-1}^{(i)}) \right\|^2 + N \left\| \nabla\mu_{r-1,T}^{(i)}(\boldsymbol{x}_{r,t-1}^{(i)})\nabla\widehat{\mu}_{r-1,T}^{(i)}(\boldsymbol{x}_{r,t-1}^{(i)}) \right\|^2 \right)
$$

$$
\overset{(c)}{\leq} \left( \frac{\omega(N-1)}{N} \left\| \partial \left( \sigma_{r-1,T}^{(i)} \right)^2 (\boldsymbol{x}_{r,t-1}^{(i)}) \right\| + \frac{(N-1)^2}{N}\epsilon \right) \,,
$$

$$\tag{34}$$

in which $(c)$ is from (28) with $a = \frac{1}{N-1}$. In addition, $(d)$ comes from Lemma C.1 and Lemma C.4.

By exploiting the inequalities above, we have

$$\frac{1}{N}\sum_{i=1}^{N}\Xi_{r,t}^{(i)}$$

$$\overset{(a)}{=}\frac{1}{N}\sum_{i=1}^{N}\left\|\nabla\mu_{r,t-1}^{(i)}(\boldsymbol{x}_{r,t-1}^{(i)})+\gamma_{r,t-1}\left(\nabla\widehat{\mu}_{r-1}(\boldsymbol{x}_{r,t-1}^{(i)})-\nabla\widehat{\mu}_{r-1,T}^{(i)}(\boldsymbol{x}_{r,t-1}^{(i)})\right)-\nabla F(\boldsymbol{x}_{r,t-1}^{(i)})\right\|^{2}$$

$$\overset{(b)}{=}\frac{1}{N}\sum_{i=1}^{N}\left\|\nabla\mu_{r,t-1}^{(i)}(\boldsymbol{x}_{r,t-1}^{(i)})-\nabla f_{i}(\boldsymbol{x}_{r,t-1}^{(i)})+\gamma_{r,t-1}\left(\frac{1}{N}\sum_{j=1,j\neq i}^{N}\left(\nabla\widehat{\mu}_{r-1,T}^{(j)}(\boldsymbol{x}_{r,t-1}^{(i)})-\nabla f_{j}(\boldsymbol{x}_{r,t-1}^{(i)})\right)\right)+\right.$$

$$\left.\frac{\gamma_{r,t-1}(N-1)}{N}\left(\nabla f_{i}(\boldsymbol{x}_{r,t-1}^{(i)})-\nabla\widehat{\mu}_{r-1,T}^{(i)}(\boldsymbol{x}_{r,t-1}^{(i)})\right)+(1-\gamma_{r,t-1})\left(\nabla f_{i}(\boldsymbol{x}_{r,t-1}^{(i)})-\nabla F(\boldsymbol{x}_{r,t-1}^{(i)})\right)\right\|^{2}$$

$$\overset{(c)}{\leq}\frac{1}{N}\sum_{i=1}^{N}\left(4\left\|\nabla\mu_{r,t-1}^{(i)}(\boldsymbol{x}_{r,t-1}^{(i)})-\nabla f_{i}(\boldsymbol{x}_{r,t-1}^{(i)})\right\|^{2}+4\gamma_{r,t-1}^{2}\left\|\frac{1}{N}\sum_{j=1,j\neq i}^{N}\left(\nabla\widehat{\mu}_{r-1,T}^{(j)}(\boldsymbol{x}_{r,t-1}^{(i)})-\nabla f_{j}(\boldsymbol{x}_{r,t-1}^{(i)})\right)\right\|^{2}+\right.$$

$$\left.\frac{4\gamma_{r,t-1}^{2}(N-1)^{2}}{N^{2}}\left\|\nabla f_{i}(\boldsymbol{x}_{r,t-1}^{(i)})-\nabla\widehat{\mu}_{r-1,T}^{(i)}(\boldsymbol{x}_{r,t-1}^{(i)})\right\|^{2}+4(1-\gamma_{r,t-1})^{2}\left\|\nabla f_{i}(\boldsymbol{x}_{r,t-1}^{(i)})-\nabla F(\boldsymbol{x}_{r,t-1}^{(i)})\right\|^{2}\right)$$

$$\overset{(d)}{\leq}\frac{4\omega}{N}\sum_{i=1}^{N}\left\|\partial\left(\sigma_{r,t-1}^{(i)}\right)(\boldsymbol{x}_{r,t-1}^{(i)})\right\|+4\gamma_{r,t-1}^{2}\left(\frac{\omega}{N^{2}}\sum_{i=1}^{N}\sum_{j=1,j\neq i}^{N}\left\|\partial\left(\sigma_{r-1,T}^{(j)}\right)^{2}(\boldsymbol{x}_{r,t-1}^{(i)})\right\|+\frac{(N-1)^{2}}{N}\epsilon\right)+$$

$$4\gamma_{r,t-1}^{2}\left(\frac{\omega(N-1)}{N^{2}}\sum_{i=1}^{N}\left\|\partial\left(\sigma_{r-1,T}^{(i)}\right)^{2}(\boldsymbol{x}_{r,t-1}^{(i)})\right\|+\frac{(N-1)^{2}}{N}\epsilon\right)+4(1-\gamma_{r,t-1})^{2}G$$

$$(35)$$

where $(c)$ is from the (29). In addition, $(d)$ is from Lemma C.1, (33) and (34).

By introducing the results in Lemma C.6 into (35), we have

$$\frac{1}{N}\sum_{i=1}^{N}\Xi_{r,t}^{(i)}\overset{(a)}{\leq}\frac{4\omega}{N}\sum_{i=1}^{N}\kappa\rho_{i}^{(r-1)T+t+1}+4\gamma_{r,t-1}^{2}\left(\frac{2\omega(N-1)}{N^{2}}\sum_{i=1}^{N}\kappa\rho_{i}^{(r-1)T}+\frac{2(N-1)^{2}}{N}\epsilon\right)$$

$$+4(1-\gamma_{r,t-1})^{2}G$$

$$\overset{(b)}{\leq}\frac{4\omega}{N}\sum_{i=1}^{N}\kappa\rho_{i}^{(r-1)T+t+1}+4\gamma_{r,t-1}^{2}\left(\frac{2\omega}{N}\sum_{i=1}^{N}\kappa\rho_{i}^{(r-1)T}+2N\epsilon\right)+4(1-\gamma_{r,t-1})^{2}G$$

$$\overset{(c)}{\leq}4\omega\kappa\rho^{(r-1)T+t+1}+4\gamma_{r,t-1}^{2}\left(2\omega\kappa\rho^{(r-1)T}+2N\epsilon\right)+4(1-\gamma_{r,t-1})^{2}G$$

$$(36)$$

where $(c)$ is from Jansen's inequality with $\rho\triangleq\frac{1}{N}\sum_{i=1}^{N}\rho_{i}$. This finally concludes our proof.

**Remark.** Of note, the upper bound in our Thm. 1 is a quadratic function w.r.t. the gradient correction length $\gamma_{r,t-1}$. As a consequence, it is easy to verify that in order to minimize the upper bound in our Thm. 1 (i.e., to achieve a better-performing (8)) w.r.t. $\gamma_{r,t-1}$, $\gamma_{r,t-1}$ needs to be chosen as

$$\gamma_{r,t-1}=\frac{G}{G+2\omega\rho^{(r-1)T}+2N\epsilon}\,,\tag{37}$$

as shown in our Cor. 1. This better-performing $\gamma_{r,t-1}$ therefore implies that *(a)* an adaptive $\gamma_{r,t-1}$ is indeed able to theoretically reduce the gradient disparity, which therefore aligns with the conclusion from our Prop. 1 and *(b)* when the estimation error of our gradient correction vector (characterized by $2\omega\rho^{rT}+2N\epsilon$) in (8) is smaller than the client heterogeneity (characterized by $G$), a large $\gamma_{t-1}$ is suggested to be applied in order to minimize the gradient disparity $\frac{1}{N}\sum_{i=1}^{N}\Xi_{r,t}^{(i)}$, as shown in our Sec. 5.1.

By introducing this $\gamma_{r,t-1}$ into the upper bound in Thm. 1, we have

$$
\begin{aligned}
\frac{1}{N}\sum_{i=1}^{N}\Xi_{r,t}^{(i)} &\overset{(a)}{\leq} 4\omega\kappa\rho^{(r-1)T+t-1} + 4\gamma_{r,t-1}^2\left(2\omega\kappa\rho^{(r-1)T} + 2N\epsilon\right) + 4(1-\gamma_{r,t-1})^2 G \\
&\overset{(b)}{=} 4\omega\kappa\rho^{(r-1)T+t-1} + \frac{4G\left(2\omega\kappa\rho^{(r-1)T} + 2N\epsilon\right)}{G + \left(2\omega\kappa\rho^{(r-1)T} + 2N\epsilon\right)} \\
&\overset{(c)}{\leq} 4\omega\kappa\rho^{(r-1)T+t-1} + 2\sqrt{2G(\omega\kappa\rho^{(r-1)T} + N\epsilon)} \\
&\overset{(d)}{\leq} 4\omega\kappa\rho^{(r-1)T+t-1} + 2\sqrt{2\omega\kappa\rho^{(r-1)T}G} + 2\sqrt{2NG\epsilon}
\end{aligned}
\tag{38}
$$

where $(c)$ is from the inequality of $G + 2\omega\rho^{(r-1)T} + 2N\epsilon \geq 2\sqrt{G(2\omega\rho^{(r-1)T} + 2N\epsilon)}$ (i.e., the relationship between the geometric mean and arithmetic mean of $G$ and $2\omega\rho^{(r-1)T} + 2N\epsilon$) and $(d)$ is from the fact that $(\sqrt{2\omega\kappa\rho^{(r-1)T}G} + \sqrt{2NG\epsilon})^2 > 2\omega\kappa\rho^{(r-1)T}G + 2NG\epsilon$. Interestingly, (38) enjoys two major aspects. *(a)* In contrast to the algorithm where $\gamma_{r,t-1} = 0$ (e.g., FedZO), the impact of client heterogeneity (i.e., $G$) is able to be reduced in our FZooS through decreasing the estimation error of our gradient surrogates (i.e., $\omega\kappa\rho^{(r-1)T}$) and the RFF approximation error (i.e., $\epsilon$) for our global gradient surrogates. *(b)* In contrast to the federated ZOO algorithms where $\gamma_{r,t-1} = 1$ (e.g., SCAFFOLD), the impact of the large estimation error of our gradient surrogates (i.e., $\omega\kappa\rho^{(r-1)T}$) is also able to be mitigated in our FZooS through a small client heterogeneity (i.e., $G$) in practice. As a result, these advantages will intuitively make our FZooS produce more robust optimization performance under different scenarios in practice, as supported by our Sec. 6 and Appx. F.

C.3 GRADIENT ESTIMATION ANALYSIS BASED ON EUCLIDEAN DISTANCE

Of note, for every iteration $t$ of round $r$, our global gradient surrogate in Sec. 4.2.1 is obtained based on the optimization trajectory $\mathcal{D}_{r-1,T}^{(i)} = \{(\boldsymbol{x}_\tau^{(i)}, y_\tau^{(i)})\}_{\tau=1}^{T(r-1)}$ and is not capable of being updated immediately although $t-1$ new function queries are given at this time. This is because the update of our global gradient surrogate only occurs when clients and server can communicate with each other, i.e., at the end of each round. Intuitively, this will result in the phenomenon that the quality of our global gradient surrogate and hence the quality of our (8) decays w.r.t. the iterations of local updates, as empirically supported in Appx. F.1. This is likely because the Euclidean distance between the input to be evaluated in our global gradient surrogate and the queried inputs from the optimization trajectory becomes larger and consequently the optimization trajectory becomes less informative. Unfortunately, such a quality decay within the local updates fails to be captured in Thm. 1 and hence may result in an impractical choice of $\gamma_{r,t-1}$ in Cor. 1. To this end, we develop another uncertainty analysis of our global gradient surrogate that is based on Euclidean distance to capture such a phenomenon in this section, which finally gives us a more practical choice of gradient correction length.

We first introduce the following lemma to ease our proof in this section.

**Lemma C.7.** *For any matrix* $\mathbf{A}$, $\mathbf{A}^\top \mathbf{A}$ *and* $\mathbf{A}\mathbf{A}^\top$ *share the same non-zero eigenvalues.*

*Proof.* Let $\lambda$ be any non-zero eigenvalue of $\mathbf{A}^\top\mathbf{A}$, for some $\boldsymbol{x} \neq \boldsymbol{0}$, we have

$$\mathbf{A}^\top\mathbf{A}\boldsymbol{x} = \lambda\boldsymbol{x} \ . \tag{39}$$

By multiplying $\mathbf{A}$ on both sides above, we have

$$\mathbf{A}\mathbf{A}^\top (\mathbf{A}\boldsymbol{x}) = \lambda (\mathbf{A}\boldsymbol{x}) \ , \tag{40}$$

which implies that $\lambda$ is also the eigenvalue of $\mathbf{A}\mathbf{A}^\top$ with $\mathbf{A}\boldsymbol{x}$ being the eigenvector. Following the same proof, it is easy to show that any non-zero eigenvalue of $\mathbf{A}\mathbf{A}^\top$ remains the eigenvalue of $\mathbf{A}^\top\mathbf{A}$, which therefore concludes the proof. $\square$

We then introduce another estimation error analysis (different from the one presented in Appx. C.2) of our global gradient surrogate as follows where we slightly abuse the notation and use $\boldsymbol{x}_\tau^{(i)} \in \mathcal{D}_{r,T}^{(i)}$ to denote that $\boldsymbol{x}_\tau^{(i)}$ is from the optimization trajectory $\mathcal{D}_{r,T}^{(i)}$.

**Proposition C.1.** *Let the shift-invariant kernel* $k(\boldsymbol{x}, \boldsymbol{x}') = k(\|\boldsymbol{x} - \boldsymbol{x}'\|^2)$ *where* $k(\cdot)$ *is assumed to be non-increasing and function* $h(\iota) = \iota\nabla k(\iota)^2$ *is assumed to be convex, the following then holds with a probability of at least* $1 - \delta$ *for any* $\boldsymbol{x} \in \mathcal{X}$,

$$\|\nabla\mu_r(\boldsymbol{x}) - \nabla F(\boldsymbol{x})\|^2 \leq \omega\kappa - \frac{4\omega\iota_r^2\nabla k(\iota_r)^2}{k(0)d + \sigma^2 d/(rT)}$$

*where* $\omega = d + 2(\sqrt{d} + 1)\ln(1/\delta)$, $\iota_r \triangleq \frac{1}{rNT}\sum_{i=1}^{N}\sum_{\boldsymbol{x}_\tau^{(i)} \in \mathcal{D}_{r,T}^{(i)}}\left\|\boldsymbol{x} - \boldsymbol{x}_\tau^{(i)}\right\|^2$, *and* $k(0) = k(\boldsymbol{x}, \boldsymbol{x})$.

*Proof.* Recall that the uncertainty measure function (see (5)) of our local gradient surrogate on client $i$ for iteration $T$ of round $r$ will be

$$\begin{aligned}
\partial\left(\sigma_{r,T}^{(i)}\right)^2(\boldsymbol{x}) &= \partial_{\boldsymbol{z}}\partial_{\boldsymbol{z}'}k(\boldsymbol{z}, \boldsymbol{z}') - \partial_{\boldsymbol{z}}\boldsymbol{k}_{r,T}^{(i)}(\boldsymbol{z})^\top \left(\mathbf{K}_{r,T}^{(i)} + \sigma^2\mathbf{I}\right)^{-1}\partial_{\boldsymbol{z}'}\boldsymbol{k}_{r,T}^{(i)}(\boldsymbol{z}')\Big|_{\boldsymbol{z}=\boldsymbol{z}'=\boldsymbol{x}} \\
&\stackrel{(a)}{\preccurlyeq} \kappa\mathbf{I} - \left(\lambda_{\max}(\mathbf{K}_{r,T}^{(i)}) + \sigma^2\right)^{-1}\partial_{\boldsymbol{z}}\boldsymbol{k}_{r,T}^{(i)}(\boldsymbol{z})^\top\partial_{\boldsymbol{z}'}\boldsymbol{k}_{r,T}^{(i)}(\boldsymbol{z}')\Big|_{\boldsymbol{z}=\boldsymbol{z}'=\boldsymbol{x}} \\
&\stackrel{(b)}{\preccurlyeq} \kappa\mathbf{I} - \frac{\partial_{\boldsymbol{z}}\boldsymbol{k}_{r,T}^{(i)}(\boldsymbol{z})^\top\partial_{\boldsymbol{z}'}\boldsymbol{k}_{r,T}^{(i)}(\boldsymbol{z}')\big|_{\boldsymbol{z}=\boldsymbol{z}'=\boldsymbol{x}}}{rT\max_{\boldsymbol{x},\boldsymbol{x}' \in \mathcal{D}_{r,T}^{(i)}}k(\boldsymbol{x}, \boldsymbol{x}') + \sigma^2}
\end{aligned} \tag{41}$$

where $(a)$ is based on the assumption on $\partial_{\boldsymbol{z}}\partial_{\boldsymbol{z}'}k(\boldsymbol{z}, \boldsymbol{z}')$ in our Sec. 2 and the definition of maximum eigenvalue. In addition, $(b)$ comes from $\lambda_{\max}(\mathbf{K}_{r,T}^{(i)}) \leq rT\max_{\boldsymbol{x},\boldsymbol{x}' \in \mathcal{D}_{r,T}^{(i)}}k(\boldsymbol{x}, \boldsymbol{x}')$ (i.e., the Gershgorin theorem).

Based on the assumption that $k(\boldsymbol{x}, \boldsymbol{x}') = k(\|\boldsymbol{x} - \boldsymbol{x}'\|^2)$ and $k(\cdot)$ is non-increasing, we have

$$\max_{\boldsymbol{x}, \boldsymbol{x}' \in \mathcal{D}_{r,T}^{(i)}} k(\boldsymbol{x}, \boldsymbol{x}') \leq k(\boldsymbol{x}, \boldsymbol{x}) = k(0) . \tag{42}$$

Moreover, define $\iota \triangleq \|\boldsymbol{z} - \boldsymbol{z}'\|^2$, the partial derivative of kernel $k(\cdot, \cdot)$ will be

$$\partial_{\boldsymbol{z}} k(\boldsymbol{z}, \boldsymbol{z}') = 2 (\boldsymbol{z} - \boldsymbol{z}') \nabla k(\iota)$$
$$\partial_{\boldsymbol{z}'} k(\boldsymbol{z}, \boldsymbol{z}') = 2 (\boldsymbol{z}' - \boldsymbol{z}) \nabla k(\iota) . \tag{43}$$

Therefore, the each element in the $rT \times rT$ matrix $\partial_{\boldsymbol{z}} \boldsymbol{k}_{r,T}^{(i)}(\boldsymbol{z}) \partial_{\boldsymbol{z}'} \boldsymbol{k}_{r,T}^{(i)}(\boldsymbol{z}')^\top |_{\boldsymbol{z}=\boldsymbol{z}'=\boldsymbol{x}}$ that is induced by the input pair $(\boldsymbol{x}_\tau^{(i)}, \boldsymbol{x}_{\tau'}^{(i)})$ with $\boldsymbol{x}_\tau^{(i)}, \boldsymbol{x}_{\tau'}^{(i)} \in \mathcal{D}_{r,T}^{(i)}$ and $\tau, \tau' \in [rT]$ will be:

$$4 \left( \boldsymbol{x} - \boldsymbol{x}_\tau^{(i)} \right)^\top \left( \boldsymbol{x} - \boldsymbol{x}_{\tau'}^{(i)} \right) \nabla k(\iota_\tau^{(i)}) \nabla k(\iota_{\tau'}^{(i)}) \tag{44}$$

where $\iota_\tau^{(i)} \triangleq \left\| \boldsymbol{x} - \boldsymbol{x}_\tau^{(i)} \right\|^2, \iota_{\tau'}^{(i)} \triangleq \left\| \boldsymbol{x} - \boldsymbol{x}_{\tau'}^{(i)} \right\|^2$. Based on these results, the trace norm $\|\cdot\|_{\mathrm{tr}}$ of $\partial_{\boldsymbol{z}} \boldsymbol{k}_{r,T}^{(i)}(\boldsymbol{z}) \partial_{\boldsymbol{z}'} \boldsymbol{k}_{r,T}^{(i)}(\boldsymbol{z}')^\top |_{\boldsymbol{z}=\boldsymbol{z}'=\boldsymbol{x}}$ will be

$$\left\| \partial_{\boldsymbol{z}} \boldsymbol{k}_{r,T}^{(i)}(\boldsymbol{z}) \partial_{\boldsymbol{z}'} \boldsymbol{k}_{r,T}^{(i)}(\boldsymbol{z}')^\top |_{\boldsymbol{z}=\boldsymbol{z}'=\boldsymbol{x}} \right\|_{\mathrm{tr}} = \sum_{\tau=1}^{rT} 4 \|\boldsymbol{x} - \boldsymbol{x}_\tau\|^2 \nabla k(\iota_\tau)^2$$
$$= \sum_{\tau=1}^{rT} 4 \iota_\tau \nabla k(\iota_\tau)^2 . \tag{45}$$

By further assuming that the function $h(\iota) = \iota \nabla k(\iota)^2$ is convex, we then have

$$\left\| \partial_{\boldsymbol{z}} \boldsymbol{k}_{r,T}^{(i)}(\boldsymbol{z})^\top \partial_{\boldsymbol{z}'} \boldsymbol{k}_{r,T}^{(i)}(\boldsymbol{z}')|_{\boldsymbol{z}=\boldsymbol{z}'=\boldsymbol{x}} \right\| \overset{(a)}{\geq} \frac{1}{d} \left\| \partial_{\boldsymbol{z}} \boldsymbol{k}_{r,T}^{(i)}(\boldsymbol{z})^\top \partial_{\boldsymbol{z}'} \boldsymbol{k}_{r,T}^{(i)}(\boldsymbol{z}')|_{\boldsymbol{z}=\boldsymbol{z}'=\boldsymbol{x}} \right\|_{\mathrm{tr}}$$
$$\overset{(b)}{=} \frac{1}{d} \left\| \partial_{\boldsymbol{z}} \boldsymbol{k}_{r,T}^{(i)}(\boldsymbol{z}) \partial_{\boldsymbol{z}'} \boldsymbol{k}_{r,T}^{(i)}(\boldsymbol{z}')^\top |_{\boldsymbol{z}=\boldsymbol{z}'=\boldsymbol{x}} \right\|_{\mathrm{tr}}$$
$$\overset{(c)}{=} \frac{1}{d} \sum_{\tau=1}^{rT} 4 \iota_\tau^{(i)} \nabla k(\iota_\tau^{(i)})^2$$
$$\overset{(d)}{\geq} \frac{4rT}{d} \iota_r^{(i)} \nabla k(\iota_r^{(i)})^2 \tag{46}$$

where $(a)$ comes from the fact the maximum eigenvalue of a matrix is always larger or equal to its averaged eigenvalues and $(b)$ is based on Lemma C.7. In addition, $(c)$ is obtained from (45) while $(d)$ results from the definition of $\iota_r^{(i)} \triangleq \frac{1}{rT} \sum_{\boldsymbol{x}_\tau^{(i)} \in \mathcal{D}_{r,T}^{(i)}} \left\| \boldsymbol{x} - \boldsymbol{x}_\tau^{(i)} \right\|^2$ as well as the Jansen's inequality for the convex function $h(\cdot)$.

Finally, by introducing the results above, i.e., (42) and (46), into (41), we have

$$\left\| \partial \left( \sigma_{r,T}^{(i)} \right)^2 (\boldsymbol{x}) \right\| \leq \kappa - \frac{4 \iota_r^{(i)} \nabla k(\iota_r^{(i)})^2}{k(0)d + \sigma^2 d/(rT)} . \tag{47}$$

Define $\iota_r \triangleq \frac{1}{N} \sum_{i=1}^N \bar{\iota}_r^{(i)}$, we then have

$$\|\nabla \mu_r(\boldsymbol{x}) - \nabla F(\boldsymbol{x})\|^2 \overset{(a)}{=} \left\| \frac{1}{N} \sum_{i=1}^N \left( \nabla \mu_{r,T}^{(i)}(\boldsymbol{x}) - \nabla f_i(\boldsymbol{x}) \right) \right\|^2$$
$$\overset{(b)}{\leq} \frac{1}{N} \sum_{i=1}^N \left\| \nabla \mu_{r,T}^{(i)}(\boldsymbol{x}) - \nabla f_i(\boldsymbol{x}) \right\|^2$$
$$\overset{(c)}{\leq} \frac{1}{N} \sum_{i=1}^N \omega \kappa - \frac{4 \omega \iota_r^{(i)} \nabla k(\iota_r^{(i)})^2}{k(0)d + \sigma^2 d/(rT)}$$
$$\overset{(d)}{\leq} \omega \kappa - \frac{4 \omega \iota_r \nabla k(\iota_r)^2}{k(0)d + \sigma^2 d/(rT)} \tag{48}$$

where $(b)$ is from the Cauchy-Schwarz inequality, $(c)$ derives from Lemma C.1, and $(d)$ results from the Jansen's inequality for convex function $h(\cdot)$. which finally concludes the proof. $\qquad\square$

**Remark.** Of note, the assumption that $k(\boldsymbol{x}, \boldsymbol{x}') = k(\|\boldsymbol{x} - \boldsymbol{x}'\|^2)$ where $k(\cdot)$ is non-increasing and function $h(\iota) = \iota \nabla k(\iota)^2$ is convex can be satisfied by the widely applied squared exponential kernel $k(\boldsymbol{x}, \boldsymbol{x}') = \exp\left(-\|\boldsymbol{x} - \boldsymbol{x}'\|^2 / (2l^2)\right)$, which has also been applied in our FZooS. To justify the validity of these assumptions on the squared exponential kernel, we first show that this kernel can be represented as $k(\iota) = \exp\left(-\iota/(2l^2)\right)$, which is non-increasing w.r.t. its input $\iota$, and $h(\iota) = \iota \exp\left(-\iota/l^2\right)/(4l^4)$ is convex when $\iota \geq 2l^2$.

Remarkably, Prop. C.1 reveals that the quality of the gradient estimation at an input $\boldsymbol{x} \in \mathcal{X}$ when using our global gradient surrogate without RFF approximation is highly related to the averaged Euclidean distance between $\boldsymbol{x}$ and $\boldsymbol{x}_\tau \in \bigcup_{i=1}^N \mathcal{D}_{r,T}^{(i)}$ (i.e., $\iota_r$ in Prop. C.1). Specifically, when the input $\boldsymbol{x}$ to be evaluated in our global gradient surrogate leads to a larger value of $\iota_r \nabla k(\iota_r)^2$, the upper bound in our Prop. C.1 demonstrates that the gradient estimation error of our global gradient surrogate tends to be more accurate. Note that when the kernel is the squared exponential kernel, we have that $h(\iota) = \iota \nabla k(\iota)^2 = \iota \exp\left(-\iota/l^2\right)/(4l^4)$ decreases w.r.t. $\iota$ and that a smaller averaged Euclidean distance between $\boldsymbol{x}$ and $\boldsymbol{x}_\tau \in \bigcup_{i=1}^N \mathcal{D}_{r,T}^{(i)}$ likely enjoys a smaller gradient estimation error. This is intuitively aligned with the common practice that $\boldsymbol{x}_\tau \in \bigcup_{i=1}^N \mathcal{D}_{r,T}^{(i)}$ is more informative when it achieves a smaller averaged Euclidean distance with $\boldsymbol{x}$. Intuitively, when the iteration $t$ of local updates is increased, the input $\boldsymbol{x}_{r,t-1}$ to be evaluated in our global gradient surrogate likely achieves a larger distance with the history of function queries $\bigcup_{i=1}^N \mathcal{D}_{r,T}^{(i)}$ and consequently the quality of our global gradient surrogate likely decays, which finally aligns with the phenomenon that we have mentioned at the beginning of this section.

**More Practical Choice of $\gamma_{r,t-1}$.** Finally, by introducing Prop. C.1 into the analysis in Appx. C.2, we achieve the following better-performing choice of gradient correction length $\gamma_{r,t-1}$:

**Corollary C.1.** *Based on our Prop. C.1, a better-performing choice choice of $\gamma_{r,t-1}$ should be*

$$\gamma_{r,t-1} = \frac{G}{G + 2\left(\omega\kappa - \frac{4\omega\iota_r \nabla k(\iota_r)^2}{k(0)d + \sigma^2 d/(rT)} + N\epsilon\right)} .$$

Cor. C.1 implies that $\gamma_{r,t-1}$ should decay w.r.t the iteration $t$ of local updates if $\iota_r \nabla k(\iota_r)^2$ decreases w.r.t. $t$. Particularly, when $k(\boldsymbol{x}, \boldsymbol{x}') = \exp\left(-\|\boldsymbol{x} - \boldsymbol{x}'\|^2 / (2l^2)\right)$ and $\iota_r \nabla k(\iota_r)^2$ decreases at a rate of $\mathcal{O}(\frac{1}{t})$ for the iteration $t$ of local updates, we then have that better-performing choice of $\gamma_{r,t-1}$ in Prop. C.1 has the form of $\gamma_{r,t-1} = \frac{G}{G + C_0 - C_1/t}$ for some constant $C_0 \geq C_1 > 0$. Since we usually have no prior knowledge of client heterogeneity $G$ as well as the constants $C_0, C_1$, we commonly apply the approximated form of $\gamma_{r,t-1} = 1/t$, which will be widely applied in our experiments as shown in our Appx. E.

## C.4 CONVERGENCE OF ALGO. 1

We first introduce the following lemmas that are inspired by the results in (Karimireddy et al., 2020a).

**Lemma C.8.** *For any $\alpha$-strongly convex and $\beta$-smooth function $f$, and any $\boldsymbol{x}, \boldsymbol{y}, \boldsymbol{z}$ in the domain of $f$, we have*

$$\nabla f(\boldsymbol{x})^\top (\boldsymbol{y} - \boldsymbol{z}) \le f(\boldsymbol{y}) - f(\boldsymbol{z}) - \alpha \|\boldsymbol{y} - \boldsymbol{z}\|^2/4 + \beta\|\boldsymbol{z} - \boldsymbol{x}\|^2$$

*Proof.* Since $f$ is both $\alpha$-strongly convex and $\beta$-smooth, we have that

$$
\begin{aligned}
f(\boldsymbol{z}) - f(\boldsymbol{x}) &\le \nabla f(\boldsymbol{x})^\top (\boldsymbol{z} - \boldsymbol{x}) + \frac{\beta}{2} \|\boldsymbol{z} - \boldsymbol{x}\|^2 \\
f(\boldsymbol{y}) - f(\boldsymbol{x}) &\ge \nabla f(\boldsymbol{x})^\top (\boldsymbol{y} - \boldsymbol{x}) + \frac{\alpha}{2} \|\boldsymbol{y} - \boldsymbol{x}\|^2 .
\end{aligned}
\tag{49}
$$

Note that when $\alpha = 0$, the inequalities above still hold. By aggregating the results above, we have

$$
\begin{aligned}
f(\boldsymbol{z}) - f(\boldsymbol{y}) &= f(\boldsymbol{z}) - f(\boldsymbol{x}) + f(\boldsymbol{x}) - f(\boldsymbol{y}) \\
&\le \nabla f(\boldsymbol{x})^\top (\boldsymbol{z} - \boldsymbol{x}) + \nabla f(\boldsymbol{x})^\top (\boldsymbol{x} - \boldsymbol{y}) + \frac{\beta}{2} \|\boldsymbol{z} - \boldsymbol{x}\|^2 - \frac{\alpha}{2} \|\boldsymbol{y} - \boldsymbol{x}\|^2 \\
&\le \nabla f(\boldsymbol{x})^\top (\boldsymbol{z} - \boldsymbol{y}) + \frac{\beta}{2} \|\boldsymbol{z} - \boldsymbol{x}\|^2 - \frac{\alpha}{4} \|\boldsymbol{y} - \boldsymbol{z}\|^2 + \frac{\alpha}{2} \|\boldsymbol{x} - \boldsymbol{z}\|^2 \\
&= \nabla f(\boldsymbol{x})^\top (\boldsymbol{z} - \boldsymbol{y}) + \frac{\beta + \alpha}{2} \|\boldsymbol{z} - \boldsymbol{x}\|^2 - \frac{\alpha}{4} \|\boldsymbol{y} - \boldsymbol{z}\|^2
\end{aligned}
\tag{50}
$$

where the second inequality comes from $\alpha \|\boldsymbol{y} - \boldsymbol{x}\|^2 /2 \ge \alpha \|\boldsymbol{y} - \boldsymbol{z}\|^2 /4 - \alpha \|\boldsymbol{x} - \boldsymbol{z}\|^2 /2$ (triangle inequality). When $\alpha > 0$, since $\beta > \alpha$, we have

$$f(\boldsymbol{z}) - f(\boldsymbol{y}) \le \nabla f(\boldsymbol{x})^\top (\boldsymbol{z} - \boldsymbol{y}) + \beta \|\boldsymbol{z} - \boldsymbol{x}\|^2 - \frac{\alpha}{4} \|\boldsymbol{y} - \boldsymbol{z}\|^2 . \tag{51}$$

By rearranging the inequality above, we can directly derive the result in Lemma C.8 with $\alpha > 0$. Even when $\alpha = 0$, since $\|\boldsymbol{z} - \boldsymbol{x}\|^2 \ge 0$, we have

$$
\begin{aligned}
f(\boldsymbol{z}) - f(\boldsymbol{y}) &\le \nabla f(\boldsymbol{x})^\top (\boldsymbol{z} - \boldsymbol{y}) + \frac{\beta}{2} \|\boldsymbol{z} - \boldsymbol{x}\|^2 \\
&\le \nabla f(\boldsymbol{x})^\top (\boldsymbol{z} - \boldsymbol{y}) + \beta \|\boldsymbol{z} - \boldsymbol{x}\|^2 .
\end{aligned}
\tag{52}
$$

By rearranging the inequality above, we show that the result in Lemma C.8 also holds for $\alpha = 0$. $\square$

**Lemma C.9.** *For any $\beta$-smooth function $f$, inputs $\boldsymbol{x}, \boldsymbol{y}$ in the domain of $f$, the following holds for any $\eta > 0$*

$$\|\boldsymbol{x} - \eta\nabla f(\boldsymbol{x}) - \boldsymbol{y} + \eta\nabla f(\boldsymbol{y})\|^2 \le (1 + \eta\beta)^2 \|\boldsymbol{x} - \boldsymbol{y}\|^2 .$$

*Proof.* Since $f$ is $\beta$-smooth, we have

$$
\begin{aligned}
\|\boldsymbol{x} - \eta\nabla f(\boldsymbol{x}) - \boldsymbol{y} + \eta\nabla f(\boldsymbol{y})\|^2 &\le \left(1 + \frac{1}{a}\right) \|\boldsymbol{x} - \boldsymbol{y}\|^2 + (1 + a)\,\eta^2 \|\nabla f(\boldsymbol{x}) - \nabla f(\boldsymbol{y})\|^2 \\
&\le \left(1 + \frac{1}{a} + (1 + a)\,\eta^2\beta^2\right) \|\boldsymbol{x} - \boldsymbol{y}\|^2
\end{aligned}
\tag{53}
$$

where the first inequality derives from Lemma C.5 and the second inequality comes from the smoothness of $f$. By choosing $a = 1/(\eta\beta)$, we conclude our proof. $\square$

**Remark.** Lemma C.9 only requires the smoothness of function $f$. When $f$ is both $\beta$-smooth and $\alpha$-strongly convex ($\alpha > 0$), we will have a tighter bound as below when $\eta < \alpha/\beta^2$ (see proof below),

$$\|\boldsymbol{x} - \eta\nabla f(\boldsymbol{x}) - \boldsymbol{y} + \eta\nabla f(\boldsymbol{y})\|^2 \le (1 - \eta\alpha)\|\boldsymbol{x} - \boldsymbol{y}\|^2 , \tag{54}$$

which can lead to a better convergence (by achieving a smaller constant term) compared with the inequality (62) we will prove later. However, for simplicity and consistency under various assumptions on the function to be optimized, we only use Lemma C.9 for the convergence analysis of our Thm. 2 in the main paper.

*Proof.* Based on the strong convexity of $f$, for any inputs $\boldsymbol{x}, \boldsymbol{y}$ in the domain of $f$, we have

$$
\begin{aligned}
f(\boldsymbol{y}) - f(\boldsymbol{x}) &\geq \nabla f(\boldsymbol{x})^\top (\boldsymbol{y} - \boldsymbol{x}) + \frac{\alpha}{2} \|\boldsymbol{y} - \boldsymbol{x}\|^2 \ , \\
f(\boldsymbol{x}) - f(\boldsymbol{y}) &\geq \nabla f(\boldsymbol{y})^\top (\boldsymbol{x} - \boldsymbol{y}) + \frac{\alpha}{2} \|\boldsymbol{y} - \boldsymbol{x}\|^2 \ .
\end{aligned}
\tag{55}
$$

By summing up these inequalities, we have

$$
\left(\nabla f(\boldsymbol{y}) - \nabla f(\boldsymbol{x})\right)^\top (\boldsymbol{y} - \boldsymbol{x}) \geq \alpha \|\boldsymbol{y} - \boldsymbol{x}\|^2 \ .
\tag{56}
$$

Finally, we have

$$
\begin{aligned}
&\|\boldsymbol{x} - \eta \nabla f(\boldsymbol{x}) - \boldsymbol{y} + \eta \nabla f(\boldsymbol{y})\|^2 \\
&\overset{(a)}{=} \|\boldsymbol{x} - \boldsymbol{y}\|^2 + \eta^2 \|\nabla f(\boldsymbol{x}) - \nabla f(\boldsymbol{y})\|^2 - 2\eta \left(\nabla f(\boldsymbol{x}) - \nabla f(\boldsymbol{y})\right)^\top (\boldsymbol{x} - \boldsymbol{y}) \\
&\overset{(b)}{\leq} \|\boldsymbol{x} - \boldsymbol{y}\|^2 + \eta^2 \beta^2 \|\boldsymbol{x} - \boldsymbol{y}\|^2 - 2\eta\alpha \|\boldsymbol{x} - \boldsymbol{y}\|^2 \\
&\overset{(c)}{=} \left(1 + \eta^2 \beta^2 - 2\eta\alpha\right) \|\boldsymbol{x} - \boldsymbol{y}\|^2
\end{aligned}
\tag{57}
$$

where $(b)$ comes from the smoothness of $f$ and (56). Since $\alpha > 0$, by introducing $\eta \leq \alpha/\beta^2$ into (57), we can complete our proof. $\qquad\square$

**Lemma C.10.** *Let $f$ be $\beta$-smooth and $\boldsymbol{x}^* = \arg\min f(\boldsymbol{x})$, then for any input $\boldsymbol{x}$ in the domain of $f$, the following holds*

$$
\|\nabla f(\boldsymbol{x})\|^2 \leq 2\beta \left(f(\boldsymbol{x}) - f(\boldsymbol{x}^*)\right)
$$

*Proof.* Since $f$ is $\beta$-smooth, we have the following inequality for any $x, y$ in the domain of $f$

$$
f(\boldsymbol{y}) \leq f(\boldsymbol{x}) + \nabla f(\boldsymbol{x})^\top (\boldsymbol{y} - \boldsymbol{x}) + \frac{\beta}{2} \|\boldsymbol{y} - \boldsymbol{x}\|^2 \ .
\tag{58}
$$

By setting $\boldsymbol{y} = \boldsymbol{x} - \nabla f(\boldsymbol{x})/\beta$, we have

$$
\begin{aligned}
f(\boldsymbol{x}^*) &\leq f(\boldsymbol{x} - \frac{1}{\beta} \nabla f(\boldsymbol{x})) \\
&\leq f(\boldsymbol{x}) + \nabla f(\boldsymbol{x})^\top \left(\boldsymbol{x} - \frac{1}{\beta} \nabla f(\boldsymbol{x}) - \boldsymbol{x}\right) + \frac{\beta}{2} \left\|\boldsymbol{x} - \frac{1}{\beta} \nabla f(\boldsymbol{x}) - \boldsymbol{x}\right\|^2 \\
&= f(\boldsymbol{x}) - \frac{1}{2\beta} \|\nabla f(\boldsymbol{x})\|^2 \ .
\end{aligned}
\tag{59}
$$

We finally conclude our proof by rearranging the inequality above. $\qquad\square$

We then bound the drift between $\boldsymbol{x}_{r,t}^{(i)}$ and $\boldsymbol{x}_r$ for every iteration $t$ of any round $r$ as below, which is the key difference between the convergence of general federated ZOO and centralized optimization.

**Lemma C.11.** *Assume that $F$ is $\beta$-smooth. Then the updated input $\boldsymbol{x}_{r,t}^{(i)}$ at any iteration $t \geq 1$ of round $r \geq 1$ on client $i$ in Algo. 1 has the following bounded drift with $\eta \leq \frac{1}{\beta T}$*

$$
\left\|\boldsymbol{x}_{r+1,t}^{(i)} - \boldsymbol{x}_r\right\|^2 \leq 2\eta^2 T \sum_{\tau=1}^{t} S^{t-\tau} \Xi_{r+1,\tau}^{(i)} + 22\eta^2 T^2 \|\nabla F(\boldsymbol{x}_r)\|^2
$$

*where $S \triangleq (T+1)^2/(T(T-1))$.*

*Proof.* Since $\boldsymbol{x}_{r+1,t}^{(i)} = \boldsymbol{x}_{r+1,t-1}^{(i)} - \eta\widehat{\boldsymbol{g}}_{r+1,t-1}^{(i)}$, we have the following inequalities when $T > 1$

$$
\left\|\boldsymbol{x}_{r+1,t}^{(i)} - \boldsymbol{x}_r\right\|^2
$$

$$
\overset{(a)}{=} \left\|\boldsymbol{x}_{r+1,t-1}^{(i)} - \eta\widehat{\boldsymbol{g}}_{r+1,t-1}^{(i)} - \boldsymbol{x}_r\right\|^2
$$

$$
\overset{(b)}{=} \left\|\boldsymbol{x}_{r+1,t-1}^{(i)} - \eta\nabla F(\boldsymbol{x}_{r+1,t-1}^{(i)}) + \eta\nabla F(\boldsymbol{x}_r) - \boldsymbol{x}_r + \eta\left(\nabla F(\boldsymbol{x}_{r+1,t-1}^{(i)}) - \widehat{\boldsymbol{g}}_{r+1,t-1}^{(i)} - \nabla F(\boldsymbol{x}_r)\right)\right\|^2
$$

$$
\overset{(c)}{\le} \frac{T}{T-1}\left\|\boldsymbol{x}_{r+1,t-1}^{(i)} - \eta\nabla F(\boldsymbol{x}_{r+1,t-1}^{(i)}) + \eta\nabla F(\boldsymbol{x}_r) - \boldsymbol{x}_r\right\|^2
$$

$$
+ \eta^2 T\left\|\nabla F(\boldsymbol{x}_{r+1,t-1}^{(i)}) - \widehat{\boldsymbol{g}}_{r+1,t-1}^{(i)} - \nabla F(\boldsymbol{x}_r)\right\|^2
$$

$$
\overset{(d)}{\le} \frac{T}{T-1}\left\|\boldsymbol{x}_{r+1,t-1}^{(i)} - \eta\nabla F(\boldsymbol{x}_{r+1,t-1}^{(i)}) + \eta\nabla F(\boldsymbol{x}_r) - \boldsymbol{x}_r\right\|^2
$$

$$
+ 2\eta^2 T\left[\left\|\nabla F(\boldsymbol{x}_{r+1,t-1}^{(i)}) - \widehat{\boldsymbol{g}}_{r+1,t-1}^{(i)}\right\|^2 + \|\nabla F(\boldsymbol{x}_r)\|^2\right]
$$

$$
\tag{60}
$$

where $(c)$ and $(d)$ come from the (28) in Lemma C.5 by setting $a = 1/(T-1)$ and $a = 1$, respectively. Since $F$ is $\beta$-smooth, we can introduce Lemma C.9 into (60) to obtain the following result given the constant $S \triangleq (T+1)^2/(T(T-1))$

$$
\left\|\boldsymbol{x}_{r+1,t}^{(i)} - \boldsymbol{x}_r\right\|^2
$$

$$
\overset{(a)}{\le} \frac{T(1+\eta\beta)^2}{T-1}\left\|\boldsymbol{x}_{r+1,t-1}^{(i)} - \boldsymbol{x}_r\right\|^2 + 2\eta^2 T\left[\left\|\nabla F(\boldsymbol{x}_{r+1,t-1}^{(i)}) - \widehat{\boldsymbol{g}}_{r+1,t-1}^{(i)}\right\|^2 + \|\nabla F(\boldsymbol{x}_r)\|^2\right]
$$

$$
\overset{(b)}{=} 2\eta^2 T\sum_{\tau=0}^{t-1}\left(\frac{T(1+\eta\beta)^2}{T-1}\right)^{t-\tau-1}\left\|\nabla F(\boldsymbol{x}_{r+1,\tau}^{(i)}) - \widehat{\boldsymbol{g}}_{r+1,\tau}^{(i)}\right\|^2 + 2\eta^2 T\|\nabla F(\boldsymbol{x}_r)\|^2\sum_{\tau=0}^{t-1}\left(\frac{(1+\eta\beta)^2 T}{T-1}\right)^{\tau}
$$

$$
\overset{(c)}{\le} 2\eta^2 T\sum_{\tau=0}^{t-1}\left(\frac{(T+1)^2}{T(T-1)}\right)^{t-\tau-1}\left\|\nabla F(\boldsymbol{x}_{r+1,\tau}^{(i)}) - \widehat{\boldsymbol{g}}_{r+1,\tau}^{(i)}\right\|^2 + 2\eta^2 T\|\nabla F(\boldsymbol{x}_r)\|^2\sum_{\tau=0}^{t-1}\left(\frac{(T+1)^2}{T(T-1)}\right)^{\tau}
$$

$$
\overset{(d)}{\le} 2\eta^2 T\sum_{\tau=0}^{t-1} S^{t-\tau-1}\left\|\nabla F(\boldsymbol{x}_{r+1,\tau}^{(i)}) - \widehat{\boldsymbol{g}}_{r+1,\tau}^{(i)}\right\|^2 + 22\eta^2 T^2\|\nabla F(\boldsymbol{x}_r)\|^2
$$

$$
\overset{(e)}{=} 2\eta^2 T\sum_{\tau=1}^{t} S^{t-\tau}\Xi_{r+1,\tau}^{(i)} + 22\eta^2 T^2\|\nabla F(\boldsymbol{x}_r)\|^2
$$

$$
\tag{61}
$$

where $(b)$ comes from the summation of geometric series and $(c)$ is from the fact that $\eta \le 1/(\beta T)$. In addition, $(d)$ results from the definition of $S$ as well as the following results

$$
\sum_{\tau=0}^{t-1}\left(\frac{(T+1)^2}{T(T-1)}\right)^{\tau} \le \sum_{\tau=0}^{T-1}\left(\frac{(T+1)^2}{T(T-1)}\right)^{\tau}
$$

$$
= \frac{\left((T+1)^2/[T(T-1)]\right)^T - 1}{(T+1)^2/[T(T-1)] - 1}
$$

$$
= \frac{T(T-1)}{3T+1}\left(\left(1 + \frac{3T+1}{T(T-1)}\right)^T - 1\right) \tag{62}
$$

$$
< \frac{T(T-1)}{3T+1}\left(\exp\left(\frac{3T+1}{T}\right) - 1\right)
$$

$$
< \frac{T}{3}\left(\exp\left(\frac{7}{2}\right) - 1\right)
$$

$$
< 11T .
$$

Finally, $(e)$ results from the definition of $\Xi_{r+1,t}^{(i)} \triangleq \left\| \widehat{g}_{r+1,t-1} - \nabla F(x_{r+1,t-1}^{(i)}) \right\|^2$ in our Sec. 3.2.

$\square$

We finally present the convergence of Algo. 1 in the following theorem for the general federated ZOO framework, which then can be easily applied to prove the convergence of our FZooS in Appx. C.5 and the convergence of existing federated ZOO algorithms in Appx. D.

**Theorem C.1.** *Define* $\Xi_{r,t}^{(i)} \triangleq \sum_{t=1}^{T} \left\| \widehat{g}_{r,t-1}^{(i)} - \nabla F(x_{r,t-1}^{(i)}) \right\|^2$, $S \triangleq (T+1)^2/(T(T-1))$, *and* $x^* \triangleq \arg\min F(x)$. *Algo. 1 then has the following convergence when $F$ is under different assumptions:*

(i) *When $F$ is $\beta$-smooth and $\alpha$-strongly convex, by defining* $p_r \triangleq \frac{(1-\alpha\eta T/4)^{R-r}}{\sum_{r=0}^{R}(1-\alpha\eta T/4)^{R-r}}$ *and choosing a constant learning rate* $\eta \leq \frac{1}{10\beta T}$,

$$\min_{r\in[R+1)} F(x_r) - F(x^*) \leq 2\alpha \exp\left(-\frac{\alpha\eta TR}{4}\right) \|x_0 - x^*\|^2$$
$$+ \sum_{r=0}^{R}\sum_{i=1}^{N}\sum_{t=1}^{T} p_r \left(\frac{\eta}{NT}\sum_{\tau=1}^{t} S^{t-\tau}\Xi_{r+1,\tau}^{(i)} + \frac{8(\eta T + 1/\alpha)}{\alpha NT}\Xi_{r+1,t}^{(i)}\right) .$$

(ii) *When $F$ is $\beta$-smooth and convex, by choosing a constant learning rate* $\eta \leq \frac{1}{10\beta T}$,

$$\min_{r\in[R+1)} F(x_r) - F(x^*) \leq \frac{2\|x_0 - x^*\|^2}{\eta RT} + \frac{1}{R}\sum_{r=0}^{R}\sum_{i=1}^{N}\sum_{t=1}^{T}\left(\frac{\eta}{NT}\sum_{\tau=1}^{t} S^{t-\tau}\Xi_{r+1,\tau}^{(i)}\right.$$
$$\left. + \frac{8\eta}{N}\Xi_{r+1,t}^{(i)} + \frac{4\sqrt{d}}{NT}\sqrt{\Xi_{r+1,t}^{(i)}}\right) .$$

(iii) *When $F$ is only $\beta$-smooth, by choosing a constant learning rate* $\eta \leq \frac{7}{100\beta T}$,

$$\min_{r\in[R+1)} \|\nabla F(x_r)\|^2 \leq \frac{13(F(x_0) - F(x^*))}{\eta RT} + \frac{13}{\eta RT}\sum_{r=0}^{R}\sum_{i=1}^{N}\sum_{t=1}^{T}\left(\frac{(0.14\eta + 1/(2\beta T))}{N}\Xi_{r+1,t}^{(i)}\right.$$
$$\left. + \frac{1.02\eta^2\beta}{N}\sum_{\tau=1}^{t} S^{t-\tau}\Xi_{r+1,\tau}^{(i)}\right) .$$

*Proof.* Recall that the global update on server in Algo. 1 is given as

$$x_{r+1} = \frac{1}{N}\sum_{i=1}^{N} x_{r+1}^{(i)} = \frac{1}{N}\sum_{i=1}^{N}\left(x_r^{(i)} - \eta\sum_{t=1}^{T}\widehat{g}_{r+1,t-1}^{(i)}\right) = x_r - \frac{\eta}{N}\sum_{i=1}^{N}\sum_{t=1}^{T}\widehat{g}_{r+1,t-1}^{(i)} . \quad (63)$$

Therefore, we have

$$\|x_{r+1} - x^*\|^2 = \left\| x_r - \frac{\eta}{N}\sum_{i=1}^{N}\sum_{t=1}^{T}\widehat{g}_{r+1,t-1}^{(i)} - x^* \right\|^2$$
$$= \|x_r - x^*\|^2 \underbrace{-2(x_r - x^*)^\top \frac{\eta}{N}\sum_{i=1}^{N}\sum_{t=1}^{T}\widehat{g}_{r+1,t-1}^{(i)}}_{①} + \underbrace{\left\|\frac{\eta}{N}\sum_{i=1}^{N}\sum_{t=1}^{T}\widehat{g}_{r+1,t-1}^{(i)}\right\|^2}_{②} .$$
$$(64)$$

We then bound ① and ② based on the different assumptions on $F$ separately.

**Strongly Convex $F$.** Since $F$ is $\beta$-smooth and $\alpha$-strongly convex, we have

$$
\textcircled{1} \overset{(a)}{=} 2\left(\boldsymbol{x}^* - \boldsymbol{x}_r\right)^\top \frac{\eta}{N} \sum_{i=1}^{N} \sum_{t=1}^{T} \left(\widehat{\boldsymbol{g}}_{r+1,t-1}^{(i)} - \nabla F(\boldsymbol{x}_{r+1,t-1}^{(i)})\right) + 2\left(\boldsymbol{x}^* - \boldsymbol{x}_r\right)^\top \frac{\eta}{N} \sum_{i=1}^{N} \sum_{t=1}^{T} \nabla F(\boldsymbol{x}_{r+1,t-1}^{(i)})
$$

$$
\overset{(b)}{\leq} 2\left\|\boldsymbol{x}^* - \boldsymbol{x}_r\right\| \frac{\eta}{N} \sum_{i=1}^{N} \sum_{t=1}^{T} \left\|\widehat{\boldsymbol{g}}_{r+1,t-1}^{(i)} - \nabla F(\boldsymbol{x}_{r+1,t-1}^{(i)})\right\|
$$

$$
+ \frac{2\eta}{N} \sum_{i=1}^{N} \sum_{t=1}^{T} \left[F(\boldsymbol{x}^*) - F(\boldsymbol{x}_r) - \frac{\alpha}{4}\left\|\boldsymbol{x}_r - \boldsymbol{x}^*\right\|^2 + \beta \left\|\boldsymbol{x}_{r,t-1}^{(i)} - \boldsymbol{x}_r\right\|^2\right]
$$

$$
\overset{(c)}{\leq} \frac{2\eta}{N} \left\|\boldsymbol{x}^* - \boldsymbol{x}_r\right\| \sum_{i=1}^{N} \sum_{t=1}^{T} \sqrt{\Xi_{r+1,t}^{(i)}} + 2\eta T\left[F(\boldsymbol{x}^*) - F(\boldsymbol{x}_r)\right] - \frac{\alpha \eta T}{2}\left\|\boldsymbol{x}_r - \boldsymbol{x}^*\right\|^2
$$

$$
+ \frac{4\eta^3 T\beta}{N} \sum_{i=1}^{N} \sum_{t=1}^{T} \sum_{\tau=1}^{t} S^{t-\tau} \Xi_{r+1,\tau}^{(i)} + 44\eta^3 T^3 \beta \left\|\nabla F(\boldsymbol{x}_r)\right\|^2
$$

$$
\overset{(d)}{\leq} -\frac{\alpha \eta T}{4}\left\|\boldsymbol{x}^* - \boldsymbol{x}_r\right\|^2 + 2\eta T\left[F(\boldsymbol{x}^*) - F(\boldsymbol{x}_r)\right] + 44\eta^3 T^3 \beta \left\|\nabla F(\boldsymbol{x}_r)\right\|^2 +
$$

$$
\sum_{i=1}^{N} \sum_{t=1}^{T} \left(\frac{4\eta^3 T\beta}{N} \sum_{\tau=1}^{t} S^{t-\tau} \Xi_{r+1,\tau}^{(i)} + \frac{4\eta}{\alpha N} \Xi_{r+1,t}^{(i)}\right) .
$$

$$(65)$$

where $(b)$ is from Lemma C.8 by setting $\boldsymbol{y} = \boldsymbol{x}^*$, $\boldsymbol{z} = \boldsymbol{x}_r$ and $\boldsymbol{x} = \boldsymbol{x}_{r,t-1}^{(i)}$ in Lemma C.8. In addition, $(c)$ comes from the definition of $\Xi_{r+1,t}^{(i)} \triangleq \left\|\widehat{\boldsymbol{g}}_{r+1,t-1} - \nabla F(\boldsymbol{x}_{r+1,t-1}^{(i)})\right\|^2$ in our Sec. 3.2 and Lemma C.11. Finally, $(d)$ comes from the following results

$$
\frac{2\eta}{N} \left\|\boldsymbol{x}^* - \boldsymbol{x}_r\right\| \sum_{i=1}^{N} \sum_{t=1}^{T} \sqrt{\Xi_{r+1,t}^{(i)}} = \frac{2\eta}{N} \sum_{i=1}^{N} \sum_{t=1}^{T} \left\|\boldsymbol{x}^* - \boldsymbol{x}_r\right\| \sqrt{\Xi_{r+1,t}^{(i)}}
$$

$$
\leq \frac{\eta}{N} \sum_{i=1}^{N} \sum_{t=1}^{T} \left(\frac{\alpha}{4}\left\|\boldsymbol{x}^* - \boldsymbol{x}_r\right\|^2 + \frac{4}{\alpha} \Xi_{r+1,t}^{(i)}\right) \qquad (66)
$$

$$
= \frac{\alpha \eta T}{4}\left\|\boldsymbol{x}^* - \boldsymbol{x}_r\right\|^2 + \frac{4\eta}{\alpha N} \sum_{i=1}^{N} \sum_{t=1}^{T} \Xi_{r+1,t}^{(i)} .
$$

We then bound term $\textcircled{2}$ in (64) as below

$$
\textcircled{2} \overset{(a)}{=} \left\|\frac{\eta}{N} \sum_{i=1}^{N} \sum_{t=1}^{T} \widehat{\boldsymbol{g}}_{r+1,t-1}^{(i)}\right\|^2
$$

$$
\overset{(b)}{=} \left\|\frac{\eta}{N} \sum_{i=1}^{N} \sum_{t=1}^{T} \left(\widehat{\boldsymbol{g}}_{r+1,t-1}^{(i)} - \nabla F(\boldsymbol{x}_{r+1,t-1}^{(i)}) + \nabla F(\boldsymbol{x}_{r+1,t-1}^{(i)}) - \nabla F(\boldsymbol{x}_r)\right) + \eta T\nabla F(\boldsymbol{x}_r)\right\|^2
$$

$$
\overset{(c)}{\leq} \frac{2\eta^2 T}{N} \sum_{i=1}^{N} \sum_{t=1}^{T} \left(2\left\|\widehat{\boldsymbol{g}}_{r+1,t-1}^{(i)} - \nabla F(\boldsymbol{x}_{r+1,t-1}^{(i)})\right\|^2 + 2\left\|\nabla F(\boldsymbol{x}_{r+1,t-1}^{(i)}) - \nabla F(\boldsymbol{x}_r)\right\|^2\right) +
$$

$$
2\eta^2 T^2 \left\|\nabla F(\boldsymbol{x}_r)\right\|^2
$$

$$
\overset{(d)}{\leq} \frac{4\eta^2 T}{N} \sum_{i=1}^{N} \sum_{t=1}^{T} \Xi_{r+1,t}^{(i)} + \frac{4\eta^2 T\beta^2}{N} \sum_{i=1}^{N} \sum_{t=1}^{T} \left\|\boldsymbol{x}_{r+1,t-1}^{(i)} - \boldsymbol{x}_r\right\|^2 + 2\eta^2 T^2 \left\|\nabla F(\boldsymbol{x}_r)\right\|^2
$$

$$
\overset{(e)}{\leq} \sum_{i=1}^{N} \sum_{t=1}^{T} \left(\frac{8\eta^4 T^2 \beta^2}{N} \sum_{\tau=1}^{t} S^{t-\tau} \Xi_{r+1,\tau}^{(i)} + \frac{4\eta^2 T}{N} \Xi_{r+1,t}^{(i)}\right) + \left(88\eta^4 T^4 \beta^2 + 2\eta^2 T^2\right) \left\|\nabla F(\boldsymbol{x}_r)\right\|^2
$$

$$(67)$$

where $(c)$ is obtained by applying Lemma C.5 multiple times and $(d)$ is from the smoothness of $F$. Besides, $(e)$ comes from our Lemma C.11 and the fact that $\eta \leq 1/(\beta T)$.

By combining (65) and (67), we have

$$
\begin{aligned}
& \|\boldsymbol{x}_{R+1} - \boldsymbol{x}^*\|^2 \\
& \overset{(a)}{\leq} \left(1 - \frac{\alpha\eta T}{4}\right) \|\boldsymbol{x}_R - \boldsymbol{x}^*\|^2 + 2\eta T\big[F(\boldsymbol{x}^*) - F(\boldsymbol{x}_R)\big] \\
& \qquad + 2\eta^2 T^2 \left(44\eta^2 T^2 \beta^2 + 22\eta T\beta + 1\right) \|\nabla F(\boldsymbol{x}_R)\|^2 \\
& \qquad + \sum_{i=1}^{N} \sum_{t=1}^{T} \left(\frac{4\eta^3 T\beta(2\eta T\beta + 1)}{N} \sum_{\tau=1}^{t} S^{t-\tau} \Xi_{R+1,\tau}^{(i)} + \frac{4\eta(\eta T + 1/\alpha)}{\alpha N} \Xi_{R+1,t}^{(i)}\right) \\
& \overset{(b)}{\leq} \left(1 - \frac{\alpha\eta T}{4}\right) \|\boldsymbol{x}_R - \boldsymbol{x}^*\|^2 + 2\eta T \left(1 - 2\eta T\beta \left(44\eta^2 T^2 \beta^2 + 22\eta T\beta + 1\right)\right) \big[F(\boldsymbol{x}^*) - F(\boldsymbol{x}_R)\big] \\
& \qquad + \sum_{i=1}^{N} \sum_{t=1}^{T} \left(\frac{4\eta^3 T\beta(2\eta T\beta + 1)}{N} \sum_{\tau=1}^{t} S^{t-\tau} \Xi_{r+1,\tau}^{(i)} + \frac{4\eta(\eta T + 1/\alpha)}{\alpha N} \Xi_{r+1,t}^{(i)}\right) \\
& \overset{(c)}{=} \left(1 - \frac{\alpha\eta T}{4}\right)^{R+1} \|\boldsymbol{x}_0 - \boldsymbol{x}^*\|^2 + \sum_{r=0}^{R} \left(1 - \frac{\alpha\eta T}{4}\right)^{R-r} H\big[F(\boldsymbol{x}^*) - F(\boldsymbol{x}_r)\big] \\
& \qquad + \sum_{r=0}^{R} \left(1 - \frac{\alpha\eta T}{4}\right)^{R-r} \sum_{i=1}^{N} \sum_{t=1}^{T} \left(\frac{4\eta^3 T\beta(2\eta T\beta + 1)}{N} \sum_{\tau=1}^{t} S^{t-\tau} \Xi_{r+1,\tau}^{(i)} + \frac{4\eta(\eta T + 1/\alpha)}{\alpha N} \Xi_{r+1,t}^{(i)}\right)
\end{aligned}
$$
$$(68)$$

where $(b)$ is from Lemma C.10 and $(c)$ is from $H \triangleq 2\eta T \left(1 - 2\eta T\beta \left(44\eta^2 T^2 \beta^2 + 22\eta T\beta + 1\right)\right)$ as well as the repeated application of $(b)$.

Define $p_r \triangleq \frac{(1-\alpha\eta T/4)^{R-r}}{\sum_{r=0}^{R}(1-\alpha\eta T/4)^{R-r}}$. Note that when choose the learning rate $\eta$ that satisfies $\eta \leq \frac{1}{10\beta T}$, we have $H \geq 0.544\,\eta T$. Based on this and $\|\boldsymbol{x}_{R+1} - \boldsymbol{x}^*\|^2 \geq 0$ for (68), we further have

$$
\begin{aligned}
\min_{r \in [R+1)} F(\boldsymbol{x}_r) - F(\boldsymbol{x}^*) & \overset{(a)}{\leq} \sum_{r=0}^{R} p_r \big[F(\boldsymbol{x}_r) - F(\boldsymbol{x}^*)\big] \\
& \overset{(b)}{\leq} \frac{(1 - \alpha\eta T/4)^{R+1} \|\boldsymbol{x}_0 - \boldsymbol{x}^*\|^2}{H \sum_{r=0}^{R} (1 - \alpha\eta T/4)^r} \\
& \qquad + \frac{1}{H} \sum_{r=0}^{R} \sum_{i=1}^{N} \sum_{t=1}^{T} p_r \left(\frac{\eta^2}{2N} \sum_{\tau=1}^{t} S^{t-\tau} \Xi_{r+1,\tau}^{(i)} + \frac{4\eta(\eta T + 1/\alpha)}{\alpha N} \Xi_{r+1,t}^{(i)}\right) \\
& \overset{(c)}{\leq} \frac{\alpha\eta T}{H} \exp\left(-\frac{\alpha\eta T R}{4}\right) \|\boldsymbol{x}_0 - \boldsymbol{x}^*\|^2 \\
& \qquad + \frac{1}{H} \sum_{r=0}^{R} \sum_{i=1}^{N} \sum_{t=1}^{T} p_r \left(\frac{\eta^2}{2N} \sum_{\tau=1}^{t} S^{t-\tau} \Xi_{r+1,\tau}^{(i)} + \frac{4\eta(\eta T + 1/\alpha)}{\alpha N} \Xi_{r+1,t}^{(i)}\right) \\
& \overset{(d)}{\leq} 2\alpha \exp\left(-\frac{\alpha\eta T R}{4}\right) \|\boldsymbol{x}_0 - \boldsymbol{x}^*\|^2 \\
& \qquad + \sum_{r=0}^{R} \sum_{i=1}^{N} \sum_{t=1}^{T} p_r \left(\frac{\eta}{NT} \sum_{\tau=1}^{t} S^{t-\tau} \Xi_{r+1,\tau}^{(i)} + \frac{8(\eta T + 1/\alpha)}{\alpha NT} \Xi_{r+1,t}^{(i)}\right)
\end{aligned}
$$
$$(69)$$

where $(b)$ is from the rearrangement of (68) and the fact that $\eta \leq \frac{1}{10\beta T}$. Besides, $(c)$ comes from the inequality $1 - x \leq \exp(-x)$ as well as the following results when $R + 1 \geq 4\ln(3/4)/(\alpha\eta T)$

$$\sum_{r=0}^{R} \left(1 - \frac{\alpha\eta T}{4}\right)^r = \frac{1 - (1 - \alpha\eta T/4)^{R+1}}{1 - (1 - \alpha\eta T/4)}$$
$$\geq \frac{4\left[1 - \exp(-\alpha\eta T(R+1)/4)\right]}{\alpha\eta T} \tag{70}$$
$$\geq \frac{1}{\alpha\eta T} .$$

Finally, $(d)$ is due to the fact that $H \geq 0.544\,\eta T$.

**Convex $F$.** When $\alpha = 0$, following the derivation in (65), we have

$$① \overset{(a)}{\leq} \frac{2\eta}{N} \|\boldsymbol{x}^* - \boldsymbol{x}_r\| \sum_{i=1}^{N}\sum_{t=1}^{T} \sqrt{\Xi_{r+1,t}^{(i)}} + 2\eta T\left[F(\boldsymbol{x}^*) - F(\boldsymbol{x}_r)\right] + 44\eta^3 T^3 \beta \left\|\nabla F(\boldsymbol{x}_r)\right\|^2$$
$$+ \frac{4\eta^3 T \beta}{N} \sum_{i=1}^{N}\sum_{t=1}^{T}\sum_{\tau=1}^{t} S^{t-\tau}\Xi_{r+1,\tau}^{(i)}$$
$$\overset{(b)}{\leq} \frac{2\eta\sqrt{d}}{N} \sum_{i=1}^{N}\sum_{t=1}^{T} \sqrt{\Xi_{r+1,t}^{(i)}} + 2\eta T\left[F(\boldsymbol{x}^*) - F(\boldsymbol{x}_r)\right] + 44\eta^3 T^3 \beta \left\|\nabla F(\boldsymbol{x}_r)\right\|^2 \tag{71}$$
$$+ \frac{4\eta^3 T \beta}{N} \sum_{i=1}^{N}\sum_{t=1}^{T}\sum_{\tau=1}^{t} S^{t-\tau}\Xi_{r+1,\tau}^{(i)}$$
$$\overset{(c)}{=} 2\eta T\left[F(\boldsymbol{x}^*) - F(\boldsymbol{x}_r)\right] + 44\eta^3 T^3 \beta \left\|\nabla F(\boldsymbol{x}_r)\right\|^2$$
$$+ \sum_{i=1}^{N}\sum_{t=1}^{T} \left(\frac{4\eta^3 T \beta}{N} \sum_{\tau=1}^{t} S^{t-\tau}\Xi_{r+1,\tau}^{(i)} + \frac{2\eta\sqrt{d}}{N} \sqrt{\Xi_{r+1,t}^{(i)}}\right)$$

where the $(b)$ comes from the diameter of $\mathcal{X}$, i.e., $\|\boldsymbol{x} - \boldsymbol{x}'\| \leq \sqrt{d}$ for any $\boldsymbol{x}, \boldsymbol{x}' \in \mathcal{X} = [0, 1]^d$. For term $②$ in (64), similar to (67), we also have

$$② \leq \sum_{i=1}^{N}\sum_{t=1}^{T} \left(\frac{8\eta^4 T^2 \beta^2}{N} \sum_{\tau=1}^{t} S^{t-\tau}\Xi_{r+1,\tau}^{(i)} + \frac{4\eta^2 T}{N}\Xi_{r+1,t}^{(i)}\right) + \left(88\eta^4 T^4 \beta^2 + 2\eta^2 T^2\right)\left\|\nabla F(\boldsymbol{x}_r)\right\|^2 . \tag{72}$$

By combining (71) and (72), we have

$$\|\boldsymbol{x}_{R+1} - \boldsymbol{x}^*\|^2$$
$$\overset{(a)}{\leq} \|\boldsymbol{x}_R - \boldsymbol{x}^*\|^2 + 2\eta T \left(1 - 2\eta T\beta \left(44\eta^2 T^2 \beta^2 + 22\eta T\beta + 1\right)\right)\left[F(\boldsymbol{x}^*) - F(\boldsymbol{x}_R)\right]$$
$$+ \sum_{i=1}^{N}\sum_{t=1}^{T} \left(\frac{4\eta^3 T \beta(2\eta T\beta + 1)}{N} \sum_{\tau=1}^{t} S^{t-\tau}\Xi_{R+1,\tau}^{(i)} + \frac{4\eta^2 T}{N}\Xi_{R+1,t}^{(i)} + \frac{2\eta\sqrt{d}}{N} \sqrt{\Xi_{R+1,t}^{(i)}}\right)$$
$$\overset{(b)}{\leq} \|\boldsymbol{x}_0 - \boldsymbol{x}^*\|^2 + \sum_{r=0}^{R} H\left[F(\boldsymbol{x}^*) - F(\boldsymbol{x}_r)\right]$$
$$+ \sum_{i=1}^{N}\sum_{t=1}^{T} \left(\frac{4\eta^3 T \beta(2\eta T\beta + 1)}{N} \sum_{\tau=1}^{t} S^{t-\tau}\Xi_{r+1,\tau}^{(i)} + \frac{4\eta^2 T}{N}\Xi_{r+1,t}^{(i)} + \frac{2\eta\sqrt{d}}{N} \sqrt{\Xi_{r+1,t}^{(i)}}\right)$$
$$\tag{73}$$

where $(a)$ is from Lemma C.10 and $(b)$ is from $H \triangleq 2\eta T \left(1 - 2\eta T\beta \left(44\eta^2 T^2 \beta^2 + 22\eta T\beta + 1\right)\right)$ as well as the repeated application of $(a)$.

Note that when choose the learning rate $\eta$ that satisfies $\eta \leq \frac{1}{10\beta T}$, we have $H \geq 0.544\,\eta T$. Based on this and $\|\boldsymbol{x}_{R+1} - \boldsymbol{x}^*\|^2 \geq 0$ for (73), we further have

$$
\begin{aligned}
\min_{r \in [R+1)} F(\boldsymbol{x}_r) - F(\boldsymbol{x}^*) &\overset{(a)}{\leq} \frac{1}{R} \sum_{r=0}^{R} \left[ F(\boldsymbol{x}_r) - F(\boldsymbol{x}^*) \right] \\
&\overset{(b)}{\leq} \frac{\|\boldsymbol{x}_0 - \boldsymbol{x}^*\|^2}{RH} + \frac{1}{RH} \sum_{r=0}^{R} \sum_{i=1}^{N} \sum_{t=1}^{T} \left( \frac{\eta^2}{2N} \sum_{\tau=1}^{t} S^{t-\tau} \Xi_{r+1,\tau}^{(i)} \right. \\
&\qquad \left. + \frac{4\eta^2 T}{N} \Xi_{r+1,t}^{(i)} + \frac{2\eta\sqrt{d}}{N} \sqrt{\Xi_{r+1,t}^{(i)}} \right) \\
&\overset{(c)}{\leq} \frac{2\|\boldsymbol{x}_0 - \boldsymbol{x}^*\|^2}{\eta R} + \frac{1}{R} \sum_{r=0}^{R} \sum_{i=1}^{N} \sum_{t=1}^{T} \left( \frac{\eta}{NT} \sum_{\tau=1}^{t} S^{t-\tau} \Xi_{r+1,\tau}^{(i)} \right. \\
&\qquad \left. + \frac{8\eta}{N} \Xi_{r+1,t}^{(i)} + \frac{4\sqrt{d}}{NT} \sqrt{\Xi_{r+1,t}^{(i)}} \right)
\end{aligned}
\tag{74}
$$

where $(c)$ is due to the fact that $H \geq 0.544\,\eta T$.

**Non-Convex $F$.** When $F$ is only $\beta$-smooth, we have

$$
\begin{aligned}
&F(\boldsymbol{x}_{r+1}) - F(\boldsymbol{x}_r) \\
&\overset{(a)}{\leq} \nabla F(\boldsymbol{x}_r)^\top (\boldsymbol{x}_{r+1} - \boldsymbol{x}_r) + \frac{\beta}{2} \|\boldsymbol{x}_{r+1} - \boldsymbol{x}_r\|^2 \\
&\overset{(b)}{=} -\frac{\eta}{N} \nabla F(\boldsymbol{x}_r)^\top \sum_{i=1}^{N} \sum_{t=1}^{T} \widehat{\boldsymbol{g}}_{r+1,t-1}^{(i)} + \frac{\beta}{2} \left\| \frac{\eta}{N} \sum_{i=1}^{N} \sum_{t=1}^{T} \widehat{\boldsymbol{g}}_{r+1,t-1}^{(i)} \right\|^2 \\
&\overset{(c)}{\leq} -\frac{\eta}{N} \nabla F(\boldsymbol{x}_r)^\top \sum_{i=1}^{N} \sum_{t=1}^{T} \left( \widehat{\boldsymbol{g}}_{r+1,t-1}^{(i)} - \nabla F(\boldsymbol{x}_{r+1,t-1}^{(i)}) + \nabla F(\boldsymbol{x}_{r+1,t-1}^{(i)}) - \nabla F(\boldsymbol{x}_r) + \nabla F(\boldsymbol{x}_r) \right) \\
&\qquad + \frac{\beta}{2} \left[ \sum_{i=1}^{N} \sum_{t=1}^{T} \left( \frac{8\eta^4 T^2 \beta^2}{N} \sum_{\tau=1}^{t} S^{t-\tau} \Xi_{r+1,\tau}^{(i)} + \frac{4\eta^2 T}{N} \Xi_{r+1,t}^{(i)} \right) + \left( 88\eta^4 T^4 \beta^2 + 2\eta^2 T^2 \right) \|\nabla F(\boldsymbol{x}_r)\|^2 \right] \\
&\overset{(d)}{\leq} \frac{\eta}{N} \sum_{i=1}^{N} \sum_{t=1}^{T} \|\nabla F(\boldsymbol{x}_r)\| \left( \left\| \widehat{\boldsymbol{g}}_{r+1,t-1}^{(i)} - \nabla F(\boldsymbol{x}_{r+1,t-1}^{(i)}) \right\| + \left\| \nabla F(\boldsymbol{x}_{r+1,t-1}^{(i)}) - \nabla F(\boldsymbol{x}_r) \right\| \right) \\
&\qquad + \sum_{i=1}^{N} \sum_{t=1}^{T} \left( \frac{4\eta^4 T^2 \beta^3}{N} \sum_{\tau=1}^{t} S^{t-\tau} \Xi_{r+1,\tau}^{(i)} + \frac{2\eta^2 \beta T}{N} \Xi_{r+1,t}^{(i)} \right) + \left( 44\eta^4 T^4 \beta^3 + \eta^2 T^2 \beta - \eta T \right) \|\nabla F(\boldsymbol{x}_r)\|^2 \\
&\overset{(e)}{\leq} \frac{\eta}{N} \sum_{i=1}^{N} \sum_{t=1}^{T} \left( \eta\beta T \|\nabla F(\boldsymbol{x}_r)\|^2 + \frac{1}{2\eta\beta T} \left\| \widehat{\boldsymbol{g}}_{r+1,t-1}^{(i)} - \nabla F(\boldsymbol{x}_{r+1,t-1}^{(i)}) \right\|^2 + \frac{\beta}{2\eta T} \left\| \boldsymbol{x}_{r+1,t-1}^{(i)} - \boldsymbol{x}_r \right\|^2 \right) + \\
&\qquad + \sum_{i=1}^{N} \sum_{t=1}^{T} \left( \frac{4\eta^4 T^2 \beta^3}{N} \sum_{\tau=1}^{t} S^{t-\tau} \Xi_{r+1,\tau}^{(i)} + \frac{2\eta^2 \beta T}{N} \Xi_{r+1,t}^{(i)} \right) + \left( 44\eta^4 T^4 \beta^3 + \eta^2 T^2 \beta - \eta T \right) \|\nabla F(\boldsymbol{x}_r)\|^2 \\
&\overset{(f)}{\leq} \left( 44\eta^4 T^4 \beta^3 + 13\eta^2 T^2 \beta - \eta T \right) \|\nabla F(\boldsymbol{x}_r)\|^2 + \sum_{i=1}^{N} \sum_{t=1}^{T} \left( \frac{\left(4\eta^4 T^2 \beta^3 + \eta^2 \beta\right)}{N} \sum_{\tau=1}^{t} S^{t-\tau} \Xi_{r+1,\tau}^{(i)} \right. \\
&\qquad \left. + \frac{\left(2\eta^2 \beta T + 1/(2\beta T)\right)}{N} \Xi_{r+1,t}^{(i)} \right)
\end{aligned}
\tag{75}
$$

where $(a)$ comes from the smoothness of $F$ and $(b)$ is from the one-round update (63) for input $\boldsymbol{x}$. In addition, $(c)$ derives from (67) and $(e)$ results from (27) in Lemma C.5 by setting $a = \eta\beta T$ in (27). Finally, $(f)$ comes from Lemma C.11.

Define $H \triangleq \eta T - 44\eta^4 T^4 \beta^3 - 13\eta^2 T^2 \beta$ and choose $\eta \leq \frac{7}{100\beta T}$, we have that $H > 0.08\eta T$. Based on this, we further have

$$
\begin{aligned}
\min_{r \in [R+1]} \|\nabla F(\boldsymbol{x}_r)\|^2 &\overset{(a)}{\leq} \frac{1}{R} \sum_{r=0}^{R} \|\nabla F(\boldsymbol{x}_r)\|^2 \\
&\overset{(b)}{\leq} \frac{1}{RH} \sum_{r=0}^{R} [F(\boldsymbol{x}_r) - F(\boldsymbol{x}_{r+1})] + \frac{1}{RH} \sum_{r=0}^{R} \sum_{i=1}^{N} \sum_{t=1}^{T} \left( \frac{(2\eta^2 \beta T + 1/(2\beta T))}{N} \Xi_{r+1,t}^{(i)} \right. \\
&\quad + \left. \frac{(4\eta^4 T^2 \beta^3 + \eta^2 \beta)}{N} \sum_{\tau=1}^{t} S^{t-\tau} \Xi_{r+1,\tau}^{(i)} \right) \\
&\overset{(c)}{\leq} \frac{13(F(\boldsymbol{x}_0) - F(\boldsymbol{x}^*))}{\eta RT} + \frac{13}{\eta RT} \sum_{r=0}^{R} \sum_{i=1}^{N} \sum_{t=1}^{T} \left( \frac{(0.14\eta + 1/(2\beta T))}{N} \Xi_{r+1,t}^{(i)} \right. \\
&\quad + \left. \frac{1.02\eta^2 \beta}{N} \sum_{\tau=1}^{t} S^{t-\tau} \Xi_{r+1,\tau}^{(i)} \right)
\end{aligned}
$$

$$(76)$$

where $(c)$ is due to the fact that $H \geq 0.08\,\eta T$. $\square$

**Remark.** Of note, Thm. C.1 has presented the convergence of the general optimization framework for federated ZOO problems (i.e., Algo. 1). So, it can be easily adapted to provide the convergence for those algorithms that follow this optimization framework (e.g., our Thm. 2 and the results in Appx. D). This advancement demonstrates superiority over existing federated optimization approaches, such as FedZO, FedProx, and SCAFFOLD, in terms of universality. Notably, these prior works primarily focus on providing convergence guarantees exclusively for their specific algorithmic designs.

## C.5 Proof of Theorem 2

To establish the proof for Thm. 2, we introduce the upper bound of gradient disparity $\frac{1}{N}\sum_{i=1}^{N}\Xi_{r,t}^{(i)}$ derived from our Thm. 1, into Thm. C.1. This is in fact facilitated by leveraging the gradient correction length in our Cor. 1 to improve the bound in our Thm. 1 (refer to the remark of Appx. C.2). To begin with, we first derive a set of inequalities below based on our (38) since they are frequently required in the results of Thm. C.1. It is important to note that for the sake of simplicity in our proof, we present the validity of these inequalities with a constant probability, without explicitly providing the exact form of this probability.

$$
\begin{aligned}
& \frac{1}{NR}\sum_{r=0}^{R}\sum_{t=1}^{T}\sum_{i=1}^{N}\sum_{\tau=1}^{t}S^{t-\tau}\Xi_{r+1,\tau}^{(i)} \\
& \overset{(a)}{=} \frac{1}{R}\sum_{r=0}^{R}\sum_{t=1}^{T}\sum_{\tau=1}^{t}S^{t-\tau}\left(4\omega\kappa\rho^{rT+\tau-1}+2\sqrt{2\omega\kappa\rho^{rT}G}+2\sqrt{2NG}\epsilon\right) \\
& \overset{(b)}{=} \sum_{t=1}^{T}\frac{1}{R}\sum_{r=0}^{R}\left(\frac{4\omega\kappa\rho^{rT}\left(S^{t}-\rho^{t}\right)}{S-\rho}+\left(2\sqrt{2\omega\kappa\rho^{rT}G}+2\sqrt{2NG}\epsilon\right)\frac{S^{t}-1}{S-1}\right) \\
& \overset{(c)}{=} \sum_{t=1}^{T}\left[\frac{4\omega\kappa\left(S^{t}-\rho^{t}\right)\left(1-\rho^{(R+1)T}\right)}{R(S-\rho)(1-\rho^{T})}+\left(\frac{2\sqrt{2\omega\kappa G}(1-\rho^{(R+1)T/2})}{R(1-\rho^{T/2})(S-1)}+\frac{2\sqrt{2NG}\epsilon}{S-1}\right)\left(S^{t}-1\right)\right] \\
& \overset{(d)}{=} \frac{4\omega\kappa(1-\rho^{(R+1)T})}{R(S-\rho)(1-\rho^{T})}\left(\frac{S(S^{T}-1)}{S-1}-\frac{\rho(1-\rho^{T})}{1-\rho}\right)+\left(\frac{2\sqrt{2\omega\kappa G}(1-\rho^{(R+1)T/2})}{R(1-\rho^{T/2})(S-1)}\right. \\
& \qquad \left.+\frac{2\sqrt{2NG}\epsilon}{S-1}\right)\left(\frac{S(S^{T}-1)}{S-1}-1\right) \\
& \overset{(e)}{=} \mathcal{O}\left(\frac{T^{2}(\sqrt{G}+1)}{R}+T^{2}\sqrt{\frac{NG}{M}}\right)
\end{aligned}
$$

(77)

where $(b),(c),(d)$ are from the summation of geometric series. In addition, $(e)$ comes from the fact that $S\triangleq\frac{(T+1)^{2}}{T(T-1)}$ (i.e., $S\leq 4.5$), $\frac{S^{T}-1}{S-1}\leq 11T$ in (62), $\frac{S}{S-1}=\frac{(T+1)^{2}}{3T+1}=\mathcal{O}(T)$ and $\epsilon=\mathcal{O}\left(\frac{1}{M}\right)$.

$$
\begin{aligned}
\frac{1}{NR}\sum_{r=0}^{R}\sum_{t=1}^{T}\sum_{i=1}^{N}\Xi_{r+1,t}^{(i)} & \overset{(a)}{=} \frac{1}{R}\sum_{r=0}^{R}\sum_{t=1}^{T}\left(4\omega\kappa\rho^{rT+t-1}+2\sqrt{2\omega\kappa\rho^{rT}G}+2\sqrt{2NG}\epsilon\right) \\
& \overset{(b)}{=} \frac{1}{R}\sum_{r=0}^{R}\left(\frac{4\omega\kappa\rho^{rT}(1-\rho^{T})}{1-\rho}+2T\sqrt{2\omega\kappa\rho^{rT}G}+2T\sqrt{2NG}\epsilon\right) \\
& \overset{(c)}{=} \frac{4\omega\kappa(1-\rho^{(R+1)T})}{R(1-\rho)}+\frac{2T\sqrt{2\omega\kappa G}(1-\rho^{(R+1)T/2})}{R(1-\rho^{T/2})}+2T\sqrt{2NG}\epsilon \\
& \overset{(d)}{=} \mathcal{O}\left(\frac{T\sqrt{G}+1}{R}+T\sqrt{NG}\epsilon\right)
\end{aligned}
$$

(78)

where $(c),(d)$ are from the summation of geometric series.

$$\frac{1}{NR}\sum_{r=0}^{R}\sum_{t=1}^{T}\sum_{i=1}^{N}\sqrt{\Xi_{r+1,t}^{(i)}} \overset{(a)}{\le} \frac{1}{R}\sum_{r=0}^{R}\sum_{t=1}^{T}\sqrt{\frac{1}{N}\sum_{i=1}^{N}\Xi_{r+1,t}^{(i)}}$$

$$\overset{(b)}{\le} \frac{1}{R}\sum_{r=1}^{R}\sum_{t=1}^{T}\left(\sqrt{4\omega\kappa\rho^{rT+t-1}}+\sqrt{2\sqrt{2\omega\kappa\rho^{rT}G}}+\sqrt{2\sqrt{2NG\epsilon}}\right)$$

$$\overset{(c)}{=} \frac{1}{R}\sum_{r=0}^{R}\left(\frac{\sqrt{4\omega\kappa\rho^{rT}}(1-\rho^{T/2})}{1-\rho^{1/2}}+T\sqrt{2\sqrt{2\omega\kappa\rho^{rT}G}}+T\sqrt{2\sqrt{2NG\epsilon}}\right)$$

$$\overset{(d)}{=} \frac{\sqrt{4\omega\kappa}(1-\rho^{T/2})(1-\rho^{(R+1)T/2})}{R(1-\rho^{1/2})(1-\rho^{T/2})}+\frac{T\sqrt[4]{8\omega\kappa G}(1-\rho^{(R+1)T/4})}{R(1-\rho^{T/4})}+T\sqrt[4]{8NG\epsilon}$$

$$\overset{(e)}{=} \mathcal{O}\left(\frac{T\sqrt[4]{G}+1}{R}+T\sqrt[4]{\frac{NG}{M}}\right)$$

$$(79)$$

where $(a)$ is from Cauchy–Schwarz inequality and $(b)$ is from the inequality of $\sum_j c_j \le \left(\sum_j \sqrt{c_j}\right)^2$ for any $c_j > 0$. Besides, $(c), (d)$ are from the summation of geometric series.

Subsequently, we proceed to establish the proof for the results in Thm. 2 that are conditioned on different assumptions of $F$ by systematically demonstrating each case individually as follows.

**Strongly Convex $F$.** Define $c \triangleq 1 - \alpha\eta T/4$. [4] When $R+1 \ge 4\ln(3/4)/(\alpha\eta T)$, we then have that $p_r \le \alpha\eta T c^{R-r}$ according to (70), which finally yields the following result

$$\frac{1}{N}\sum_{r=1}^{R}p_r\sum_{t=1}^{T}\sum_{i=1}^{N}\sum_{\tau=1}^{t}S^{t-\tau}\Xi_{r,\tau}^{(i)}$$

$$\overset{(a)}{=} \sum_{r=1}^{R}\frac{4p_r\omega\kappa\rho^{rT}}{S-\rho}\left(\frac{S(S^T-1)}{S-1}-\frac{\rho(1-\rho^T)}{1-\rho}\right)+\sum_{r=1}^{R}\frac{2p_r\sqrt{2\omega\kappa G}\rho^{rT/2}}{S-1}\left(\frac{S(S^T-1)}{S-1}-1\right)$$

$$+\frac{2\sqrt{2NG\epsilon}}{S-1}\left(\frac{S(S^T-1)}{S-1}-1\right)$$

$$\overset{(b)}{\le} \frac{4\alpha\eta T\omega\kappa(c^{R+1}-\rho^{(R+1)T})}{(S-\rho)(c-\rho^T)}\left(\frac{S(S^T-1)}{S-1}-\frac{\rho(1-\rho^T)}{1-\rho}\right)$$

$$+\frac{2\alpha\eta T\sqrt{2\omega\kappa G}(c^{R+1}-\rho^{(R+1)T/2})}{(S-1)(c-\rho^{T/2})}\left(\frac{S(S^T-1)}{S-1}-1\right)+\frac{2\sqrt{2NG\epsilon}}{S-1}\left(\frac{S(S^T-1)}{S-1}-1\right)$$

$$\overset{(c)}{\le} \mathcal{O}\left(\alpha\eta T^3 c^R(\sqrt{G}+1)+T^2\sqrt{\frac{NG}{M}}\right)$$

$$\overset{(d)}{=} \mathcal{O}\left(\frac{\alpha T^2 c^R}{\beta}(\sqrt{G}+1)+T^2\sqrt{\frac{NG}{M}}\right)$$

$$(80)$$

where $(a)$ follows from the derivation in (77) and $(b)$ is due to the fact that $p_r \le \alpha\eta T c^{R-r}$ as well as the summation of geometric series. Besides, $(c)$ comes from $c^{R+1} > \rho^{(R+1)T/2} > \rho^{(R+1)T}$ and $c > \rho^{T/2} > \rho^T$ when we choose $c$ properly in the proof of (69) as well as $\epsilon = \mathcal{O}\left(\frac{1}{M}\right)$. Finally, $(d)$ results from the fact that $\eta \le \frac{1}{10\beta T}$ and $\alpha < \beta$.

---

[4] Note that according to (66), we can always find a $\sqrt{\rho} < c < 1$ such that (69) still holds with only different constant terms. As a result, $c^{R+1} > \rho^{(R+1)T/2} > \rho^{(R+1)T}$ and $c > \rho^{T/2} > \rho^T$.

Following from the derivation above, we also have

$$
\frac{1}{N} \sum_{r=0}^{R} p_r \sum_{t=1}^{T} \sum_{i=1}^{N} \Xi_{r+1,t}^{(i)}
$$

$$
= \sum_{r=0}^{R} p_r \left( \frac{4\omega\kappa\rho^{rT}(1-\rho^T)}{1-\rho} + 2T\sqrt{2\omega\kappa\rho^{rT}G} + 2T\sqrt{2NG\epsilon} \right)
$$

$$
\leq \frac{4\alpha\eta T\omega\kappa(1-\rho^T)(c^{R+1}-\rho^{(R+1)T})}{(1-\rho)(c-\rho^T)} + \frac{2\alpha\eta T^2\sqrt{2\omega\kappa G}(c^{R+1}-\rho^{(R+1)T/2})}{(c-\rho^{T/2})} + 2T\sqrt{2NG\epsilon}
$$

$$
= \mathcal{O}\left( \frac{\alpha c^R}{\beta}(T\sqrt{G}+1) + T\sqrt{\frac{NG}{M}} \right) .
$$

$$(81)$$

Finally, by introducing (80) and (81) into Thm. C.1, we have

$$
\min_{r\in[R+1]} F(\boldsymbol{x}_r) - F(\boldsymbol{x}^*)
$$

$$
\overset{(a)}{\leq} 2\alpha\exp\left(-\frac{\alpha\eta TR}{4}\right)\|\boldsymbol{x}_0 - \boldsymbol{x}^*\|^2 + \sum_{r=0}^{R}\sum_{i=1}^{N}\sum_{t=1}^{T} p_r \left( \frac{\eta}{NT}\sum_{\tau=1}^{t} S^{t-\tau}\Xi_{r+1,\tau}^{(i)} + \frac{8(\eta T + 1/\alpha)}{\alpha NT}\Xi_{r+1,t}^{(i)} \right)
$$

$$
\overset{(b)}{\leq} \mathcal{O}\left( \alpha\exp\left(-\frac{\alpha\eta TR}{4}\right)D_0 + \frac{1}{\beta T^2}\left( \frac{\alpha c^R T^2}{\beta}(\sqrt{G}+1) + T^2\sqrt{\frac{NG}{M}} \right) \right.
$$

$$
\left. + \frac{1/\beta + 1/\alpha}{\alpha T}\left( \frac{\alpha c^R}{\beta}(T\sqrt{G}+1) + T\sqrt{\frac{NG}{M}} \right) \right)
$$

$$
\overset{(c)}{=} \mathcal{O}\left( \exp(-\eta RT)D_0 + c^R\sqrt{G} + \sqrt{\frac{NG}{M}} \right)
$$

$$(82)$$

where $(b)$ is due to the fact that $\eta \leq \frac{1}{10\beta T}$. Let each item above achieve an $\epsilon/4$ error, we then realize the result in our Thm. 2 when $F$ is $\alpha$-strongly convex and $\beta$-smooth.

**Convex $F$.** By introducing (77), (78) and (79) into Thm. C.1, we have

$$
\min_{r\in[R+1]} F(\boldsymbol{x}_r) - F(\boldsymbol{x}^*)
$$

$$
\overset{(a)}{\leq} \frac{2\|\boldsymbol{x}_0 - \boldsymbol{x}^*\|^2}{\eta RT} + \frac{1}{R}\sum_{r=0}^{R}\sum_{i=1}^{N}\sum_{t=1}^{T}\left( \frac{\eta}{NT}\sum_{\tau=1}^{t} S^{t-\tau}\Xi_{r+1,\tau}^{(i)} + \frac{8\eta}{N}\Xi_{r+1,t}^{(i)} + \frac{4\sqrt{d}}{NT}\sqrt{\Xi_{r+1,t}^{(i)}} \right)
$$

$$
\overset{(b)}{\leq} \mathcal{O}\left( \frac{D_0}{\eta RT} + \frac{1}{\beta T^2}\left( \frac{T^2(\sqrt{G}+1)}{R} + T^2\sqrt{\frac{NG}{M}} \right) + \frac{1}{\beta T}\left( \frac{T\sqrt{G}+1}{R} + T\sqrt{\frac{NG}{M}} \right) \right.
$$

$$
\left. + \frac{\sqrt{d}}{T}\left( \frac{T\sqrt[4]{G}+1}{R} + T\sqrt[4]{\frac{NG}{M}} \right) \right)
$$

$$
\overset{(c)}{=} \mathcal{O}\left( \frac{D_0}{\eta RT} + \frac{\sqrt{G}+\sqrt[4]{d^2 G}}{R} + \sqrt{\frac{NG}{M}} + \sqrt[4]{\frac{NG}{M}} \right)
$$

$$(83)$$

where $(b)$ is due to the fact that $\eta \leq \frac{1}{10\beta T}$. Let each item above achieve an $\epsilon/4$ error, we then realize the result in our Thm. 2 when $F$ is convex and $\beta$-smooth.

**Non-Convex $F$.** By introducing (77) and (78) into Thm. C.1, we have

$$\min_{r\in[R+1)} \|\nabla F(\boldsymbol{x}_r)\|^2$$

$$\overset{(a)}{\leq} \frac{13(F(\boldsymbol{x}_0) - F(\boldsymbol{x}^*))}{\eta RT} + \frac{13}{\eta RT} \sum_{r=0}^{R} \sum_{i=1}^{N} \sum_{t=1}^{T} \left( \frac{(0.14\eta + 1/(2\beta T))}{N} \Xi_{r+1,t}^{(i)} \right.$$

$$\left. + \frac{1.02\eta^2\beta}{N} \sum_{\tau=1}^{t} S^{t-\tau} \Xi_{r+1,\tau}^{(i)} \right) \tag{84}$$

$$\overset{(b)}{\leq} \mathcal{O}\left( \frac{D_1}{\eta RT} + \frac{1}{T}\left( \frac{T\sqrt{G}+1}{R} + T\sqrt{\frac{NG}{M}} \right) + \frac{1}{\beta T^2}\left( \frac{T^2(\sqrt{G}+1)}{R} + T^2\sqrt{\frac{NG}{M}} \right) \right)$$

$$\overset{(c)}{=} \mathcal{O}\left( \frac{D_1}{\eta RT} + \frac{\sqrt{G}}{R} + \sqrt{\frac{NG}{M}} \right)$$

where $(b)$ is due to the fact that $\eta \leq \frac{7}{100\beta T}$. Let each item above achieve an $\epsilon/3$ error, we then realize the result in our Thm. 2 when $F$ is non-convex and $\beta$-smooth. This hence finally concludes our proof of Thm. 2.

## APPENDIX D    THEORETICAL RESULTS FOR EXISTING FEDERATED ZOO ALGORITHMS

### D.1    GRADIENT ESTIMATION IN EXISTING FEDERATED ZOO ALGORITHMS

We first introduce the following lemma from the Thm. 2.6 in (Berahas et al., 2022) to bound the gradient estimation error of the standard FD method, which usually serves as the foundation of existing federated ZOO baselines, e.g., (Fang et al., 2022).

**Lemma D.1.** *Let $\delta \in (0,1)$. Assume that function $f$ is $\beta$-smooth in its domain and $\boldsymbol{u}_q \sim \mathcal{N}(\boldsymbol{0}, \mathbf{I})$ in (3), then the following holds with a probability of at least $1 - \delta$,*

$$\|\boldsymbol{\Delta}(\boldsymbol{x}) - \nabla f(\boldsymbol{x})\| \leq \beta \lambda \sqrt{d} + \frac{\epsilon \sqrt{d}}{\lambda} + \sqrt{\frac{3n}{\delta Q} \left( 3 \|\nabla f(\boldsymbol{x})\|^2 + \frac{\beta^2 \lambda^2}{4}(d+2)(d+4) + \frac{4\epsilon^2}{\lambda^2} \right)}$$

*where $\sup_{\boldsymbol{x} \in \mathcal{X}} |y(\boldsymbol{x}) - f(\boldsymbol{x})| \leq \epsilon$.*

**Remark.** In our setting (see Sec. 2), we in fact have the following result with a probability of at least $1 - \delta$ by applying the Chernoff bound on the Gaussian observation noise $\zeta$:

$$\epsilon = \sqrt{2 \ln(2/\delta)} \sigma , \tag{85}$$

which is regarded as a constant in our following proofs. By additionally assuming that the gradient of $f$ be bounded (i.e., $\|\nabla f(\boldsymbol{x})\| \leq c$ for any $\boldsymbol{x}$ in the domain of $f$ and some $c > 0$), we have

$$\|\boldsymbol{\Delta}(\boldsymbol{x}) - \nabla f(\boldsymbol{x})\| \leq \Lambda + \mathcal{O}\left(\frac{1}{\sqrt{Q}}\right) \tag{86}$$

where the constant $\Lambda$ is defined as $\Lambda \triangleq \beta \lambda \sqrt{d} + \frac{\epsilon \sqrt{d}}{\lambda}$. Note that this additional constant term in (86) can not be avoided, which thus is another pitfall of the FD method in addition to its query inefficiency as discussed in our Sec. 3.2.

Based on the results above, we can get the following upper bounds for the gradient estimation methods in the existing federated ZOO algorithms. Note that, we usually keep the constant before $\mathcal{O}\left(\frac{1}{Q}\right)$ to deliver a more detailed comparison among different federated ZOO algorithms throughout this section.

**FedZO Algorithm.** For FedZO (Fang et al., 2022), it applies the following gradient estimation for every local update in Algo. 1:

$$\widehat{\boldsymbol{g}}_{r,t-1}^{(i)} = \boldsymbol{\Delta}^{(i)}(\boldsymbol{x}_{r,t-1}^{(i)}) . \tag{87}$$

That is, $\gamma_{r,t-1}^{(i)} = 0$ and $\boldsymbol{g}_{r,t-1}^{(i)} = \boldsymbol{\Delta}^{(i)}(\boldsymbol{x}_{r,t-1}^{(i)})$ in (2). We provide the following gradient disparity bound for such a gradient estimation method when it is applied in Algo. 1.

**Proposition D.1.** *Assume that $\frac{1}{N}\sum_{i=1}^{N} \|\nabla f_i(\boldsymbol{x}) - \nabla F(\boldsymbol{x})\|^2 \leq G$ for any $\boldsymbol{x} \in \mathcal{X}$ and $f_i$ is $\beta$-smooth with bounded gradient for any $i \in [N]$. When applying (87) in Algo. 1, the following then holds with a constant probability for some $\Lambda > 0$,*

$$\frac{1}{N} \sum_{i=1}^{N} \Xi_{r,t}^{(i)} \leq 4\Lambda^2 + 2G + 4\mathcal{O}\left(\frac{1}{Q}\right) .$$

*Proof.*

$$\frac{1}{N}\sum_{i=1}^{N}\Xi_{r,t}^{(i)} \overset{(a)}{=} \frac{1}{N}\sum_{i=1}^{N}\left\|\mathbf{\Delta}^{(i)}(\boldsymbol{x}_{r,t-1}^{(i)}) - \nabla F(\boldsymbol{x}_{r,t-1}^{(i)})\right\|^2$$

$$\overset{(b)}{=} \frac{1}{N}\sum_{i=1}^{N}\left\|\mathbf{\Delta}^{(i)}(\boldsymbol{x}_{r,t-1}^{(i)}) - \nabla f_i(\boldsymbol{x}_{r,t-1}^{(i)}) + \nabla f_i(\boldsymbol{x}_{r,t-1}^{(i)}) - \nabla F(\boldsymbol{x}_{r,t-1}^{(i)})\right\|^2$$

$$\overset{(c)}{\leq} \frac{1}{N}\sum_{i=1}^{N}2\left(\left\|\mathbf{\Delta}^{(i)}(\boldsymbol{x}_{r,t-1}^{(i)}) - \nabla f_i(\boldsymbol{x}_{r,t-1}^{(i)})\right\|^2 + \left\|\nabla f_i(\boldsymbol{x}_{r,t-1}^{(i)}) - \nabla F(\boldsymbol{x}_{r,t-1}^{(i)})\right\|^2\right)$$

$$\overset{(d)}{\leq} 4\Lambda^2 + 2G + 4\mathcal{O}\left(\frac{1}{Q}\right)$$

(88)

where $(c)$ comes from Lemma C.5 and $(d)$ is based on Lemma C.5 as well as the result in (86). □

**FedProx Algorithm.** For FedProx in the federated ZOO setting (i.e., by simply combining Fed-Prox from (Li et al., 2020a) with the standard FD method in (3)), it has the gradient estimation form as follows:

$$\widehat{\boldsymbol{g}}_{r,t-1}^{(i)} = \mathbf{\Delta}^{(i)}(\boldsymbol{x}_{r,t-1}^{(i)}) + \gamma(\boldsymbol{x}_{r,t-1}^{(i)} - \boldsymbol{x}_{r-1})$$ (89)

where $\gamma$ is a constant. That is, $\gamma_{r,t-1}^{(i)} = \gamma$, $\boldsymbol{g}_{r,t-1}^{(i)} = \mathbf{\Delta}^{(i)}(\boldsymbol{x}_{r,t-1}^{(i)})$ and $\boldsymbol{g}_{r-1}(\boldsymbol{x}') - \boldsymbol{g}_{r-1}^{(i)}(\boldsymbol{x}'') = \boldsymbol{x}_{r,t-1}^{(i)} - \boldsymbol{x}_{r-1}$ in (2). We provide the following gradient disparity bound for such a gradient estimation method when it is applied in Algo. 1.

**Proposition D.2.** *Assume that $\frac{1}{N}\sum_{i=1}^{N}\|\nabla f_i(\boldsymbol{x}) - \nabla F(\boldsymbol{x})\|^2 \leq G$ for any $\boldsymbol{x} \in \mathcal{X}$ and $f_i$ is $\beta$-smooth with bounded gradient for any $i \in [N]$. When applying (89) in Algo. 1, the following then holds with a constant probability for some $\Lambda > 0$,*

$$\frac{1}{N}\sum_{i=1}^{N}\Xi_{r,t}^{(i)} \leq 6\Lambda^2 + 3G + \frac{3\gamma^2}{N}\sum_{i=1}^{N}\left\|\boldsymbol{x}_{r,t-1}^{(i)} - \boldsymbol{x}_{r-1}\right\|^2 + 6\mathcal{O}\left(\frac{1}{Q}\right).$$

*Proof.*

$$\frac{1}{N}\sum_{i=1}^{N}\Xi_{r,t}^{(i)} \overset{(a)}{=} \frac{1}{N}\sum_{i=1}^{N}\left\|\mathbf{\Delta}^{(i)}(\boldsymbol{x}_{r,t-1}^{(i)}) + \gamma\left(\boldsymbol{x}_{r,t-1}^{(i)} - \boldsymbol{x}_{r-1}\right) - \nabla F(\boldsymbol{x}_{r,t-1}^{(i)})\right\|^2$$

$$\overset{(b)}{=} \frac{1}{N}\sum_{i=1}^{N}\left\|\mathbf{\Delta}^{(i)}(\boldsymbol{x}_{r,t-1}^{(i)}) - \nabla f_i(\boldsymbol{x}_{r,t-1}^{(i)}) + \nabla f_i(\boldsymbol{x}_{r,t-1}^{(i)}) - \nabla F(\boldsymbol{x}_{r,t-1}^{(i)}) + \gamma\left(\boldsymbol{x}_{r,t-1}^{(i)} - \boldsymbol{x}_{r-1}\right)\right\|^2$$

$$\overset{(c)}{\leq} \frac{1}{N}\sum_{i=1}^{N}3\left(\left\|\mathbf{\Delta}^{(i)}(\boldsymbol{x}_{r,t-1}^{(i)}) - \nabla f_i(\boldsymbol{x}_{r,t-1}^{(i)})\right\|^2 + \left\|\nabla f_i(\boldsymbol{x}_{r,t-1}^{(i)}) - \nabla F(\boldsymbol{x}_{r,t-1}^{(i)})\right\|^2\right)$$

$$\quad + \frac{3\gamma^2}{N}\sum_{i=1}^{N}\left\|\boldsymbol{x}_{r,t-1}^{(i)} - \boldsymbol{x}_{r-1}\right\|^2$$

$$\overset{(d)}{\leq} 6\Lambda^2 + 3G + \frac{3\gamma^2}{N}\sum_{i=1}^{N}\left\|\boldsymbol{x}_{r,t-1}^{(i)} - \boldsymbol{x}_{r-1}\right\|^2 + 6\mathcal{O}\left(\frac{1}{Q}\right).$$

(90)

Similarly, $(c)$ is from Lemma C.5 and $(d)$ is based on Lemma C.5 as well as the result in (86). □

**SCAFFOLD (Type I) Algorithm.** For SCAFFOLD using its Type I gradient correction in the federated ZOO setting (i.e., by simply combining SCAFFOLD (Type I) from (Karimireddy et al., 2020a) with the standard FD method in (3)), it has the gradient estimation form as follows:

$$\widehat{\boldsymbol{g}}_{r,t-1}^{(i)} = \mathbf{\Delta}^{(i)}(\boldsymbol{x}_{r,t-1}^{(i)}) + \frac{1}{N}\sum_{j=1}^{N}\mathbf{\Delta}^{(j)}(\boldsymbol{x}_{r-1}) - \mathbf{\Delta}^{(i)}(\boldsymbol{x}_{r-1}).$$ (91)

That is, $\gamma_{r,t-1}^{(i)} = 1$, $\boldsymbol{g}_{r,t-1}^{(i)} = \boldsymbol{\Delta}^{(i)}(\boldsymbol{x}_{r,t-1}^{(i)})$ and $\boldsymbol{g}_{r-1}(\boldsymbol{x}') - \boldsymbol{g}_{r-1}^{(i)}(\boldsymbol{x}'') = \frac{1}{N}\sum_{j=1}^{N}\boldsymbol{\Delta}^{(j)}(\boldsymbol{x}_{r-1}) - \boldsymbol{\Delta}^{(i)}(\boldsymbol{x}_{r-1})$ in (2). Of note, similar to our FZooS where an additional transmission is required when we actively query in the neighborhod of $\boldsymbol{x}_r$ in line 7 of Algo. 2, SCAFFOLD (Type I) also needs another server-client transmission of $\frac{1}{N}\sum_{j=1}^{N}\boldsymbol{\Delta}^{(j)}(\boldsymbol{x}_{r-1})$ for gradient correction. We provide the following gradient disparity bound for such a gradient estimation method when it is applied in Algo. 1.

**Proposition D.3.** *Assume that $f_i$ is $\beta$-smooth with bounded gradient for any $i \in [N]$. When applying (91) in Algo. 1, the following then holds with a constant probability for some $\Lambda > 0$,*

$$\frac{1}{N}\sum_{i=1}^{N}\Xi_{r,t}^{(i)} \le 18\Lambda^2 + \frac{6\beta^2}{N}\sum_{i=1}^{N}\left\|\boldsymbol{x}_{r,t-1}^{(i)} - \boldsymbol{x}_{r-1}\right\|^2 + 18\mathcal{O}\left(\frac{1}{Q}\right).$$

*Proof.*

$$
\begin{aligned}
\frac{1}{N}\sum_{i=1}^{N}\Xi_{r,t}^{(i)} &\overset{(a)}{=} \frac{1}{N}\sum_{i=1}^{N}\left\|\boldsymbol{\Delta}^{(i)}(\boldsymbol{x}_{r,t-1}^{(i)}) + \left(\frac{1}{N}\sum_{j=1}^{N}\boldsymbol{\Delta}^{(j)}(\boldsymbol{x}_{r-1}) - \boldsymbol{\Delta}^{(i)}(\boldsymbol{x}_{r-1})\right) - \nabla F(\boldsymbol{x}_{r,t-1}^{(i)})\right\|^2 \\
&\overset{(b)}{=} \frac{1}{N}\sum_{i=1}^{N}\left\|\boldsymbol{\Delta}^{(i)}(\boldsymbol{x}_{r,t-1}^{(i)}) - \nabla f_i(\boldsymbol{x}_{r,t-1}^{(i)}) + \frac{1}{N}\sum_{j=1,j\neq i}^{N}\left(\boldsymbol{\Delta}^{(j)}(\boldsymbol{x}_{r-1}) - \nabla f_j(\boldsymbol{x}_{r,t-1}^{(i)})\right)\right. \\
&\qquad\qquad \left. + \frac{N-1}{N}\left(\nabla f_i(\boldsymbol{x}_{r,t-1}^{(i)}) - \boldsymbol{\Delta}^{(i)}(\boldsymbol{x}_{r-1})\right)\right\|^2 \\
&\overset{(c)}{\le} \frac{3}{N}\sum_{i=1}^{N}\left\|\boldsymbol{\Delta}^{(i)}(\boldsymbol{x}_{r,t-1}^{(i)}) - \nabla f_i(\boldsymbol{x}_{r,t-1}^{(i)})\right\|^2 + \frac{3}{N^3}\sum_{i=1}^{N}\left\|\sum_{j=1,j\neq i}^{N}\left(\boldsymbol{\Delta}^{(j)}(\boldsymbol{x}_{r-1}) - \nabla f_j(\boldsymbol{x}_{r,t-1}^{(i)})\right)\right\|^2 \\
&\qquad\qquad + \frac{3(N-1)^2}{N^3}\sum_{i=1}^{N}\left\|\nabla f_i(\boldsymbol{x}_{r,t-1}^{(i)}) - \boldsymbol{\Delta}^{(i)}(\boldsymbol{x}_{r-1})\right\|^2 \\
&\overset{(d)}{\le} \frac{3}{N}\sum_{i=1}^{N}\left\|\boldsymbol{\Delta}^{(i)}(\boldsymbol{x}_{r,t-1}^{(i)}) - \nabla f_i(\boldsymbol{x}_{r,t-1}^{(i)})\right\|^2 + \frac{3(N-1)}{N^3}\sum_{i=1}^{N}\sum_{j=1,j\neq i}^{N}\left\|\boldsymbol{\Delta}^{(j)}(\boldsymbol{x}_{r-1}) - \nabla f_j(\boldsymbol{x}_{r,t-1}^{(i)})\right\|^2 \\
&\qquad\qquad + \frac{3(N-1)^2}{N^3}\sum_{i=1}^{N}\left\|\nabla f_i(\boldsymbol{x}_{r,t-1}^{(i)}) - \boldsymbol{\Delta}^{(i)}(\boldsymbol{x}_{r-1})\right\|^2 \\
&\overset{(e)}{\le} \frac{3}{N}\sum_{i=1}^{N}\left\|\boldsymbol{\Delta}^{(i)}(\boldsymbol{x}_{r,t-1}^{(i)}) - \nabla f_i(\boldsymbol{x}_{r,t-1}^{(i)})\right\|^2 + \frac{6(N-1)}{N^2}\sum_{j=1}^{N}\left\|\boldsymbol{\Delta}^{(j)}(\boldsymbol{x}_{r,t-1}^{(i)}) - \nabla f_j(\boldsymbol{x}_{r,t-1}^{(i)})\right\|^2 \\
&\qquad\qquad + \frac{6\beta^2(N-1)^2}{N^2}\sum_{j=1}^{N}\left\|\boldsymbol{x}_{r,t-1}^{(i)} - \boldsymbol{x}_{r-1}\right\|^2 \\
&\overset{(f)}{\le} \frac{9}{N}\sum_{i=1}^{N}\left\|\boldsymbol{\Delta}^{(i)}(\boldsymbol{x}_{r,t-1}^{(i)}) - \nabla f_i(\boldsymbol{x}_{r,t-1}^{(i)})\right\|^2 + 6\beta^2\sum_{i=1}^{N}\left\|\boldsymbol{x}_{r,t-1}^{(i)} - \boldsymbol{x}_{r-1}\right\|^2 \\
&\overset{(g)}{\le} 18\Lambda^2 + 6\beta^2\sum_{i=1}^{N}\left\|\boldsymbol{x}_{r,t-1}^{(i)} - \boldsymbol{x}_{r-1}\right\|^2 + 18\mathcal{O}\left(\frac{1}{Q}\right)
\end{aligned}
$$

$$(92)$$

Similarly, $(c), (d)$ are from Lemma C.5 and $(e)$ is because of the smoothness of $F$ as well as (28) with $a = \frac{1}{N-1}$. Finally, $(g)$ follows from the results in (86) as well as the result in Lemma C.5. $\quad\square$

**SCAFFOLD (Type II) Algorithm.** For SCAFFOLD using its Type II gradient correction in the federated ZOO setting (i.e., by simply combining SCAFFOLD (Type II) from (Karimireddy et al.,

2020a) with the standard FD method in (3)), it has the gradient estimation form as follows:

$$\widehat{\boldsymbol{g}}_{r,t-1}^{(i)} = \boldsymbol{\Delta}^{(i)}(\boldsymbol{x}_{r,t-1}^{(i)}) + \frac{1}{NT} \sum_{j=1}^{N} \sum_{\tau=1}^{T} \boldsymbol{\Delta}^{(j)}(\boldsymbol{x}_{r-1,\tau-1}^{(j)}) - \frac{1}{T} \sum_{\tau=1}^{T} \boldsymbol{\Delta}^{(i)}(\boldsymbol{x}_{r-1,\tau-1}^{(i)}) . \quad (93)$$

That is, $\boldsymbol{g}_{r-1}(\boldsymbol{x}') - \boldsymbol{g}_{r-1}^{(i)}(\boldsymbol{x}'') = \frac{1}{NT} \sum_{j=1}^{N} \sum_{\tau=1}^{T} \boldsymbol{\Delta}^{(j)}(\boldsymbol{x}_{r-1,\tau-1}^{(j)}) - \frac{1}{T} \sum_{\tau=1}^{T} \boldsymbol{\Delta}^{(i)}(\boldsymbol{x}_{r-1,\tau-1}^{(i)})$, $\boldsymbol{g}_{r,t-1}^{(i)} = \boldsymbol{\Delta}^{(i)}(\boldsymbol{x}_{r,t-1}^{(i)})$ and $\gamma_{r,t-1}^{(i)} = 1$ in (2). Interestingly, SCAFFOLD (Type II) servers as an approximation of SCAFFOLD (Type I), which in fact does not require another server-client transmission for gradient correction as discussed in (Karimireddy et al., 2020a). This is because $\frac{1}{NT} \sum_{j=1}^{N} \sum_{\tau=1}^{T} \boldsymbol{\Delta}^{(j)}(\boldsymbol{x}_{r-1,\tau-1}^{(j)})$ can be computed before the aggregation of $\{\boldsymbol{x}_{r-1,T}^{(i)}\}_{i=1}^{N}$ on server. We provide the following gradient disparity bound for such a gradient estimation method when it is applied in Algo. 1.

**Proposition D.4.** *Assume that $f_i$ is c-continuous and $\beta$-smooth for any $i \in [N]$ and the randomly sampled $\{\boldsymbol{u}_q\}_{q=1}^{Q}$ in (3) are shared across all iterations and rounds. When applying (93) in Algo. 1, the following then holds with a constant probability for some $\Lambda, a > 0$,*

$$\frac{1}{N} \sum_{i=1}^{N} \Xi_{r,t}^{(i)} \leq 18\Lambda^2 + \frac{24ac^2}{\lambda^2 T} \sum_{i=1}^{N} \sum_{\tau=1}^{T} \left\| \boldsymbol{x}_{r,t-1}^{(i)} - \boldsymbol{x}_{r-1,\tau-1}^{(i)} \right\|^2 + 6\mathcal{O}\left(\frac{1}{Q}\right) + 12\mathcal{O}\left(\frac{1}{TQ}\right) .$$

*Proof.* We slightly abuse notation and use $\boldsymbol{\Delta}_T^{(i)}(\boldsymbol{x}_{r,t-1}^{(i)})$ to denote the FD method in (3) using $TQ$ function queries for the gradient estimation at input $\boldsymbol{x}_{r,t-1}^{(i)}$ on client $i$. Based on this notation, we

then have

$$\frac{1}{N} \sum_{i=1}^{N} \Xi_{r,t}^{(i)}$$

$$\overset{(a)}{=} \frac{1}{N} \sum_{i=1}^{N} \left\| \boldsymbol{\Delta}^{(i)}(\boldsymbol{x}_{r,t-1}^{(i)}) + \left( \frac{1}{NT} \sum_{j=1}^{N} \sum_{\tau=1}^{T} \boldsymbol{\Delta}^{(j)}(\boldsymbol{x}_{r-1,\tau-1}^{(j)}) - \frac{1}{T} \sum_{\tau=1}^{T} \boldsymbol{\Delta}^{(i)}(\boldsymbol{x}_{r-1,\tau-1}^{(i)}) \right) - \nabla F(\boldsymbol{x}_{r,t-1}^{(i)}) \right\|^2$$

$$\overset{(b)}{=} \frac{1}{N} \sum_{i=1}^{N} \left\| \boldsymbol{\Delta}^{(i)}(\boldsymbol{x}_{r,t-1}^{(i)}) - \nabla f_i(\boldsymbol{x}_{r,t-1}^{(i)}) + \frac{N-1}{N} \left( \nabla f_i(\boldsymbol{x}_{r,t-1}^{(i)}) - \frac{1}{T} \sum_{\tau=1}^{T} \boldsymbol{\Delta}^{(i)}(\boldsymbol{x}_{r-1,\tau-1}^{(i)}) \right) \right.$$

$$\left. + \frac{1}{NT} \sum_{j=1, j\neq i}^{N} \sum_{\tau=1}^{T} \left( \boldsymbol{\Delta}^{(j)}(\boldsymbol{x}_{r-1,\tau-1}^{(j)}) - \nabla f_j(\boldsymbol{x}_{r,t-1}^{(j)}) \right) \right\|^2$$

$$\overset{(c)}{\leq} \frac{3}{N} \sum_{i=1}^{N} \left\| \boldsymbol{\Delta}^{(i)}(\boldsymbol{x}_{r,t-1}^{(i)}) - \nabla f_i(\boldsymbol{x}_{r,t-1}^{(i)}) \right\|^2$$

$$+ \frac{3(N-1)^2}{N^3} \sum_{i=1}^{N} \left\| \left( \nabla f_i(\boldsymbol{x}_{r,t-1}^{(i)} - \boldsymbol{\Delta}_T^{(i)}(\boldsymbol{x}_{r,t-1}^{(i)})) \right) + \left( \boldsymbol{\Delta}_T^{(i)}(\boldsymbol{x}_{r,t-1}^{(i)}) - \frac{1}{T} \sum_{\tau=1}^{T} \boldsymbol{\Delta}^{(i)}(\boldsymbol{x}_{r-1,\tau-1}^{(i)}) \right) \right\|^2$$

$$+ \frac{3}{N^3} \sum_{i=1}^{N} \left\| \sum_{j=1, j\neq 1}^{N} \left[ \left( \nabla f_j(\boldsymbol{x}_{r,t-1}^{(j)}) - \boldsymbol{\Delta}_T^{(j)}(\boldsymbol{x}_{r,t-1}^{(j)}) \right) + \left( \boldsymbol{\Delta}_T^{(j)}(\boldsymbol{x}_{r,t-1}^{(j)}) - \frac{1}{T} \sum_{\tau=1}^{T} \boldsymbol{\Delta}^{(j)}(\boldsymbol{x}_{r-1,\tau-1}^{(j)}) \right) \right] \right\|^2$$

$$\overset{(d)}{\leq} \frac{3}{N} \sum_{i=1}^{N} \left\| \boldsymbol{\Delta}^{(i)}(\boldsymbol{x}_{r,t-1}^{(i)}) - \nabla f_i(\boldsymbol{x}_{r,t-1}^{(i)}) \right\|^2$$

$$+ \frac{3(N-1)^2}{N^3} \sum_{i=1}^{N} \left( \left( 1 + \frac{1}{N-1} \right) \left\| \nabla f_i(\boldsymbol{x}_{r,t-1}^{(i)} - \boldsymbol{\Delta}_T^{(i)}(\boldsymbol{x}_{r,t-1}^{(i)})) \right\|^2 \right.$$

$$\left. + N \left\| \boldsymbol{\Delta}_T^{(i)}(\boldsymbol{x}_{r,t-1}^{(i)}) - \frac{1}{T} \sum_{\tau=1}^{T} \boldsymbol{\Delta}^{(i)}(\boldsymbol{x}_{r-1,\tau-1}^{(i)}) \right\|^2 \right)$$

$$+ \frac{3(N-1)}{N^3} \sum_{i=1}^{N} \sum_{j=1, j\neq 1}^{N} \left( \left( 1 + \frac{1}{N-1} \right) \left\| \nabla f_j(\boldsymbol{x}_{r,t-1}^{(j)}) - \boldsymbol{\Delta}_T^{(j)}(\boldsymbol{x}_{r,t-1}^{(j)}) \right\|^2 \right.$$

$$\left. + N \left\| \boldsymbol{\Delta}_T^{(j)}(\boldsymbol{x}_{r,t-1}^{(j)}) - \frac{1}{T} \sum_{\tau=1}^{T} \boldsymbol{\Delta}^{(j)}(\boldsymbol{x}_{r-1,\tau-1}^{(j)}) \right\|^2 \right) \tag{94}$$

Similarly, $(c)$ are from (29) in Lemma C.5 and $(d)$ is because of (28) in Lemma C.5 with $a = \frac{N}{N-1}$.

We then bound $\left\| \mathbf{\Delta}_T^{(i)}(\boldsymbol{x}_{r,t-1}^{(i)}) - \frac{1}{T}\sum_{\tau=1}^T \mathbf{\Delta}^{(i)}(\boldsymbol{x}_{r-1,\tau-1}^{(i)}) \right\|^2$ as below

$$
\left\| \mathbf{\Delta}_T^{(i)}(\boldsymbol{x}_{r,t-1}^{(i)}) - \frac{1}{T}\sum_{\tau=1}^T \mathbf{\Delta}^{(i)}(\boldsymbol{x}_{r-1,\tau-1}^{(i)}) \right\|^2
$$

$$
\overset{(a)}{\leq} \frac{1}{T}\sum_{\tau=1}^T \left\| \mathbf{\Delta}^{(i)}(\boldsymbol{x}_{r,t-1}^{(i)}) - \mathbf{\Delta}^{(i)}(\boldsymbol{x}_{r-1,\tau-1}^{(i)}) \right\|^2
$$

$$
\overset{(b)}{=} \frac{1}{T}\sum_{\tau=1}^T \left\| \frac{1}{Q}\sum_{q=1}^Q \left( y_i(\boldsymbol{x}_{r-1,\tau-1}^{(i)} + \lambda\boldsymbol{u}_q) - y_i(\boldsymbol{x}_{r,t-1}^{(i)} + \lambda\boldsymbol{u}_q) + y_i(\boldsymbol{x}_{r,t-1}^{(i)}) - y_i(\boldsymbol{x}_{r-1,\tau-1}^{(i)}) \right) \frac{\boldsymbol{u}_q}{\lambda} \right\|^2
$$

$$
\overset{(c)}{\leq} \frac{1}{\lambda^2 TQ}\sum_{\tau=1}^T\sum_{q=1}^Q \left| y_i(\boldsymbol{x}_{r-1,\tau-1}^{(i)} + \lambda\boldsymbol{u}_q) - y_i(\boldsymbol{x}_{r,t-1}^{(i)} + \lambda\boldsymbol{u}_q) + y_i(\boldsymbol{x}_{r,t-1}^{(i)}) - y_i(\boldsymbol{x}_{r-1,\tau-1}^{(i)}) \right|^2 \|\boldsymbol{u}_q\|^2
$$

$$
\overset{(d)}{=} \frac{1}{\lambda^2 TQ}\sum_{\tau=1}^T\sum_{q=1}^Q 2\left| f_i(\boldsymbol{x}_{r-1,\tau-1}^{(i)} + \lambda\boldsymbol{u}_q) - f_i(\boldsymbol{x}_{r,t-1}^{(i)} + \lambda\boldsymbol{u}_q) + f_i(\boldsymbol{x}_{r,t-1}^{(i)}) - f_i(\boldsymbol{x}_{r-1,\tau-1}^{(i)}) \right|^2 \|\boldsymbol{u}_q\|^2
$$

$$
+ \frac{1}{\lambda^2 TQ}\sum_{q=1}^Q 2\left| \zeta_{r-1,\tau-1}^{(i)} - \zeta_{r,t-1}^{(i)} + \zeta_{r-1,\tau-1}^{(i)'} - \zeta_{r,t-1}^{(i)'} \right|^2 \|\boldsymbol{u}_q\|^2
$$

$$
\overset{(e)}{\leq} \frac{1}{\lambda^2 TQ}\sum_{\tau=1}^T\sum_{q=1}^Q 4\left( \left| f_i(\boldsymbol{x}_{r-1,\tau-1}^{(i)} + \lambda\boldsymbol{u}_q) - f_i(\boldsymbol{x}_{r,t-1}^{(i)} + \lambda\boldsymbol{u}_q) \right|^2 + \left| f_i(\boldsymbol{x}_{r,t-1}^{(i)}) - f_i(\boldsymbol{x}_{r-1,\tau-1}^{(i)}) \right|^2 \right) \|\boldsymbol{u}_q\|^2
$$

$$
+ \frac{1}{\lambda^2 TQ}\sum_{q=1}^Q 8\epsilon^2 \|\boldsymbol{u}_q\|^2
$$

$$
\overset{(f)}{\leq} \frac{8}{\lambda^2 T}\sum_{\tau=1}^T \left( c^2 \left\| \boldsymbol{x}_{r,t-1}^{(i)} - \boldsymbol{x}_{r-1,\tau-1}^{(i)} \right\|^2 + \epsilon^2 \right)\left( \frac{1}{Q}\sum_{q=1}^Q \|\boldsymbol{u}_q\|^2 \right)
$$

$$
\overset{(g)}{\leq} \frac{8a}{\lambda^2 T}\sum_{\tau=1}^T \left( c^2 \left\| \boldsymbol{x}_{r,t-1}^{(i)} - \boldsymbol{x}_{r-1,\tau-1}^{(i)} \right\|^2 + \epsilon^2 \right)
$$

$$(95)$$

where $(a)$, $(d)$, $(e)$ are due to (29) in Lemma C.5. Note that $(d)$ is valid because $\{\boldsymbol{u}_q\}_{q=1}^Q$ in (3) is assumed to be shared across all iterations and rounds. In addition, $(c)$ is from the Cauchy–Schwarz inequality and $(f)$ is based on the continuity of $F$, i.e., $\|F(\boldsymbol{x}) - F(\boldsymbol{x}')\| \leq c$ for any $\boldsymbol{x}, \boldsymbol{x}' \in \mathcal{X}$. Finally, $(g)$ is from Lemma C.2 and $a \triangleq d + 2\sqrt{dQ^{-1}\ln(1/\delta)} + 2Q^{-1}\ln(1/\delta)$.

Finally, by introducing (95) into (94), we have

$$
\frac{1}{N}\sum_{i=1}^N \Xi_{r,t}^{(i)} \overset{(a)}{\leq} \frac{3}{N}\sum_{i=1}^N \left\| \mathbf{\Delta}^{(i)}(\boldsymbol{x}_{r,t-1}^{(i)}) - \nabla f_i(\boldsymbol{x}_{r,t-1}^{(i)}) \right\|^2 + \frac{6(N-1)}{N^2}\sum_{i=1}^N \left\| \nabla f_i(\boldsymbol{x}_{r,t-1}^{(i)}) - \mathbf{\Delta}_T^{(i)}(\boldsymbol{x}_{r,\tau-1}^{(i)}) \right\|^2
$$

$$
+ \frac{24a(N-1)^2}{\lambda^2 TN^2}\sum_{i=1}^N\sum_{\tau=1}^T \left( c^2 \left\| \boldsymbol{x}_{r,t-1}^{(i)} - \boldsymbol{x}_{r-1,\tau-1}^{(i)} \right\|^2 + \epsilon^2 \right)
$$

$$
+ \frac{24a(N-1)}{\lambda^2 TN^2}\sum_{j=1,j\neq1}^N\sum_{\tau=1}^T \left( c^2 \left\| \boldsymbol{x}_{r,t-1}^{(j)} - \boldsymbol{x}_{r-1,\tau-1}^{(j)} \right\|^2 + \epsilon^2 \right)
$$

$$
\overset{(b)}{\leq} 18\Lambda^2 + \frac{24ac^2}{\lambda^2 T}\sum_{i=1}^N\sum_{\tau=1}^T \left\| \boldsymbol{x}_{r,t-1}^{(i)} - \boldsymbol{x}_{r-1,\tau-1}^{(i)} \right\|^2 + 6\mathcal{O}\left(\frac{1}{Q}\right) + 12\mathcal{O}\left(\frac{1}{TQ}\right)
$$

$$(96)$$

Finally, $(b)$ follows from the results in (86) as well as the result in Lemma C.5, which finally concludes our proof. $\qquad\square$

**Comparison and Discussion.** By comparing the upper bounds in Prop. D.1, D.2, D.3, and D.4 above with the one in our Thm. 1, we can summarize certain interesting insights as follows, which, to the best of our knowledge, has never been formally presented in the literature of federated ZOO.

(i) The gradient disparity of existing federated ZOO algorithms consistently has an additional constant error term (i.e., $\Lambda^2$) that can not be avoided. Remarkably, no additional constant error term occurs in the gradient disparity bound of our (8).

(ii) The gradient disparity of existing federated ZOO algorithms typically can only be reduced at a polynomial rate of $Q$ whereas our (8) is able to achieve an exponential rate of reduction for its gradient disparity.

(iii) FedProx achieves an even worse gradient disparity when compared with FedZO by introducing an additional error term $\frac{3\gamma^2}{N}\sum_{i=1}^{N}\left\|\boldsymbol{x}_{r,t-1}^{(i)} - \boldsymbol{x}_{r-1}\right\|^2$. This may explain its worst convergence in Sec. 6.

(iv) SCAFFOLD (Type I) and SCAFFOLD (Type II) are typically able to mitigate the impact of client heterogeneity (i.e., $G$) by enlarging the impact of the gradient estimation error that is resulting from the FD method applied in these two algorithms. This may lead to worse practical performance when the gradient estimation error outweighs the client heterogeneity, as shown in our Sec. 6.

(v) Although SCAFFOLD (Type II) is proposed to approximate SCAFFOLD (Type I) in the original paper (Karimireddy et al., 2020a), SCAFFOLD (Type II) in fact has the advantage of achieving a smaller gradient estimation error for gradient correction by increasing the number of additional function queries (i.e., the term $\mathcal{O}\left(\frac{1}{TQ}\right)$ in Prop. D.4), which is however at the cost of a likely increased input disparity (i.e., the term $\frac{24ac^2}{\lambda^2 T}\sum_{i=1}^{N}\sum_{\tau=1}^{T}\left\|\boldsymbol{x}_{r,t-1}^{(i)} - \boldsymbol{x}_{r-1,\tau-1}^{(i)}\right\|^2$ in Prop. D.4). Interestingly, federated ZOO usually prefers gradient correction of smaller gradient estimation errors, as suggested by the empirical results in our Sec. 6. This explains the reason why SCAFFOLD (Type II) usually outperforms SCAFFOLD (Type I) in federated ZOO, which differs from the scenario of federated FOO and therefore highlights the importance of an accurate gradient correction in federated ZOO.

## D.2 CONVERGENCE OF EXISTING FEDERATED ZOO ALGORITHMS

To establish the proof for the convergence of existing federated ZOO algorithms, we introduce the upper bound of gradient disparity $\frac{1}{N}\sum_{i=1}^{N}\Xi_{r,t}^{(i)}$ derived from our Prop. D.1, D.2, D.3, and D.4, into Thm. C.1. Particularly, to ease our proof, we mainly prove the convergence of existing federated ZOO algorithms when $F$ is non-convex and $\beta$-smooth. Similar to our Thm. 2, we define $D_0 \triangleq \|\boldsymbol{x}_0 - \boldsymbol{x}^*\|^2$ and $D_1 \triangleq F(\boldsymbol{x}_0) - F(\boldsymbol{x}^*)$, and assume that $\frac{1}{N}\sum_{i=1}^{N}\|\nabla f_i(\boldsymbol{x}) - \nabla F(\boldsymbol{x})\|^2 \leq G$ for any $\boldsymbol{x} \in \mathcal{X}$.

**Theorem D.1.** *FedZO enjoys the following convergence with a constant probability for some $\Lambda > 0$ when $\eta \leq \frac{7}{100\beta T}$,*

$$\min_{r \in [R+1)} \|\nabla F(\boldsymbol{x}_r)\|^2 \leq \mathcal{O}\left(\frac{D_1}{\eta RT} + \Lambda^2 + G + \frac{1}{Q}\right) .$$

*Proof.* Following the proof in our Appx. C.5, we have

$$
\begin{aligned}
\min_{r \in [R+1)} \|\nabla F(\boldsymbol{x}_r)\|^2 &\leq \frac{13(F(\boldsymbol{x}_0) - F(\boldsymbol{x}^*))}{\eta RT} + \frac{13}{\eta RT}\sum_{r=0}^{R}\sum_{i=1}^{N}\sum_{t=1}^{T}\left(\frac{(0.14\eta + 1/(2\beta T))}{N}\Xi_{r+1,t}^{(i)}\right.\\
&\quad \left. + \frac{1.02\eta^2\beta}{N}\sum_{\tau=1}^{t}S^{t-\tau}\Xi_{r+1,\tau}^{(i)}\right)\\
&\leq \mathcal{O}\left(\frac{D_1}{\eta RT} + \left(\Lambda^2 + G + \frac{1}{Q}\right) + \frac{1}{\beta}\left(\Lambda^2 + G + \frac{1}{Q}\right)\right)\\
&= \mathcal{O}\left(\frac{D_1}{\eta RT} + \Lambda^2 + G + \frac{1}{Q}\right) ,
\end{aligned}
$$

$$\tag{97}$$

which concludes our proof. □

**Remark.** Of note, this convergence aligns with one provided in (Fang et al., 2022), which hence supports the validity of our Thm. C.1 and Prop. D.1.

**Discussion.** Of note, the key to proving the convergence of other existing federated ZOO algorithms (i.e., FedProx and SCAFFOLD) lies in the bounded client drift (i.e., Lemma C.11) when additional input disparity is introduced in these algorithms. This in fact takes up a lot of space as shown in their original paper and is also out of the scope of this paper. As a consequence, we leave out the proof of the convergence of FedProx and SCAFFOLD in federated ZOO. Fortunately, the convergence (i.e., Thm. C.1) for the general optimization framework Algo. 1 implies that the key difference among the convergence of various federated ZOO algorithms in fact lies in their difference of gradient disparity. In light of this, based on our theoretical insights about the gradient disparity in different federated ZOO algorithms (Sec. D.1), we are still able to present the following insights into the advantages of our FZooS intuitively from the perspective of convergence:

(i) In general, the convergence of our FZooS in Appx. C.5 avoids the constant error term that can not be omitted in existing federated ZOO algorithms. Note that even the error term caused by RFF approximation (see Thm. 2) is in fact able to be mitigated by using a large number $M$ of random features.

(ii) Compared with the convergence of FedZO in Thm. D.1, the convergence of FZooS in Appx. C.5 demonstrates that the client heterogeneity can be effectively mitigated in FZooS and the gradient estimation term enjoys a better reduction rate (i.e., exponential rate vs. polynomial rate).

(iii) The bounded client drift in Lemma C.11 for the framework Algo. 1 implies that the additional input disparity from the FedProx in Prop. D.2, the SCAFFOLD (Type I) in Prop. D.3 and the SCAFFOLD (Type II) in Prop. D.4 likely leads to a larger client drift and consequently results in worse convergence compared with our FZooS, which has been empirically supported by the results in our Sec. 6 and Appx. F.

## APPENDIX E   EXPERIMENTAL SETTINGS

**General Settings.**   The gradient correction length is set to be $\gamma_{r,t-1}^{(i)} = 1/t$ such that it decays with the iteration of local updates $t$. We set the learning rate $\eta = 0.01$ and use Adam as the optimizer. As we described in line 7-8 of Algo. 2 and in Sec. 4.2.1, at each local update iteration, we actively query in the neighborhood of the input $\boldsymbol{x}_{r,t}^{(i)}$ on each client. Each time we generate 100 values of $\boldsymbol{x}_{r,t}^{(i)} + \boldsymbol{\delta}$ where each dimension of $\boldsymbol{\delta}'$ is uniformly sampled from $[-0.01, 0.01]$. We select the top 5 values with the highest uncertainty $\left\| \partial (\sigma_{r,t}^{(i)})^2 (\boldsymbol{x}_{r,t}^{(i)} + \boldsymbol{\delta}) \right\|$. We set the number of random features $M = 10000$ for the squared exponential kernel with a length scale of 1. Each dimension of the function input is normalized to be within $[0, 1]$ using the min-max normalization. The number of clients $N$, the number of local updates $T$, and the number of rounds $R$ vary for different experiments.

### E.1   SYNTHETIC EXPERIMENTS

Let input $\boldsymbol{x} = [x_j]_{j=1}^d \in [-10, 10]^d$, $\boldsymbol{a}^{(i)} = [a_j^{(i)}]_{j=1}^d$, and $\boldsymbol{b}^{(i)} = [b_j^{(i)}]_{j=1}^d$, then the quadratic functions on each client $i$ that has been applied in our Sec. 6.1 is in the form of

$$f_i(\boldsymbol{x}) = \frac{1}{10d} \left( \sum_{j \in [d]} \left[ \left( 1 + C \left( a_j^{(i)} - \frac{1}{N} \right) \right) x_j^2 + \left( 1 + C \left( b_j^{(i)} - \frac{1}{N} \right) \right) x_j \right] + 1 \right) \tag{98}$$

where every $[a_j^{(i)}]_{i=1}^N$ and $[b_j^{(i)}]_{i=1}^N$ are independently randomly sampled from the same Dirichlet distribution $\text{Dir}(\boldsymbol{\alpha})$ where $\boldsymbol{\alpha} = \frac{1}{N} \cdot \boldsymbol{1}$. So, given any $C > 0$, the final objective function remains

$$F(\boldsymbol{x}) = \frac{1}{10d} \left( \sum_{j \in [d]} \left[ x_j^2 + x_j \right] + 1 \right) . \tag{99}$$

Of note, $C$ is the constant that controls the client shift in our federated setting. Specifically, a larger $C$ typically leads to larger client shifts whereas a smaller $C$ usually enjoys smaller client shifts. We set the number of clients to be $N = 5$. We set $C \in \{0.5, 5, 50\}$ to vary the degree of heterogeneity (i.e., client shifts) among the local functions. The dimension of the function input is set to be $d = 300$. We set the number of local updates to be $T = 10$ and the number of rounds to be $R = 50$.

### E.2   FEDERATED BLACK-BOX ADVERSARIAL ATTACK

We set the number of clients $N = 10$ in this experiment. Before we conduct the adversarial attack, we need to train $N = 10$ models on different datasets to get the heterogeneous local model functions. To control the degree of heterogeneity among these functions, each time we sample $P \times 10$ classes among the 10 classes of the dataset (i.e., MNIST or CIFAR-10) and construct a dataset that only contains data points from these $P \times 10$ classes where $P \in [0, 1]$. Repeat the above procedures for 10 times to get 10 different datasets. Consequently, a higher $P$ means that the degree of heterogeneity among the local model functions is lower. As an example, when $P = 1$, all the local models of these clients will be exactly the same since they are all trained on the dataset with all 10 classes data points. For MNIST, we train a convolutional neural network (CNN) with two convolution layers followed by two fully connected layers on each dataset. For CIFAR-10, we train a ResNet18 on each dataset.

After obtaining these 10 local model functions for the clients, we proceed to select 15 data points from the test dataset. Specifically, we choose these data points among the ones that have been correctly classified by all of the 10 local models. These selected data points will be used as the targets for our attack. The goal is to find a perturbation $\boldsymbol{x}$, such that the modified image $\boldsymbol{z} + \boldsymbol{x}$ will be classified incorrectly by the model of each client. The local function takes the perturbed image $\boldsymbol{z} + \boldsymbol{x}$ as input and outputs the difference between the logit of the true class and the highest logit among all other classes except the true class. The condition for the attack to be successful is that the averaged output of $N = 10$ models misclassify the image $\boldsymbol{z} + \boldsymbol{x}$. The success rate is the portion of images that are successfully attacked among the selected 15 images. We set the number of local updates $T = 10$ and the number of rounds to be $R = 100$.

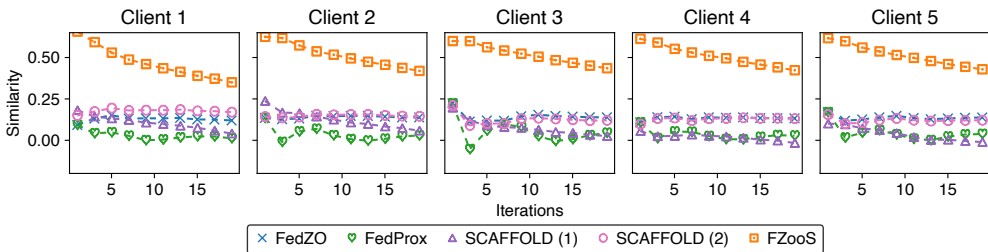

Figure 4: Comparison of the cosine similarity between $\widehat{g}_{r,t-1}^{(i)}$ and $\nabla F(x_{r,t-1})$ within one round (with local iterations $T = 20$) among different federated ZOO algorithms, where the $y$-axis denotes the cumulatively averaged similarity w.r.t. the $x$-axis (i.e., the iterations of local updates). Of note, for every iteration, our (8) will actively query only 5 additional function values, which is much fewer than the 20 additional queries in other existing algorithms based on FD methods.

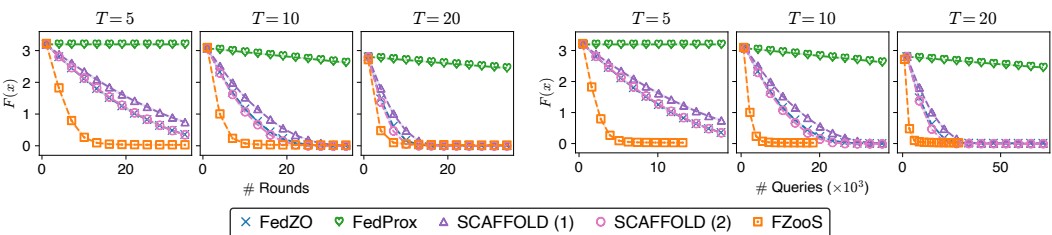

Figure 5: Comparison of the communication round and query efficiency between our FZooS and other existing baselines on the federated synthetic functions with a varying number $T$ of local updates.

### E.3 FEDERATED NON-DIFFERENTIABLE METRIC OPTIMIZATION

Following the practice in (Shu et al., 2023), we first train a 3-layer MLP model on the training dataset of Covertype (Dua and Graff, 2017) using the Cross-Entropy loss to obtain its fully converged parameters $\theta^*$. This is to simulate the federated learning (i.e., fine-tuning) of a pre-trained model with other non-differentiable metrics. Similar to the setting in Appx. E.2, we construct $N = 7$ datasets by sampling $P \times 7$ ($P \in [0, 1]$) classes from the test dataset each time. Again, the degree of heterogeneity among the local functions of the clients is controlled by $P$. The higher the value of $P$, the more heterogeneous local functions will be. In this experiment, we aim to find a perturbation $x$ to the model parameters $\theta^*$, such that $\theta^* + x$ will yield better performance for other non-differentiable metrics, e.g., precision and recall, by using the distributed datasets on clients. Specifically, the local function takes the perturbed model parameter as input and outputs the result of a non-differentiable metric (e.g., $1 -$ precision) that evaluates the performance of the model on the corresponding constructed dataset. We set $T = 10$ and $R = 50$. As in (Shu et al., 2023), we conduct experiments on four non-differentiable metrics, namely precision, recall, Jaccard score, and F1 score.

## APPENDIX F    MORE RESULTS

### F.1    SYNTHETIC EXPERIMENTS

In this section, we first compare the gradient disparity of existing federated ZOO algorithms and our FZooS algorithm using the quadratic functions (see Appx. E.1) with $d = 300$, $N = 5$, and $C = 5$. The results are in Fig. 4, showing that our proposed adaptive gradient estimation is indeed able to realize significantly improved estimation quality than other existing methods while requiring fewer function queries. This consequently verified the theoretical insights of Thm. 1. Interestingly, we notice that the quality of our (8) decreases when the number of iterations for local updates is increased, which is likely because the performance of our gradient surrogates suffers when the input $x$ for gradient estimation is far away from the historical function queries (i.e., few function information at $x$ can be used for predictions), as theoretically supported in our Appx. C.3. This also indicates the importance of active queries in our FZooS for consistently high-quality (8) by collecting more function information in the neighborhood of the potential updated inputs within the local updates.

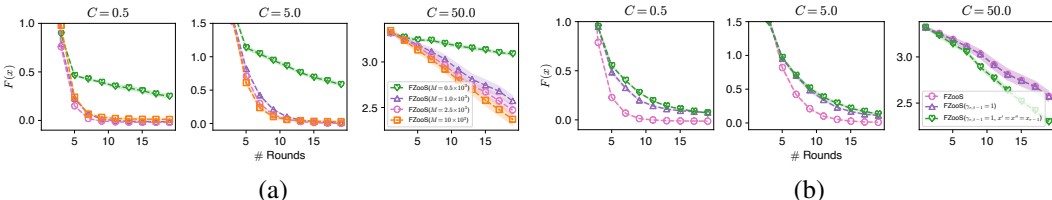

Figure 6: Comparison of the communication round efficiency of our FZooS (a) with a varying number $M$ of random features and (b) without adaptive gradient correction. Of note, $\gamma_{r,t-1} = 1$ means a fixed gradient correction length and $\boldsymbol{x}' = \boldsymbol{x}'' = \boldsymbol{x}_{r-1}$ stands for a fixed gradient correction vector as in SCAFFOLD.

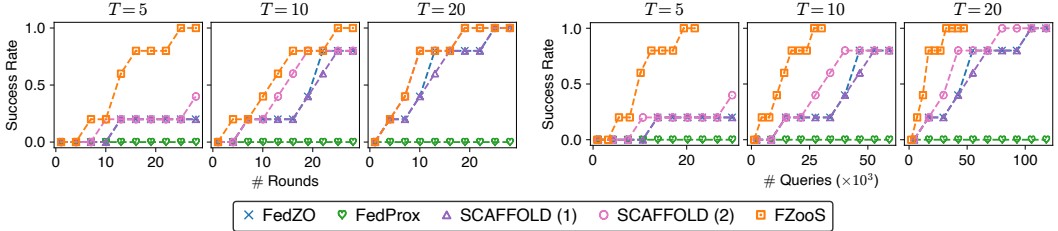

Figure 7: Comparison of the success rate achieved by FZooS and other existing federated ZOO algorithms on CIFAR-10 under a varying number $T$ of local updates.

In addition to the comparison using a quadratic function that is under varying heterogeneity through different $C$ in our Fig. 1, we present the comparison using a quadratic function that is under a varying number $T$ of local updates in Fig. 5. Remarkably, our FZooS still considerably outperforms other baselines in terms of both communication round efficiency and query efficiency. Interestingly, Fig. 5 shows that a larger $T$ usually improves the communication round efficiency of both our FZooS, as theoretically supported in our Thm. 2. However, such an improvement is usually smaller than the increasing scale of $T$. This also aligns with our Thm. 2 since our Thm. 2 demonstrates that the increasing $T$ fails to mitigate the impact of client heterogeneity. That is, term $G$ in Thm. 2 can not be reduced when $T$ is increased.

We finally present the comparison of the communication round efficiency of our FZooS (a) with a varying number $M$ of random features and (b) without adaptive gradient correction under varying client heterogeneity in Fig. 6. Of note, in Fig. 6, we only apply $M = 1000$ random features to facilitate a clear and direct comparison. Interestingly, Fig. 6(a) demonstrates that our FZooS of a larger number $M$ of random features generally is preferred for an improved communication round efficiency when the client heterogeneity (i.e., $C$) is increased, which thus aligns with the theoretical insights from our Thm. 2 in Sec. 5.2. Nevertheless, when client heterogeneity is small (e.g., $C \leq 5.0$), a moderate number of random features can already produce compelling and competitive convergence. Meanwhile, Fig. 6(b) illustrates that, in general, both our adaptive gradient correction vector and adaptive gradient correction length are essential for our FZooS to achieve remarkable convergence in practice. Surprisingly, our FZooS with fixed gradient correction outperforms its counterpart with adaptive gradient correction when client heterogeneity is large (i.e., $C = 50$). This is likely because a small number of random features (i.e., $M = 1000$) are applied when $C = 50$, making adaptive gradient correction generally inaccurate for a long horizon of local updates since the quality of our gradient surrogates decays w.r.t. the horizon (i.e., iterations) as shown in Fig. 4. This can also be verified from Fig. 6(a). On the contrary, the fixed gradient correction is already of reasonably good quality due to the smoothness of the global function $F$ (i.e., its gradients are continuous), which consequently can provide consistently good gradient correction along a long horizon of local updates when client heterogeneity is large (i.e., $C = 50$).

### F.2 FEDERATED BLACK-BOX ADVERSARIAL ATTACK

In addition to depicting the success rate of attacks on CIFAR-10 in Fig.2, which accounts for varying client heterogeneity, we also present the success rate of attacks on CIFAR-10 considering a variable

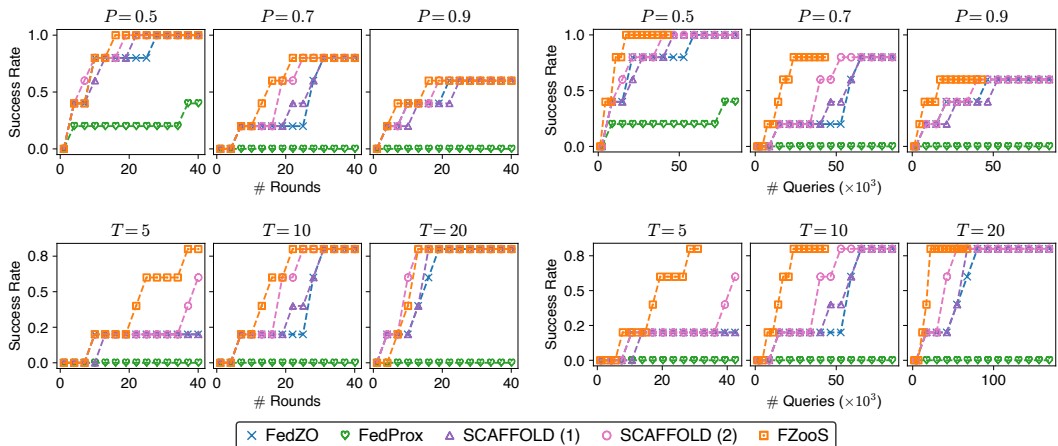

Figure 8: Comparison of the success rate in federated black-box adversarial attack achieved by FZooS and other existing federated ZOO algorithms on MNIST under varying client heterogeneity (controlled by $P \in [0, 1]$, a larger $P$ implies smaller client heterogeneity) and a varying number $T$ of local updates. The $x$ and $y$-axis are the number of rounds/queries and the corresponding success rate (higher is better).

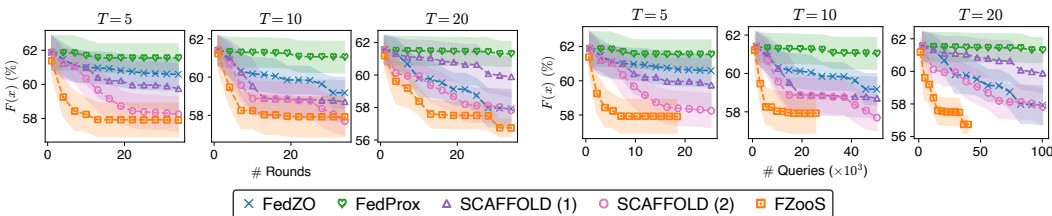

Figure 9: Comparison of the non-differentiable metric optimization between FZooS and other existing federated ZOO algorithms under a varying number $T$ of local updates. Note that the $y$-axis is $(1 - \text{precision}) \times 100\%$ and each curve is the mean $\pm$ standard error from five independent runs.

number of local updates, as showcased in Fig.7. Furthermore, we provide an illustration of the attack success rate on MNIST, considering both varying client heterogeneity and a variable number of local updates, as presented in Fig. 8. Notably, our proposed algorithm consistently demonstrates enhanced efficiency in terms of communication rounds when compared to other baselines, across different levels of client heterogeneity and varying numbers of local updates.

## F.3 FEDERATED NON-DIFFERENTIABLE METRIC OPTIMIZATION

Besides the non-differentiable metric optimization result for the precision score that is under a varying heterogeneity through different $P$ in Fig. 3, we also report the corresponding result under a varying number $T$ of local updates in Fig. 9. Moreover, we provide results for recall, F1 score, and Jaccard as the non-differentiable metric in Fig. 10, Fig. 11, and Fig. 12 respectively. Notably, our FZooS still consistently outperforms other baselines in terms of both communication round efficiency and query efficiency when under the comparison of varying client heterogeneity and a varying number of local updates with different non-differentiable metrics.

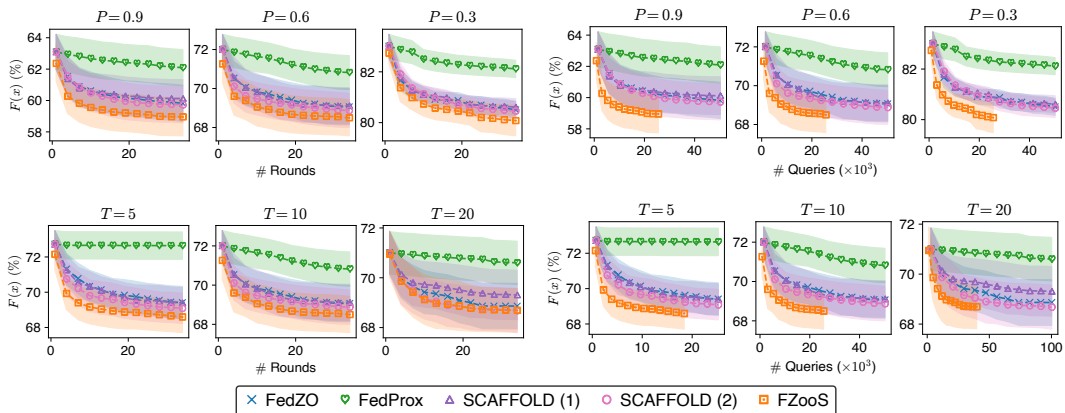

Figure 10: Comparison of the non-differentiable metric optimization between FZooS and other existing federated ZOO algorithms under varying client heterogeneity (controlled by $P \in [0, 1]$, a larger $P$ implies smaller client heterogeneity) and a varying number $T$ of local updates. Note that the $y$-axis is $(1 - \text{recall}) \times 100\%$ and each curve is the mean $\pm$ standard error from five independent runs.

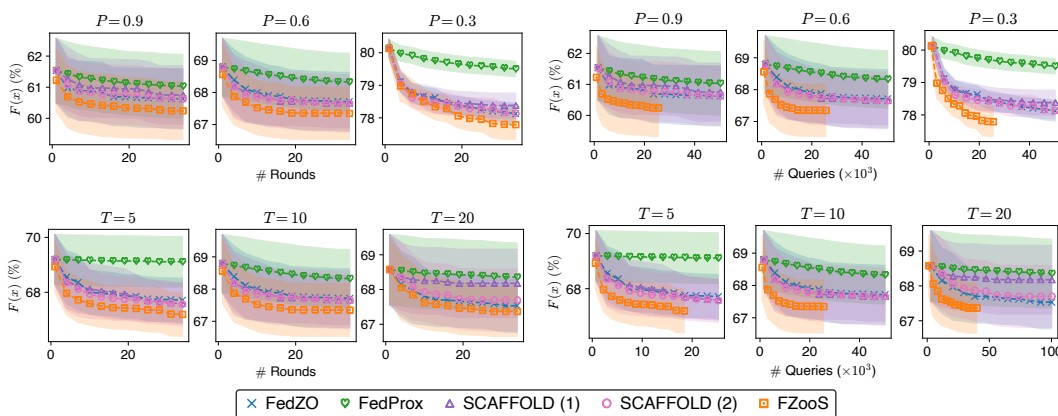

Figure 11: Comparison of the non-differentiable metric optimization between FZooS and other existing federated ZOO algorithms under varying client heterogeneity (controlled by $P \in [0, 1]$, a larger $P$ implies smaller client heterogeneity) and a varying number $T$ of local updates. Note that the $y$-axis is $(1 - \text{F1 score}) \times 100\%$ and each curve is the mean $\pm$ standard error from five independent runs.

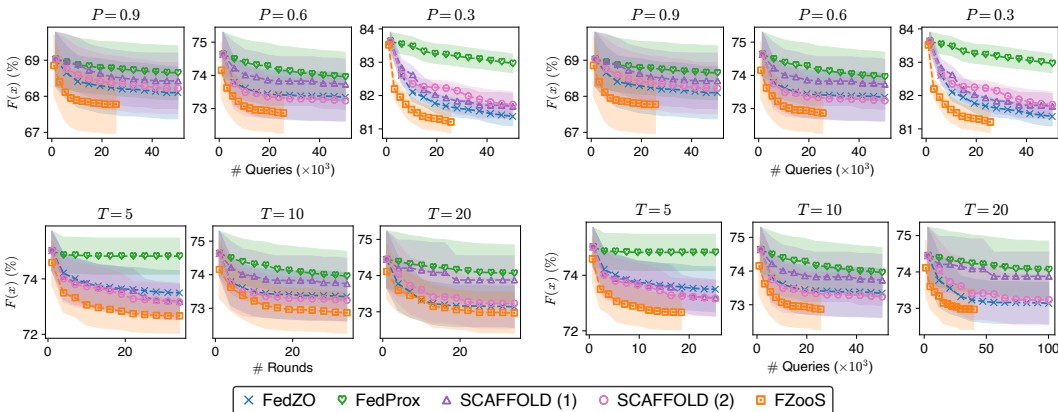

Figure 12: Comparison of the non-differentiable metric optimization between FZooS and other existing federated ZOO algorithms under varying client heterogeneity (controlled by $P \in [0, 1]$, a larger $P$ implies smaller client heterogeneity) and a varying number $T$ of local updates. The $y$-axis is $(1 - \text{Jaccard score}) \times 100\%$ and each curve is the mean $\pm$ standard error from five independent runs.

