# OpenReview forum: "Federated Zeroth-Order Optimization using Trajectory-Informed Surrogate Gradients"
_ICLR.cc/2024/Conference — Submitted to ICLR 2024_

### Official Review · Reviewer_1Jp7 · 2023-10-23

**Soundness:** 3 good
**Presentation:** 3 good
**Contribution:** 2 fair
**Rating:** 3
**Confidence:** 3

**Summary:**

The paper studies federated zeroth-order optimization and introduces a new algorithm FZooS for query and communication-round efficiency. Both theoretical and empirical results show this algorithm outperforms existing federated zeroth-order optimization algorithms.

**Strengths:**

The paper is well-structured and clearly explains every component of the proposed algorithm. The adaptive gradient correction to reduce the disparity might be interesting on its own to the federated optimization community. The zeroth-order method considered in this paper is not based on finite difference or directional derivative as in the optimization literature. The paper finds a way to apply it in federated learning to design a novel algorithm.

**Weaknesses:**

1. Some comparisons with existing works are not fair.

$(a)$ The assumption that each $f_i$ is sampled from a Gaussian process and one has observations to $f_i+\mathcal{N}(0, \sigma^2)$ might be standard in some Bayesian optimization literature, but is not common in federated optimization. All existing algorithms that the paper compares do not make this assumption. Such an assumption is not verified in the real-world experiments this paper provides as well. The paper should review and compare existing works on federated Bayesian optimization that employ the same assumption. Moreover, under which condition does the average of functions sampled from the Gaussian process to be strongly-convex and convex?

$(b)$ The algorithm needs to invert a matrix that can be as large as $RT\times RT$ to compute $\nabla\mu$, which is much more computationally heavy than existing methods. The numerical results are only plotted against the number of rounds, which could be not fair as different algorithms have different operations in one single round.

$(c)$ The algorithm needs to communicate a vector of size at least $1/\epsilon^2$ at each round while existing zeroth-order methods only need to communicate two scalars, one for finite difference, the other one for the random seed to generate the random vector, when $Q=1$.  As per Theorem 2, the total communication cost can be $MR=1/\epsilon^5$ when $F$ is convex. Does existing works have a worse communication cost $QR$ compared to that?

2. Some concerns regarding the theoretical results.

$(a)$ Is $\epsilon$ in Theorem 2 the same as the $\epsilon$ in Theorem 1? The $\epsilon$ convergence error is never defined in the paper. Does it mean $F-F^*$ for the convex case and $\Vert\nabla F\Vert^2$ for the nonconvex case?

$(b)$ Why is there only dependence in the dimension $d$ for the convex setting and no dependence in $d$ for the strongly-convex and nonconvex setting?

$(c)$ What does "$\rho<1$ likely to be satisfied" means? Does it hold with a high probability or require additional conditions? Will this affect the results in Theorem 2? Is this the reason why Theorem 2 only holds with constant probability?

$(d)$ The setting of $\gamma$ requires knowledge of $G$. How to estimate $G$ to run Algorithm 2?

3. Minor: It is not formal to start sentences with any conjunction like "so"; Why is it required in Algorithm 2 that "send $x_r$ first and then $w_r$"? Does the order matter? "urther" at the end of line 15 on page 7 should be "further"?

**Questions:**

See Weaknesses.

---

### Official Review · Reviewer_RiQm · 2023-11-07

**Soundness:** 2 fair
**Presentation:** 2 fair
**Contribution:** 2 fair
**Rating:** 3
**Confidence:** 2

**Summary:**

This paper explores zeroth-order optimization in federated learning and identifies two challenges faced by the existing works: (i) query inefficiency and (ii) communication round inefficiency. To handle these two limitations, the authors propose FZooS, which makes use of trajectory-informed gradient estimation and gradient correction. The paper shows the convergence results of the proposed algorithm and verifies that FZooS reduces the query costs and communication rounds. Additionally, the experiments validate the dominance over the existing works.

**Strengths:**

1. This paper gives very strong theoretical analysis, and the convergence results look promising and intriguing.
2. The experiments show the proposed method outperforms the state-of-the-art baselines.

**Weaknesses:**

1. The problem formulated in Eq (1) is weird. Traditional federated learning optimizes the parameters $x$ in the domain of $\mathbb{R}^d$, and (Fang et al., 2022) optimizes this one as well. However, this paper optimizes the parameters in the domain of $[0, 1]^d$ and assumes $|f_i(x)| \leq 1$. The authors should explain why they formulate the problem different from the existing works.
2. Followed by the first point, the work should make use of proximal methods to restrain the parameter updates. However, the proposed algorithm allows the parameters $x$ updated in the domain of $\mathbb{R}^d$. This conflicts with the problem formulated in Eq. (1).
3. Theorem 2 targets to achieve $\epsilon$-convergence error. When $\epsilon$ approaches 0, $M$ will be very large, which will increase the communication costs, although the total communication rounds diminish.
4. The experiments study a 5/7-client federated learning, which is too small. I suggest the authors enlarge the scale, e.g., the number of clients should be 50/100.

**Questions:**

In addition to the weaknesses above, I have one more question for this work:

SCAFFOLD and FedProx are the first-order optimization approaches. I am not very sure how these methods apply to zeroth-order optimization. It will be helpful if the authors can show the iteratitve local steps (i.e., $t \rightarrow t+1$) and the global aggregation (i.e., $r \rightarrow r+1$) in terms of the model parameters.

---

### Official Review · Reviewer_T6cv · 2023-11-10

**Soundness:** 2 fair
**Presentation:** 3 good
**Contribution:** 2 fair
**Rating:** 5
**Confidence:** 4

**Summary:**

The paper focuses on the problem of improving query and communication round efficiency in federated zeroth-order optimization. The key idea here is to assume that the local functions at clients is sampled from a Gaussian Process (GP). This then implies that the gradients of the local functions follow a derived posterior Gaussian Process. Based on this observation, the authors propose to approximate the local gradient with the mean of the derived Gaussian process, which can be computed using the history of function queries at the local client. This is in contrast to existing zero-order methods which do not use any historical information and instead make multiple function queries to approximate the local gradient. Furthermore, inspired by SCAFFOLD, the authors also propose an adaptive gradient correction technique based on the idea of approximating the kernel function with a finite number of random Fourier features. Experimental results are provided for a synthetic setting, federated black-box adversarial attack and federated non-differential metric optimization.

**Strengths:**

* The novelty of this work lies in a nice combination of several existing orthogonal ideas. Namely, the paper uses the idea of approximating the local gradient with the derived posterior Gaussian Process from Shu et. al, 2023, gradient correction from Karimireddy et al., 2020a (SCAFFOLD paper) and random Fourier features approximation from Rahimi and Recht, 2007.

* The work for the most part is clearly written with good organization and motivation for each step of the algorithm.

* Convergence analysis for the proposed algorithm is provided for the case where the local functions are strongly convex, convex and non-convex.

* Experimental results are good, showing that the proposed FZooS is more query and communication efficient than existing zero-order algorithms in a range of experimental settings.

**Weaknesses:**

* **Computation Efficiency:** I am not convinced that using historical information is the right alternative to additional queries when it comes to computation efficiency in a FL setting. Firstly given that the algorithm runs for $R$ rounds, client $i$ would need to store $x\_{r,t}^{(i)}$ for all $r \in [R], t \in [T]$ leading to a storage cost of $O(RTd)$ bits which can quickly become unsustainable, especially when $d$ is in the millions. Secondly, there is the cost of inverting the kernel matrix at every iteration. I would encourage authors to add a figure showing the total computation time/FLOPS used in FZooS vs FedZO for their experiments.

* **Partial Client Participation:** The authors do not account for partial client participation (which is common in FL settings) either in their analysis or experiments. In my understanding, partial client participation would exacerbate the error in the gradient approximation in FZooS. Firstly, each client would have fewer stored iterates since it participates less. Secondly, the distance between the stored iterates would also increase.

* **Dependence on bounded heterogeneity assumption:** I haven't gone through the proof carefully, but I don't see why the authors need a bounded heterogeneity assumption if they are applying gradient correction. For instance, Theorem 3 in SCAFFOLD does not need any bounded heterogeneity assumption. In fact, the entire motivation for gradient correction is to remove the dependence on $G$. Therefore, seeing a dependence on $G$ in Theorem 2 even after gradient correction seems a bit strange to me.

* **Advantage of adaptive gradient correction:** The advantage of the adaptive gradient correction needs more theoretical justification. In Section 5.2 it would be good to add the rates of convergence when standard gradient correction is used, as in the case of SCAFFOLD. Also, I would like to see more discussion on the benefit of using $\gamma_{r,t-1} = 1$ vs the choice of $\gamma_{r,t-1}$ proposed in Corollary 1.

* **Linear Speedup and dependence on $d$ in Theorem 2:** We do not see an effect of linear speedup in Theorem 2, i.e., $R$ does not reduce as $N$ increases. Also, why does the convex case have a dependence on $d$ but not the strongly convex and non-convex case? Additionally, can the dependence of $M$ on $d^2$ in the convex case be improved? Dependence on $d^2$ makes the communication cost similar to a second order method.

**Questions:**

* Why is the domain $\mathcal{X}$ restricted to $[0,1]^d$ and not $\mathbb{R}^d$?
* Can the authors elaborate on the 'query around $x_r$' step in Line 7 of Algorithm 2? I didn't really understand how this connects with the algorithm since the computation of $\mathbf{w}_{r,T}^{(i)}$ does not depend on $x_r$?
* Please state all assumptions for Theorem 2 in a formal environment. Right now the assumptions are scattered throughout the text.
* How are gradients computed for FedProx and SCAFFOLD? Are these standard query based?
* Why is the performance of FedProx consistently poor across experiments? If we tune the $\mu$ parameter in FedProx, the performance should be at least as good as FedAvg.
* How is zero-order optimization used in the federated black-box adversarial attack?

---

### Meta-Review · Area_Chair_483d · 2023-12-03

**Metareview:**

Paper proposes a new algorithm for Federated Zeroth-order optimization to tackle querying and communication round complexities. It combines three separate ideas from past work on trajectory informed gradients estimators, variance reduction, and random Fourier feature to approximate kernels. Paper is mostly well written and it provides theoretical and empirical justification for reducing communication round and querying complexities.

However, reviewers had many unaddressed concerns including (a) lack of discussion on the computational, memory and communication “size” complexities and hence the practicality, (b) dimension dependence only in convex setting, (c) lack of justification of assumption and comparison to related from Bayesian optimization, (d) experiments lacking other zeroth order methods and larger number of clients, and (e) lack of formality in presenting theoretical results.

**Justification For Why Not Higher Score:**

Potential impracticality and above unaddressed reviewer concerns

**Justification For Why Not Lower Score:**

N/A

---

### Decision · Program_Chairs · 2024-01-16

Reject